# Learning Cellular Dynamics with Cell–Cell Interaction–Aware Optimal Transport

## Abstract

Inferring dynamics from population snapshots is a core challenge in machine learning and biology. In scRNA-sequencing (scRNA-seq), destructive measurements yield irregular, high-dimensional samples of cell states, obscuring how populations evolve. Existing trajectory inference methods either use graph heuristics or cast alignment as an Optimal Transport (OT) problem. However, they treat cells as independent points, ignoring intercellular interactions. In this work, we ask whether incorporating cell–cell interactions can improve the reconstruction of cellular dynamics from scRNA-seq snapshots. We introduce `IADOT` (*Interaction-Aware Dynamic Optimal Transport*), which integrates cell-cell interaction networks into an OT objective and then learns a time-continuous vector field via Conditional Flow Matching. Across a synthetic task and diverse scRNA-seq datasets, we find that incorporating interaction structure can improve snapshot alignment and inference of cellular dynamics versus feature-only baselines. `IADOT` also supports in-silico ligand–receptor perturbation analyses: we show on lung cancer data that inferred trajectories are sensitive to edits of the ligand–receptor catalog, consistent with known effects of targeted pathway inhibition.

## 1 Introduction

Single-cell technologies have turned the study of gene expression into a high-resolution, data-driven science (Picelli, 2016). By exposing cellular heterogeneity directly, these methods are reshaping how we approach complex biological systems (Cha & Lee, 2020). For instance, in embryonic development, they have traced lineage bifurcations that give rise to distinct tissues (Qiu et al., 2022). In oncology, they exposed how cancer populations branch and adapt (Yeo et al., 2022). More broadly, the capacity to measure cellular states at scale calls for computational methods that can recover the underlying dynamical rules of biology (Schiebinger et al., 2019). Importantly, such approaches hold major implications for pharmaceutical research, where experimental campaigns to explore disease mechanisms or evaluate therapeutic interventions are prohibitively costly and time-consuming (Sertkaya et al., 2024). By enabling *in silico* reconstruction and prediction of cellular dynamics, computational models can guide experiment design, prioritize drug targets, and reduce the need for exhaustive laboratory screening (Yue & Dutta, 2022).

**Challenges of inferring cellular dynamics.** Despite these advances, reconstructing cellular dynamics from single-cell measurements presents fundamental difficulties (Bunne et al., 2024). Measurements are destructive: the same cell cannot be followed over time, so there is no one-to-one correspondence between adjacent snapshots. Populations are imbalanced, with varying numbers of cells in each state, making one-to-one mappings ill-suited (Schiebinger et al., 2019). Gene expression measurements are noisy and sampled irregularly, and the ambient dimensionality of thousands of genes exacerbates statistical and computational difficulties (Adil et al., 2021). Reconstructing dynamics from such data means inferring smooth trajectories from noisy, unaligned population snapshots under partial observability.

**Aligning snapshots with Optimal Transport.** Classical trajectory inference constructs a cell–cell $k$NN graph in a low-dimensional embedding and then extracts pseudotime and branches via principal curves, diffusion distances (Haghverdi et al., 2016), or graph geodesics/spanning trees (Street et al., 2018). These locality-based heuristics implicitly assume geometric proximity within a snapshot reflects temporal adjacency and differentiation proceeds along geodesics of the learned mani-

fold. This often results in biased pseudotimes and spurious lineage structure (Saelens et al., 2022). More recent methods (Schiebinger et al., 2019; Bunne et al., 2023b) instead recast cell alignment as a global, uncertainty-aware coupling between *multiple* distributions via Optimal Transport (OT). This formulation has distinct advantages, as it produces soft correspondences, naturally handles unequal sample sizes, and encourages low-action trajectories via its prior.

**Cell-cell interactions.** Conventional OT-based alignment matches cells by minimizing distances in gene-expression space, effectively treating cells as independent points and ignoring the interaction networks that connect them. Consequently, they overlook potential smoothness and directionality in cell–cell interactions (CCIs) unless external structure (e.g., spatial coordinates) is provided (Klein et al., 2025a). This omission is at odds with the central role of directed CCIs in many applications, including pharmacological targeting (He & Xu, 2020; Liu et al., 2023). This motivates the following question: *Can structure derived from CCIs provide useful information to improve the reconstruction of cellular dynamics from scRNA-seq snapshots only?* To answer this, we introduce IADOT (*Interaction-Aware Dynamic Optimal Transport*),

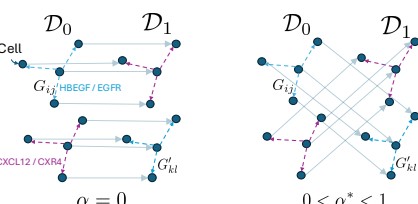

Figure 1: Augmenting feature distances with typed interactions improves alignment between snapshots $\mathcal{D}_0$ and $\mathcal{D}_1$ and encourages transport plans consistent with communication patterns.

a framework that integrates gene-expression features and interaction networks into a single OT objective. IADOT constructs a directed CCI tensor from ligand–receptor expression at each snapshot, and optimizes an OT objective with two components: a feature cost in expression space and a structure cost that favors couplings *preserving specific CCI patterns across time.* The resulting soft couplings align snapshots while respecting the CCI structure, and we use them to learn continuous-time dynamics by training a velocity field via different flow matching techniques.

> **Contributions**
>
> *Conceptually*, we formalize trajectory inference as learning dynamics in interacting subsystems, introducing a structure-regularized OT objective where the persistence of directed, typed interaction networks (e.g., ligand–receptor signaling) serves as a prior. *Technically*, we propose IADOT which learns couplings between snapshots by integrating feature similarity with interaction structure in a multi-dimensional Fused-Gromov Wasserstein objective. Based on these couplings, we then learn continuous-time dynamics of cells by regressing velocity fields with Conditional Flow Matching. *Empirically*, we find on synthetic and diverse single-cell datasets that incorporating directed, typed CCI structure into the OT problem can improve trajectory inference. We also perform in-silico interventions on the LR prior (ablating specific interactions) to assess the dependence of inferred dynamics on the structural assumption.

## 2 BACKGROUND

**Problem formulation: cell trajectory inference.** We consider $k$ population snapshots $\{\mathcal{D}_i\}_{i=1}^k$, where each $\mathcal{D}_i \subset \mathbb{R}^d$ is a set of single-cell states measured at time $t_i$. The goal is to learn a time-continuous flow $\psi : \mathbb{R}^d \times \mathbb{R}_+ \to \mathbb{R}^d$ such that $\psi(x, t)$ returns the state obtained by evolving an initial state $x$ to time $t$. Because scRNA-seq is *destructive*, the same cell cannot be observed at two times, so there is no one-to-one correspondence between cells in $\mathcal{D}_i$ and $\mathcal{D}_{i+1}$. Classical time-series and ODE-fitting methods that require repeated observations of the same object are thus not directly applicable; trajectory inference must instead recover dynamics from *unaligned snapshots*.

**Global alignment of snapshots.** Rather than inferring trajectories from neighborhoods within a single snapshot (Haghverdi et al., 2016), recent work aligns *multiple snapshots at the population level* (Schiebinger et al., 2019), treating each snapshot as a probability distribution over cell states. For two timepoints $t_0 < t_1$ with datasets $\mathcal{D}_0 = \{x_i\}_{i=1}^{n_0}$ and $\mathcal{D}_1 = \{y_j\}_{j=1}^{n_1}$, where $x_i, y_j \in \mathbb{R}^d$ are gene-expression vectors, we form the empirical measures $\rho_0 = \sum_{i=1}^{n_0} a_i \delta_{x_i}$ and $\rho_1 = \sum_{j=1}^{n_1} b_j \delta_{y_j}$, with $a \in \Sigma_{n_0}$, $b \in \Sigma_{n_1}$, and $\Sigma_n := \{w \in \mathbb{R}_+^n : \sum_{k=1}^n w_k = 1\}$ (e.g., $a_i = 1/n_0$ for uniform weights). The alignment problem seeks a coupling $\Gamma^\star$ between $\rho_0$ and $\rho_1$ that respects the marginals, i.e.,

$$\Gamma^{\star} \in \Pi(a,b) := \left\{ \Gamma \in \mathbb{R}_+^{n_0 \times n_1} \mid \Gamma \mathbf{1}_{n_1} = a, \ \Gamma^\top \mathbf{1}_{n_0} = b \right\}, \tag{1}$$

where $\mathbf{1}_n$ is the all-ones vector. Without additional structure, any $\Gamma \in \Pi(a,b)$ is admissible and the problem is underdetermined. Biological priors must therefore rule out implausible matchings. A widely used prior is the *principle of least action*: cell states change smoothly over time, making matchings that incur small feature-wise changes more likely. This recasts snapshot alignment as an Optimal Transport (OT) problem (Villani et al., 2008).

**The static Optimal Transport problem.** Optimal Transport (OT) provides a geometric framework for comparing probability distributions, enforcing *the principle of least-action*. In the context of cell trajectory inference, this principle assumes that the change between snapshots arises from the smallest rearrangement of cellular states consistent with biology: states evolve smoothly in expression space. Given a cost matrix $C \in \mathbb{R}_+^{n_0 \times n_1}$, where $C_{ij} = c(x_i, y_j)$ is the cost of transporting a unit of mass from $x_i$ to $y_j$, the discrete Kantorovich formulation seeks a coupling

$$\Gamma^* \in \arg \min_{\Gamma \in \Pi(a,b)} \langle \Gamma, C \rangle_F \tag{2}$$

where $\langle \cdot, \cdot \rangle_F$ denotes the Frobenius dot product. The optimal coupling $\Gamma^*$ therefore represents the most efficient mapping from a geometric standpoint, as it is defined based on the gene expression profiles. However, by focusing solely on intrinsic state changes, this formulation neglects the extrinsic cell–cell interactions that coordinate population dynamics.

**Incorporating intra-snapshot structure.** Beyond inter-snapshot distances, it is frequent to have access to structural information in each snapshot. However, the optimization problem in Equation (2) does not account for it, as it is purely based on inter-snapshot distances. The Gromov–Wasserstein (GW) problem extends OT to compare two distributions using their *pairwise relational* structure. We assume that this relational structure can be represented by two matrices $G^{(0)} \in \mathbb{R}^{n_0 \times n_0}$ (source) and $G^{(1)} \in \mathbb{R}^{n_1 \times n_1}$ (target). The GW problem seeks a coupling $\Gamma^* \in \Pi(a,b)$ that minimizes the distortion between the intra-domain structure matrices, $G^{(0)}$ and $G^{(1)}$. More precisely, the GW objective is the following quadratic program:

$$\text{GW}(G^{(0)}, G^{(1)}, a, b) = \min_{\Gamma \in \Pi(a,b)} \sum_{i,k=1}^{n_0} \sum_{j,l=1}^{n_1} L(G_{ik}^{(0)}, G_{jl}^{(1)}) \Gamma_{ij} \Gamma_{kl} \tag{3}$$

where $L$ denotes a pairwise distortion function. Finally, it is possible to compare distributions based on *both* their features and their relational structures, combining the Kantorovich and the GW formulations. For a given hyperparameter $\alpha \in [0,1]$, the Fused Gromov-Wasserstein problem is defined by:

$$\text{FGW}_\alpha(G^{(0)}, G^{(1)}, C, a, b) = \min_{\Gamma \in \Pi(a,b)} (1-\alpha) \langle \Gamma, C \rangle_F + \alpha \sum_{i,k=1}^{n_0} \sum_{j,l=1}^{n_1} L(G_{ik}^{(0)}, G_{jl}^{(1)}) \Gamma_{ij} \Gamma_{kl} \tag{4}$$

The parameter $\alpha$ acts as a trade-off, balancing the importance of aligning individual cell features against preserving the structure between cells (controlled by the Gromov-Wasserstein term): setting $\alpha = 0$ recovers the Kantorovich problem, while $\alpha = 1$ recovers the GW problem.

## 3 RELATED WORK

**Distributional alignment for trajectory inference.** Classical trajectory-inference tools reconstruct cellular progressions from neighborhood graphs with pseudotime and branching heuristics (e.g., Monocle 2, DPT, Slingshot, PAGA) (Qiu et al., 2017; Haghverdi et al., 2016; Street et al., 2018; Wolf et al., 2019), typically within a *single* snapshot. Optimal transport (OT) (Villani et al., 2008; Peyré & Cuturi, 2019) provides an alternative that couples *distributions* across timepoints rather than stitching local paths. Waddington-OT (WOT) extends OT to sequences of time-labeled snapshots, estimating adjacent-time couplings (Schiebinger et al., 2019). Continuous-time counterparts such as TrajectoryNet learn neural ODE flows constrained by transport to interpolate distributions over time (Tong et al., 2020). However, these families typically optimize match quality primarily in expression space, treating each cell as an isolated point and overlooking intercellular communication. Table 1 contrasts IADOT with other OT-based methods and an extended discussion is provided in Appendix A.

Table 1: **Comparison of trajectory methods.** *Legend:* ✓ supported, ~ partial, ✗ not supported.

| Method | Dynamic | Trajectories | In-silico Perturbation | Structure-Aware | scRNA Data Sufficient | Reference |
|--------|---------|--------------|------------------------|-----------------|----------------------|-----------|
| PAGA (Scanpy) | ✗ | ✗ | ✗ | ✗ | ✓ | (Wolf et al., 2019) |
| Waddington-OT | ✓ | ~ | ✗ | ✗ | ✓ | (Schiebinger et al., 2019) |
| SCOT | ✗ | ✗ | ✗ | ~ | ✗ | (Demetci et al., 2022a) |
| CellOT | ✗ | ✗ | ✓ | ✗ | ✗ | (Bunne et al., 2023a) |
| OT-CFM | ✓ | ✓ | ✗ | ✗ | ✓ | (Tong et al., 2024) |
| TrajectoryNet | ✓ | ✓ | ✗ | ✗ | ✓ | (Tong et al., 2020) |
| Schrödinger Bridge | ✓ | ✓ | ✗ | ✗ | ✓ | (Hong et al., 2025) |
| scVelo | ✓ | ✗ | ✗ | ✗ | ✓ | (Bergen et al., 2020) |
| IADOT (ours) | ✓ | ✓ | ✓ | ✓ | ✓ | — |

**Structure-aware alignments.** Gromov–Wasserstein (GW) compares samples via their intrinsic geometry, and Fused GW (FGW) optimizes a joint feature+structure objective (Vayer et al., 2020a). In single-cell settings, GW/FGW pipelines typically rely on *undirected $k$NN* graphs that capture generic topology but lack *communication semantics* (Demetci et al., 2022b; Lange et al., 2024). We instead inject a *directed, typed* prior derived from ligand–receptor (LR) expression into an FGW objective. This encourages alignments that preserve signaling context and allows to probe the effect of specific LR interactions on the inferred dynamics. Orthogonal lines of work infer *directionality* from spliced/unspliced counts and propagate it on $k$NN graphs (Bergen et al., 2020). CellRank further combines velocity with transcriptomic similarity to estimate fate probabilities (La Manno et al., 2018; Bergen et al., 2020; Lange et al., 2022). Spatial OT approaches instead exploit *physical proximity* to couple cells and infer possible communication. For example, NicheFlow (Sakalyan et al., 2025) models cells' microenvironment but it assumes access to spatial transcriptomics data. In general, the structure term in spatial OT remains geometric rather than *typed* signaling (Cang & Nie, 2020). Contrasting these works, IADOT tackles the setting where only scRNA-seq data is available. Meta Flow Matching (Atanackovic et al., 2024) learns an amortized vector field by encoding the initial distribution with a graph neural network; however it assumes access to $N \geq 2$ datasets for training, which makes it inapplicable to the setting tackled by IADOT.

**Inductive biases in flow-based modeling.** Recent advances in flow matching have focused on incorporating specific domain priors. MIOFlow (Huguet et al., 2022) and Metric Flow Matching (Kapusniak et al., 2024) impose *geometric* inductive biases, restricting dynamics to the data manifold or a learned Riemannian metric. Other approaches address biological mass conservation: UOT-FM (Eyring et al., 2023) relaxes the exact mass constraint via unbalanced OT to model variable population sizes, while VGFM (Wang et al., 2025) explicitly incorporates *cellular growth* rates into the generative flow. In contrast, IADOT integrates an *orthogonal* prior, based on *directed, typed* ligand–receptor pairs.

## 4 IADOT: INTERACTION-AWARE OPTIMAL TRANSPORT

**Overview.** Our objective is to evaluate whether incorporating a structural prior on cell–cell interactions (CCIs) (specifically, a bias toward transport maps that preserve CCI structure across snapshots) can improve trajectory inference. Accordingly, we introduce *Interaction-Aware Dynamic Optimal Transport* (IADOT), a framework that integrates gene-expression features and interaction networks into a unified OT objective. Given source and target snapshots $\mathcal{D}_0$ and $\mathcal{D}_1$, IADOT proceeds in two stages. It first computes a *static* cross-snapshot coupling representing a probabilistic assignment from source cells to target cells. IADOT enforces two desiderata regarding this coupling: **(D1) Feature coherence**— the coupling should reflect smooth cell evolution in expression space; **(D2) Communication preservation**— the coupling should capture the persistence of some directed CCI geometry based on ligand/receptor expression. IADOT satisfies these two desiderata by optimizing a *Fused Gromov–Wasserstein* objective balancing feature similarity and CCI preservation, yielding a coupling $\Gamma^\star$. In the second stage, IADOT fits a *continuous-time* velocity field from interpolants derived from $\Gamma^\star$ using a *Conditional Flow Matching* loss. We can then integrate this velocity field to obtain cell trajectories starting from any given initial state.

### 4.1 INTERACTION-AWARE TRANSPORT VIA MULTI LR-PAIR FGW

**Modeling cell–cell interactions from scRNA.** Given a ligand–receptor (LR) catalog $\mathcal{P} = \{(l_k, r_k) \mid k \in [K]\}$ of $K$ ligand-receptor pairs and a dataset of $n$ cells, our aim is to construct

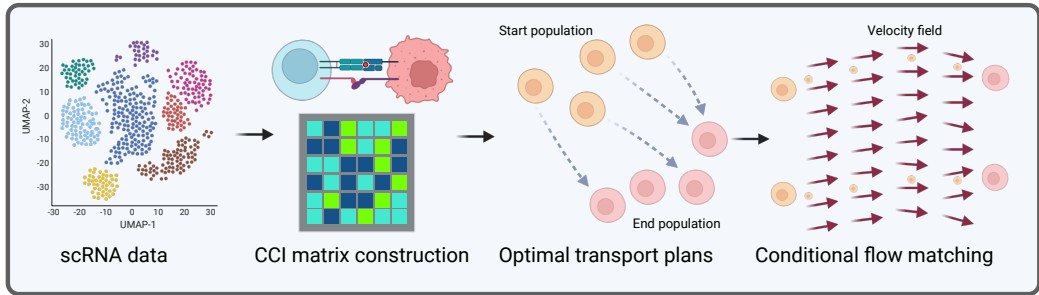

Figure 2: **Overview of IADOT.** From a ligand–receptor catalogue, we build directed, multi LR-pair CCI matrices. A structure-aware optimal transport problem balances feature similarity with interaction structure to produce a cross-snapshot coupling, used to train a time-continuous vector field learned via Conditional Flow Matching to recover cell trajectories.

a directed and nonnegative CCI tensor $G \in \mathbb{R}^{n \times n \times K}$ that summarizes potential signaling from any sender cell $i$ to receiver cell $j$. Starting from raw expression counts, we first apply library-size normalization to make cells comparable. Rather than a $\log(1+\cdot)$ transform, which can suppress biologically meaningful high-expression events, we keep normalized counts and map each gene to $[0,1]$ using a Hill-saturation function. For gene $g$, we then define $s_{cg} = x_{cg}^{h_g}/(x_{cg}^{h_g} + K_g^{h_g})$, with robust scale $K_g$ (e.g., the $q = 0.9$ quantile of nonzero values in $\{x_{cg} \mid c \in [n]\}$ where $x_{cg}$ is the normalized expression of gene $g$ for cell $c$) and exponent $h_g$.

This gives bounded activations where near-saturating expression contributes strongly. For an LR pair $p_k = (l_k, r_k)$ and cells $i$ (sender) and $j$ (receiver), we score the interaction as $q_{i \to j}^{(p_k)} = s_{i\ell_k} s_{jr_k}$, capturing the intuitive requirement that ligand availability and receptor readiness must co-occur. We then define the value of $G$ at $(i, j, k)$ as $G_{ijk} = q_{i \to j}^{(p_k)}$. The CCI tensor $G$ then serves as the directed structure we aim to preserve during cross-snapshot alignment.

*Remark.* For denoising purposes, cells can optionally be aggregated into metacells (e.g., by clustering in a low-dimensional embedding) before constructing the CCI tensors. We empirically evaluate this variant in Section 5.5.

**Interaction-aware transport via multi LR-pair FGW.** Given the two snapshots $\mathcal{D}_0 = \{x_i\}_{i=1}^{n_0}$ and $\mathcal{D}_1 = \{y_j\}_{j=1}^{n_1}$, we define a feature cost matrix $C \in \mathbb{R}_{\geq 0}^{n_0 \times n_1}$, such that for all $i, j$ we have $C_{ij} = c(x_i, y_j)$, where $c$ is typically the squared Euclidean distance. From the CCI construction described above, we obtain directed, nonnegative tensors $(G^{(0)}, G^{(1)})$ corresponding to the source and target snapshots respectively. Our objective is to find a coupling $\Gamma \in \mathbb{R}_{\geq 0}^{n_0 \times n_1}$ that aligns cells while respecting the CCI structures. To jointly account for feature distances and multi-LR pair CCIs, we optimize a *Fused Gromov–Wasserstein* objective that balances a feature term $\mathcal{F}(\Gamma)$ and a structure-preservation term $\mathcal{S}(\Gamma)$ defined with a similarity measure $\varphi$:

$$\min_{\Gamma \in \Pi(a,b)} (1-\alpha) \underbrace{\langle \Gamma, C \rangle}_{\mathcal{F}(\Gamma)} + \alpha \underbrace{\sum_{i,k=1}^{n_0} \sum_{j,\ell=1}^{n_1} \varphi\big(G_{ik}^{(0)}, G_{j\ell}^{(1)}\big) \Gamma_{ij}\Gamma_{k\ell}}_{\mathcal{S}(\Gamma)} . \tag{5}$$

The structure term $\mathcal{S}(\Gamma)$ favours couplings that preserve the CCI patterns encoded in $G^{(0)}$ and $G^{(1)}$. Unlike the classical FGW setting (Vayer et al., 2020b), IADOT handles *multi–typed* interactions: each entry $G_{ij}$ is a vector in $\mathbb{R}^K$ rather than a scalar, allowing multiple LR pairs per cell–cell relation. We compare these interaction vectors with a similarity $\varphi$. By default we use the squared Euclidean norm $\varphi(u, v) = \|u - v\|^2$. Furthermore, Equation (5) is a non-linear and non-convex problem because of the structure term $\mathcal{S}(\Gamma)$. To efficiently solve this non-convex objective, we introduce a customized conditional-gradient solver adapted from (Braun et al., 2022). This tailored optimization routine, detailed in Section D.4, is specifically designed to handle the structural constraints of the interaction-aware coupling.

**Scale normalization.** To balance the feature and structure terms in Equation (5), we *normalize by endpoints*. Concretely, we first solve the feature-only problem ($\alpha = 0$) and the structure-only problem ($\alpha = 1$), obtaining $\Gamma^\star_{\alpha=0}$ and $\Gamma^\star_{\alpha=1}$. We then rescale the feature cost and the CCI tensors using the corresponding objective values at these two optima, so that their magnitudes are comparable (see Section D.5 for more details).

**Unbalanced setting.** The formalism above assumes that the coupling $\Gamma \in \Pi(a, b)$ preserves the marginals $a$ and $b$. In practice, some developmental and perturbation settings exhibit net proliferation or apoptosis between snapshots. To account for that, `IADOT` can be extended to an unbalanced setting by relaxing the hard constraint $\Gamma \in \Pi(a, b)$ with divergence penalties on the row and column sums of $\Gamma$ (more details are given in Section D.11). We provide results with this extension in Section 5.3 and Section E.4.

## 4.2 Learning continuous dynamics via Conditional Flow Matching

**Objective.** The goal of `IADOT` is to learn a time–dependent velocity field that transports the source dataset $\mathcal{D}_0$ to the target dataset $\mathcal{D}_1$, and can be integrated up to any time $t > 0$. We leverage the optimal coupling obtained from Equation (5) to align the two snapshots and convert this *static* correspondence into a *time–dependent* velocity field using Conditional Flow Matching (CFM) (Tong et al., 2024; Lipman et al., 2022). Concretely, we first construct a coupling–induced probability path $\{\rho_t\}_{t\in[0,1]}$ and then fit a velocity field to generate this probability path.

**Probability path.** Let $\rho_0$ and $\rho_1$ denote the empirical distributions defined by $\mathcal{D}_0$ and $\mathcal{D}_1$, respectively. Let $\Gamma^\star \in \mathbb{R}_+^{n_0 \times n_1}$ be the optimal coupling from Equation (5), with normalization constant $M = \sum_{i,j} \Gamma^\star_{ij}$. We define a joint distribution $\Pi$ on $\mathcal{D}_0 \times \mathcal{D}_1$ by $\Pi = \sum_{i=1}^{n_0} \sum_{j=1}^{n_1} \frac{\Gamma^\star_{ij}}{M} \delta_{(x_i, y_j)}$, where $\delta_{(x_i, y_j)}$ denotes the Dirac measure at $(x_i, y_j)$. Therefore, the marginals of $\Pi$ are $\rho_0$ and $\rho_1$. For $t \in [0, 1]$, we then consider the affine interpolation $Z_t = (1 - t)X + tY$, with $(X, Y) \sim \Pi$, and let $\rho_t = \mathcal{L}(Z_t)$ be the distribution of $Z_t$, yielding a probability path $\{\rho_t\}_{t\in[0,1]}$. By construction, $\rho_0$ and $\rho_1$ are the endpoints of this path.

**Learning the vector field with CFM.** Given the coupling–induced path $\{\rho_t\}_{t\in[0,1]}$, we learn a time-dependent velocity field $v_\theta : \mathbb{R}^d \times [0, 1] \to \mathbb{R}^d$ that generates it. For $(X, Y) \sim \Pi$ and $Z_t = (1 - t)X + tY$, the interpolation implies a constant drift across time $u_t(Z_t \mid X, Y) = Y - X$, conditioned on $(X, Y)$.

We train $v_\theta$ by regressing to this drift along the path, yielding the following CFM objective:

$$\mathcal{L}_{\mathrm{CFM}}(\theta) = \mathbb{E}_{\substack{(X,Y)\sim\Pi \\ t\sim\mathrm{Unif}[0,1]}} \left[ \left\| v_\theta(Z_t, t) - u_t(Z_t \mid X, Y) \right\|_2^2 \right] \tag{6}$$

$$= \mathbb{E}_{\substack{(X,Y)\sim\Pi \\ t\sim\mathrm{Unif}[0,1]}} \left[ \left\| v_\theta(Z_t, t) - (Y - X) \right\|_2^2 \right]. \tag{7}$$

Thus, converting the coupling to a velocity field reduces to supervised regression. As shown in (Lipman et al., 2024), the minimizer of this loss generates the probability path $\{\rho_t\}_{t\in[0,1]}$. After training, we can then sample trajectories starting from any point $x \in \mathbb{R}^d$ at time 0 by integrating the ODE $\dot{z}(t) = v_\theta(z(t), t)$ from 0 to $t > 0$, with the initial condition $z(0) = x$.

**Extensions.** Because our CCI prior is defined independently of how the velocity field is parameterized and regressed, it acts as an orthogonal component to the underlying flow-matching objective. In practice, this means that `IADOT` can be used as a plug-and-play prior on top of more advanced flow-matching methods without requiring any change to their architectures or training procedures. For example, we can directly combine `IADOT` with Metric Flow Matching (Kapusniak et al., 2024) by simply replacing the base CFM objective with its metric variant, while keeping the coupling derived from `IADOT`. We detail this in Section D.12 and we report results with this variant in Section 5.3 and Section E.4.

## 5 EXPERIMENTS

We evaluate whether incorporating the cell–cell interaction (CCI) structure improves cross-snapshot alignment and continuous-time trajectory inference over feature-only baselines. In Section 5.1, we present a controlled synthetic study, showing that the solution to the structure-aware OT problem (Equation (5)) can exactly recover the ground-truth transport map. In Sections 5.2 and 5.3, we benchmark IADOT on three scRNA-seq datasets spanning diverse tissues and observe consistent gains over baselines in interpolation metrics, when incorporating the CCIs. We then provide biological insights by performing targeted edits to the ligand–receptor catalog, and quantify the resulting shifts in inferred dynamics in Section 5.4. Finally, we conduct a sensitivity analysis over the CCI construction choices in Section 5.5 and discuss potential failure modes of IADOT in Section 5.6.

### 5.1 SYNTHETIC SETUP

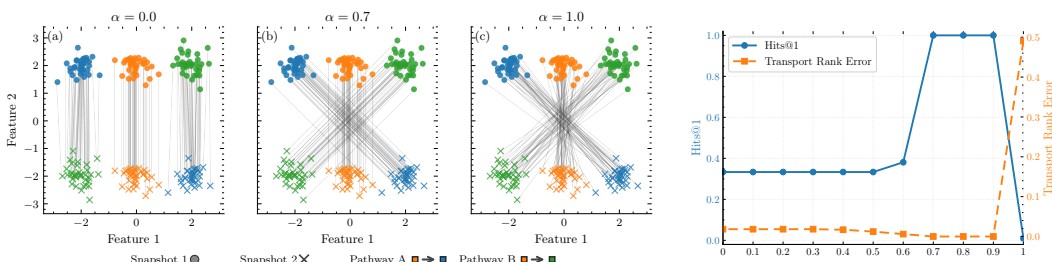

Figure 3: **Structure-aware coupling recovers the ground-truth transport map.** We show representative couplings *(left)* and matching metrics *(right)*. Feature-only OT ($\alpha$=0) ignores structure and misaligns clusters, structure-only ($\alpha$=1) distorts geometry within the interaction types. A balanced trade-off ($\alpha\approx0.7$) recovers the intended one-to-one mapping.

**Setup.** We consider two 2D snapshots, each composed of three clusters. The second snapshot is obtained by translating each cluster by a distinct vector, inducing a known one-to-one ground-truth transport. We define an interaction structure with two types : the middle cluster points to the left *(Type 1)* and to the right *(Type 2)*, mirrored in the target snapshot (see Section C.1 for more details). We then obtain a coupling for each $\alpha \in \{0, 0.1, \ldots, 1\}$ by solving the FGW problem defined in Equation (5) with the ground-truth interaction structures.

**Results.** Representative couplings across $\alpha$ are shown in Figure 3 *(left)*. With $\alpha = 0$ (feature-only), the interaction structure is ignored and clusters are misaligned; with $\alpha = 1$ (structure-only), interaction types are satisfied but geometry is distorted. An intermediate setting ($\alpha \approx 0.7$) preserves the directed relations while maintaining within-interaction geometry. We quantify these observations by computing *Hits@1*, the fraction of source samples whose top-weighted target equals the ground-truth match, and *Transport Rank Error (TRE)*, the average fraction of targets ranked above the ground-truth match. Figure 3 *(right)* shows that *Hits@1* peaks and *TRE* is minimized at mid-range $\alpha$, indicating that a balanced mix of features and structure gives the most faithful transport map. We refer to Section F.1 for a theoretical analysis of this synthetic setup.

### 5.2 CROSS-SNAPSHOT INTERPOLATION FROM STRUCTURE-AWARE COUPLINGS

**Datasets.** We evaluate IADOT on *real-world datasets* whose characteristics are summarized in Table 6. We selected these datasets because their temporal coverage provides a favorable window in which ligand–receptor (LR) interactions are expected to remain approximately persistent. Following standard preprocessing, we project gene-expression profiles onto the top $d = 20$ principal components (Section D.2) and standardize them as in (Tong et al., 2024). Additional details on dataset collection are provided in Section C, and results on further datasets are in Section E.

**Setup.** We build CCI tensors by selecting dataset-specific ligand–receptor pairs via an automated procedure that accounts for stability of expression levels across snapshots (cf. Section D.6 for more details). We then assess the couplings produced by IADOT in an interpolation setup. Given three time points $t_0 < t_1 < t_2$, we hold out the snapshot at $t_1$. Using only $t_0$ and $t_2$, and for a chosen

LR catalog $\mathcal{P}$ and hyperparameter $\alpha \in \{0, 0.1, \ldots, 1.0\}$, we obtain a coupling $\Gamma(\alpha, \mathcal{P})$ by solving the OT problem defined in Equation (5). We define the marginal at $t_1$ by affine interpolation and denote it by $\rho_{t_1}(\alpha, \mathcal{P})$. For each $\alpha$ and $\mathcal{P}$, we compare $\rho_{t_1}(\alpha, \mathcal{P})$ with the empirical distribution $\rho_{t_1}$ observed at $t_1$, computing the Wasserstein-1 and Wasserstein-2 distances $W_1(\rho_{t_1}(\alpha, \mathcal{P}), \rho_{t_1})$ and $W_2(\rho_{t_1}(\alpha, \mathcal{P}), \rho_{t_1})$.

**Results.** We report these metrics in Figure 4. Across datasets, incorporating CCI structure improves alignment, with optimal performance at a dataset-specific $\alpha^* > 0$. We observe two regimes: a U-shaped curve with $0 < \alpha^* < 1$, indicating that combining CCI with feature-only OT is best, and an almost monotonic decrease with a minimum at $\alpha^* = 1$ for the *Dendritic Stimulus* dataset.

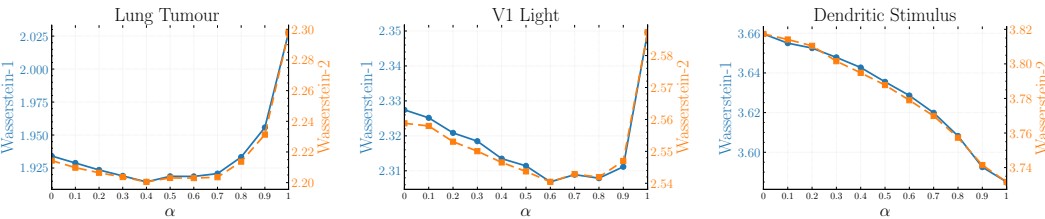

Figure 4: **Interpolation error.** We plot the $W_1$ and $W_2$ distances between the interpolated and empirical $t_1$ snapshots as $\alpha$ varies. Optimal performance occurs at dataset-specific $\alpha^* > 0$.

## 5.3 CROSS-SNAPSHOT TRAJECTORY INFERENCE VIA FLOW MATCHING

**Setup.** Having shown that incorporating structure yields better couplings for these datasets, we now verify whether it also improves continuous-time dynamics learnt with `IADOT`. Starting from an optimal coupling $\Gamma$, we fit a time-conditional vector field $v_\theta$ using the conditional Flow matching loss (Equation (6)). We integrate $v_\theta$ to transport cells observed at $t_0$ to the held-out time $t_1$, and compare the transported distribution to the empirical snapshot at $t_1$ using Wasserstein-1 and -2 distances. We repeat this for $\alpha \in \{0, 0.5, 1\}$. As baselines, we compare against neural ODE–based methods (`TrajectoryNet` (Tong et al., 2020), `MIOFlow` (Huguet et al., 2022)), diffusion-based Schrödinger-bridge methods (`Diffusion Schrödinger Bridges (DSB)` (De Bortoli et al., 2021), `SF2M` (Tong et al., 2023)), flow-matching methods (`MFM` (Kapusniak et al., 2024), `UOT-FM` (Eyring et al., 2023), `VGFM` (Wang et al., 2025)), and `MOSCOT` (Klein et al., 2025b). Since `IADOT` provides a plug-and-play CCI prior that is complementary to the underlying OT or flow-matching objective, we also report "`IADOT+`" variants of several baselines (e.g. `IADOT+MFM`, `IADOT+UOT-FM`, `IADOT+SF2M`). We refer to Section D.12, Section D.11, and Section D.13 for details about these variants.

**Results.** Table 2 reports $W_1$ and $W_2$ at the held-out time $t_1$ (lower is better). Results indicate that the procedure used to fit the velocity field affects performance: conditional flow matching yields consistently lower errors than all the baselines across datasets. Second, within `IADOT`, structure helps: settings with $\alpha > 0$ outperform the feature-only case ($\alpha = 0$), with the best results at $\alpha \in \{0.5, 1\}$. These findings align with Section 5.2, indicating that CCI structure benefits both static alignment *and* the learned continuous-time dynamics. Furthermore, we see that plugging `IADOT` with other priors leads to improved performance, showing the wide benefits of incorporating CCI information.

## 5.4 PROBING TRAJECTORY SENSITIVITY TO LIGAND–RECEPTOR CATALOG EDITS

**Setup.** Having demonstrated that interaction structure guides trajectory inference, we now leverage `IADOT` to simulate intercellular perturbations. We focus on the *Lung Tumor* dataset and construct alternative ligand–receptor catalogs in which specific signaling pathways are ablated. From these modified catalogs, we recompute the CCI tensors and resolve the OT problem (Equation (5)). Crucially, this intervention modifies only the interaction prior governing the coupling, while holding the initial gene expression at $t = 0$ fixed. This design mimics a *pharmacological blockade*, where external signaling is inhibited before cells transcriptionally adapt (Lee et al., 2016), providing a mechanism-specific counterfactual that predicts how the loss of communication redirects the popu-

Table 2: **Interpolation error for continuous time dynamics (lower is better).** `IADOT` with varying structure weight $\alpha$ vs. baselines across the three datasets. We report mean±std over 5 runs.

| Method | $\alpha$ | V1 Light | | Dendritic Stimulus | | Lung tumor | |
|---|---|---|---|---|---|---|---|
| | | $W_1$ | $W_2$ | $W_1$ | $W_2$ | $W_1$ | $W_2$ |
| TrajectoryNet | — | 3.022(0.061) | 3.338(0.056) | 4.410(0.102) | 4.607(0.107) | 2.712(0.090) | 3.056(0.099) |
| DSB | — | 3.819(0.152) | 3.875(0.143) | 4.099(0.155) | 4.249(0.153) | 3.700(0.116) | 3.967(0.102) |
| OT-CFM | — | 2.392(0.005) | 2.625(0.004) | 3.696(0.007) | 3.857(0.009) | 1.993(0.004) | 2.275(0.005) |
| OT-MFM | — | 2.401(0.003) | 2.636(0.003) | 3.714(0.008) | 3.880(0.009) | 1.984(0.004) | 2.285(0.004) |
| UOT-FM | — | 2.411(0.005) | 2.649(0.006) | 3.701(0.006) | 3.867(0.007) | 1.998(0.004) | 2.348(0.004) |
| SF2M | — | 3.254(0.192) | 3.368(0.182) | 4.333(0.279) | 4.436(0.282) | 3.826(0.265) | 3.974(0.308) |
| VGFM | — | 6.446(0.114) | 6.745(0.102) | 7.087(0.022) | 7.261(0.026) | 2.175(0.017) | 2.478(0.019) |
| MIOFlow | — | 6.360(0.010) | 6.655(0.009) | 6.970(0.043) | 7.159(0.034) | 2.001(0.003) | 2.316(0.009) |
| Moscot | — | 6.242(0.000) | 6.545(0.000) | 7.115(0.000) | 7.331(0.000) | 2.000(0.000) | 2.335(0.000) |
| IADOT+SF2M | 0.5 | 3.199(0.117) | 3.315(0.110) | 4.303(0.213) | 4.397(0.205) | 3.809(0.302) | 3.968(0.374) |
| | 1 | 3.226(0.075) | 3.339(0.073) | 4.289(0.110) | 4.387(0.108) | 3.638(0.308) | 3.739(0.335) |
| IADOT+MFM | 0.5 | 2.393(0.007) | 2.631(0.008) | 3.679(0.007) | 3.838(0.009) | 1.978(0.004) | 2.277(0.003) |
| | 1 | 2.363(0.002) | 2.606(0.002) | 3.668(0.010) | 3.824(0.011) | 2.013(0.003) | 2.304(0.003) |
| IADOT+UOT-FM | 0.5 | 2.377(0.004) | 2.619(0.005) | 3.688(0.012) | 3.854(0.012) | **1.971(0.005)** | 2.322(0.005) |
| | 1 | **2.360(0.002)** | 2.605(0.001) | **3.624(0.004)** | **3.780(0.002)** | 1.993(0.004) | 2.335(0.005) |
| IADOT+CFM | 0.5 | 2.381(0.004) | 2.618(0.003) | 3.679(0.009) | 3.835(0.010) | 1.989(0.004) | **2.272(0.005)** |
| | 1 | 2.362(0.003) | **2.601(0.005)** | 3.639(0.021) | 3.788(0.021) | 2.057(0.005) | 2.329(0.005) |

lation's downstream trajectory. We quantify these shifts relative to the unperturbed baseline using the 20 Hallmarks of Cancer gene sets (see Section D.9.1) over a 24h interpolation window.

**Results.** Figure 5 shows the relative decrease in tumour-associated progression scores under different catalog edits. Attenuating signaling through EGFR, ALK, or MET produces measurable reductions (up to 15.5%), indicating that the inferred trajectories are sensitive to these pathways. This aligns with their established therapeutic relevance in non–small cell lung cancer, where EGFR inhibitors (e.g., gefitinib, osimertinib), ALK inhibitors (e.g., crizotinib, alectinib), and MET inhibitors (e.g., capmatinib, tepotinib) are used clinically (Domvri et al., 2013). By contrast, edits to unrelated cardio–renal pathways (RAAS, vasopressin, natriuretic peptides) yield negligible changes, suggesting that `IADOT` responds specifically to biologically relevant ligand–receptor structure rather than arbitrary perturbations.

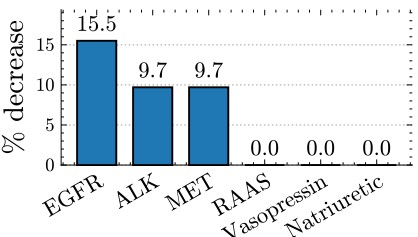

Figure 5: **In *silico* interventions.** Edits to the LR catalog (i.e. removing EGFR/ALK/MET interactions in the CCIs) reduce a Hallmark-based tumour progression proxy.

### 5.5 SENSITIVITY TO CCI CONSTRUCTION CHOICES

**Setup.** Motivated by the previous observation that changing the CCI structure affects the learnt dynamics, we now conduct a sensitivity analysis via three controlled perturbations of the CCI construction process: *Random LR catalog*—replace the curated ligand–receptor (LR) catalog with a random subset of the same size as the initial catalog; *Shuffling*—randomly permute all entries of the CCI tensors, destroying coherent structure; *Metacells*—aggregate cells into metacells before constructing CCIs and then lift interactions back to the cell level (see Section D.3 for details), thereby smoothing the signal. We evaluate all variants under the interpolation protocol of Section 5.2 with $\alpha = 1$ (structure-only OT) to isolate structural effects.

Table 3: **Sensitivity analysis on the CCI construction.**

| Method | Tumor | | Dendritic | | Light | |
|---|---|---|---|---|---|---|
| | $W_1$ | $W_2$ | $W_1$ | $W_2$ | $W_1$ | $W_2$ |
| *Shuffle* | 2.181 | 2.378 | 3.637 | 3.760 | 2.434 | 2.644 |
| *Random LR* | 2.186 | 2.408 | 3.587 | 3.722 | 2.441 | 2.646 |
| *Metacell* | 2.054 | 2.345 | 3.575 | 3.722 | 2.327 | 2.564 |
| IADOT | 2.028 | 2.298 | 3.585 | 3.732 | 2.350 | 2.587 |

**Results.** Table 3 summarizes the results. Shuffling the CCI leads to a performance drop, confirming that the *structural organization* of LR interactions drives the gains. Using a random LR catalog also degrades the interpolation, highlighting the importance of *ligand–receptor specificity*. The CCI constructed with metacells yields intermediate results, as it improves results on two of the datasets. This can be attributed to its smoothing role, especially useful against dropout effect. However, it is

not always optimal, as oversmoothing can bias the CCI and degrade performance, an observation consistent with previous spatiotemporal analyses (Klein et al., 2025a).

### 5.6 DOES STRUCTURE ALWAYS IMPROVE CROSS-SNAPSHOT ALIGNMENT?

**Setup.** Our OT formulation penalizes couplings that do not preserve the CCI structure between two snapshots. Therefore, it assumes that this structure is at least approximately persistent across snapshots. When the system undergoes rapid and large-scale remodeling, this assumption can fail and the induced structure may no longer be informative. We illustrate this with a developing mouse embryo dataset (Moon et al., 2019), where tissue composition, size, and function change quickly during development (Qiu et al., 2024). Furthermore, the time interval between consecutive snapshots is substantial (6 days). As such, we expect the CCI structure at one stage to be poorly related to the next.

**Results.** We report the interpolation results for this dataset in Figure 6. In this setting, IADOT offers no additional gains over feature-only OT ($\alpha = 0$), confirming that the CCI structure is not transferable across these days-long developmental intervals and thus becomes uninformative. This leads to an essentially flat curve with respect to $\alpha$, with worse degradations at larger $\alpha$. It also yields a practical guideline: when cross-snapshot interaction geometry does not persist over time, the structural term should be downweighted or omitted.

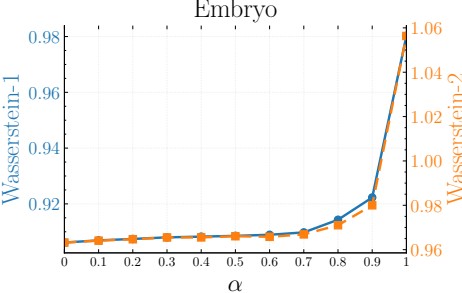

Figure 6: **Embryo dataset.** Incorporating CCIs does not improve performance over the feature-only baseline ($\alpha = 0$).

## 6 DISCUSSION

**Identifiability with biological structure.** Standard optimal transport relies solely on feature similarity to align populations, a formulation often insufficient for the high-dimensional, sparse, and non-linear nature of single-cell dynamics. In this work, we introduced IADOT, a framework that addresses this fundamental identifiability problem by injecting a biologically grounded inductive bias: the persistence of ligand–receptor communication. By formulating alignment as a multi-channel Fused Gromov-Wasserstein problem, IADOT extracts directed signaling topologies and recovers trajectories that are not only geometrically smooth but also preserve CCIs. Because structure is an explicit, editable prior, IADOT enables principled counterfactuals: we can quantify resulting shifts in inferred trajectories when perturbing pathway-specific LR libraries in the CCI construction.

**A modular prior.** A key strength of IADOT is its modularity. Rather than being tied to a specific dynamics model, it serves as a flexible "plug-and-play" cost function. Our extensive experiments demonstrate that the IADOT coupling consistently improves performance across diverse paradigms, including deterministic transport (CFM), geometric interpolation (Metric Flow Matching), stochastic bridges (SF2M), and unbalanced mass transport (UOT). This universality suggests that interaction-aware priors are orthogonal and complementary to recent advances in generative modeling, offering a generic recipe to refine any OT-based trajectory inference method. Finally, our ablation study shows that *biologically meaningful* interaction matrices are important and drive the observed performance gains.

**Limitations.** We stress that IADOT is not a silver bullet. Our approach assumes that key interaction structure is at least partly conserved between adjacent snapshots. In rapidly remodeling systems (e.g. embryo development), we have shown that incorporating the structure does not necessarily yield benefits. Furthermore, we evaluated IADOT on real scRNA-seq datasets from prior studies, but broader *scalability* to atlas level datasets is an interesting avenue for future work.

**Broader impact.** IADOT offers a simple recipe to inject typed interaction priors to disambiguate alignments. Beyond biology, IADOT offers a principled path to modeling dynamics in systems of interacting entities including financial markets, social networks and multi agent environments where structure aware couplings can improve alignments.

REPRODUCIBILITY STATEMENT

We specify the OT objective in Section 4.1, and detail the multi-channel FGW solver as well as the normalization used to balance feature and structure terms in Sections D.4 and D.5. Our continuous-time dynamics and Conditional Flow Matching objective are described in Section 4.2, with model architectures and all training hyperparameters listed in Section D.7. Datasets, sampling timepoints, and sizes are summarized in Section C, and the end-to-end preprocessing pipeline is documented in Section D.2. The construction of ligand–receptor catalogs is detailed in Section D.6. Baselines use authors' implementations with exact settings listed in Section D.8. Software versions and key libraries are reported in Table 7. Code will be released upon acceptance.

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

# APPENDIX

## A  Extended related works

**Inferring cellular trajectories.**   Methods for trajectory inference differ in both their assumptions and the temporal scales they target. Pseudotime approaches order cells along low-dimensional embeddings (such as UMAP), capturing smooth expression trends but relying on manifold geometry rather than explicit dynamical models (Erbe et al., 2023). RNA velocity augments expression with spliced and unspliced counts to estimate short-term directional change (typically minutes to hours), but its accuracy degrades over longer horizons and in the presence of sparse signals (Chen et al., 2022). More recent formulations use continuous-time models to interpolate between snapshots: Neural ODEs and dynamic optimal transport (OT) learn flows across cell states, with variants tailored to gene-regulatory-network dynamics or intervention-aware recovery in scRNA-seq (Lin et al., 2025), and others incorporating biological priors to regularize the inferred trajectories (e.g., PHOENIX) (Hossain et al., 2024). DeepVelo applies Neural ODEs to high-dimensional, sparse measurements, yielding predictive flows without committing to a mechanistic model (Chen et al., 2022). Related work couples dynamics with OT on learned manifolds to better respect transcriptomic geometry during alignment (Huguet et al., 2022). A closely related line casts dynamics as a Schrödinger bridge used for trajectory inference and generative modeling in single-cell RNA data (Hong et al., 2025).

**Optimal transport in biology.**   Optimal transport (OT) is widely used for static alignment of cellular populations in biology and has been extended to dynamic settings for modeling complex scRNA-seq trajectories (Tong et al., 2020). High dimensionality is a central challenge, and remedies include dimensionality reduction, regularization, and scalable solvers (Cuturi et al., 2023), often operating in learned or low-dimensional representations (e.g. PCA or manifold embeddings) to better respect transcriptomic geometry (Huguet et al., 2022). OT-based frameworks have been applied to recover cellular trajectories in development (Schiebinger et al., 2019), and incorporating inductive biases (such as lineage information) can further improve identifiability and accuracy (Forrow & Schiebinger, 2021). Recent work relates OT and continuous-time dynamics via flow matching Klein et al. (2024) and explores multi-modal integration directly within the OT formulation (Klein et al., 2025a). An overview of different OT-based trajectory methods and their mathematical fomulation is provided in Table 4.

Table 4: **OT-based methods: objectives and assumptions.** BB = Benamou–Brenier; FGW = Fused Gromov–Wasserstein; SB/DSB = (Diffusion) Schrödinger Bridge.

| Method | Static / dynamic | OT formulation | Optimization objective (schematic) | Structure used | Timepoints | Data assumed |
|---|---|---|---|---|---|---|
| Waddington-OT (Schiebinger et al., 2019) | dynamic (discrete) | Unbalanced entropic Kantorovich | $\min_\Gamma \langle \Gamma, C \rangle + \tau \mathrm{KL}(\Gamma \mathbf{1} \| a) + \tau \mathrm{KL}(\Gamma^\top \mathbf{1} \| b)$ | Growth priors (no LR) | $\geq 2$ | scRNA-seq (+growth) |
| SCOT (Demetci et al., 2022b) | static | Fused GW | $\min_\Gamma \langle \Gamma, C \rangle + \lambda \langle L(D^x, D^y), \Gamma \otimes \Gamma \rangle + \varepsilon \mathrm{H}(\Gamma)$ | $k$-NN geometry (untyped) | 2 | scRNA-seq |
| CellOT (Bunne et al., 2023b) | static | Kantorovich (dual/convex) | $\max_f \min_g \hat{C} - \mathbb{E}_{\rho_s}[\langle x, \nabla g(x) \rangle - f(\nabla g(x))] - \mathbb{E}_{\rho_t}[f(y)]$ | None | 2 | scRNA-seq |
| TrajectoryNet (Tong et al., 2020) | dynamic (continuous) | BB OT + neural ODE prior | $\min_\theta -\sum_i \log P_{t_i}(x_{t_i}) + \int \|v\|^2 dt + \text{bio priors}$ | None | $\geq 2$ | scRNA-seq (often splicing) |
| OT-CFM (Tong et al., 2024) | dynamic (continuous) | Kantorovich + flow matching | $\min_\Gamma \langle \Gamma, C \rangle$ ; $\min_\theta \mathbb{E}_{(x,y) \sim \Gamma} \|v_\theta - v^{\mathrm{OT}}\|^2$ | None | 2 | scRNA-seq |
| Schrödinger Bridge (DSB) | dynamic (stochastic) | SB/DSB | $\min_{p_{(x_0,T)}} \mathrm{KL}(p \| \text{ref-diffusion})$ s.t. $p_0 = \mu_0,\ p_T = \mu_T$ | None | $\geq 2$ | scRNA-seq (+noise model) |
| **IADOT (ours)** | dynamic (continuous) | Multi-channel FGW + CFM | $\min_\Gamma (1-\alpha)\langle \Gamma, C \rangle + \alpha \langle \varphi(G^{(0)}, G^{(1)}), \Gamma \otimes \Gamma \rangle$ ; $\min_\theta \text{CFM}(\theta \mid \Gamma)$ | **Typed, directed LR CCI** | 2 (extendable) | scRNA-seq |

**Interaction modeling between cells**   A line of work seeks to infer cell–cell communication directly from single-cell gene expression, using curated ligand–receptor (LR) knowledge to score putative interactions between sender–receiver pairs (Browaeys et al., 2019). Tools such as Cell-PhoneDB systematically enumerate LR co-expression across cell types (Efremova et al., 2020), and related approaches have been extended to spatial transcriptomics to incorporate physical proximity as an additional constraint on feasible communications (Cang et al., 2023). Beyond purely geometric priors, multi-modal OT frameworks like MOSCOT can integrate diverse structure (e.g. spatial adjacency) into the coupling itself (Klein et al., 2025a). Finally, meta-frameworks like LIANA+ unify and standardize CCI scoring across multiple LR resources and methods, facilitating method-agnostic comparisons and consensus analyses (Dimitrov et al., 2024a).

**Comparison with Related Works**   Table 1 compares our proposed framework against existing state-of-the-art methods across five key capabilities essential for modeling complex cellular dynamics. We define these criteria as follows:

- **Dynamic** indicates whether the method explicitly models temporal evolution across multiple experimental timepoints, as opposed to inferring dynamics or trajectories from a single static snapshot.
- **Trajectories** distinguishes methods that recover a continuous smooth path enabling predictions at unobserved intermediate timepoints from those that solely compute discrete couplings or transport maps between timepoints.
- **In-silico Perturbation** refers to the capability to perform principled interventions, allowing users to simulate and predict the system's response to specific stimuli or perturbations.
- **Structure-Aware** assesses whether the optimization objective explicitly models interactions between cells (e.g., via cell-cell communication or topological constraints) rather than treating cells as independent, isolated entities.
- **scRNA Data Sufficient** confirms whether the method can operate effectively using standard single-cell RNA sequencing inputs alone, without requiring auxiliary spatial transcriptomics data or multi-modal integration that are often unavailable.

## B Potential applications of IADOT

Snapshots of cellular systems using single-cell RNA sequencing are now pervasive across diverse areas of biology and medicine. A few representative longitudinal datasets are summarized in Table 5. IADOT provides a principled framework to analyze such data by combining snapshot measurements with biologically typed ligand–receptor structure. This enables the reconstruction of coherent cell-state trajectories through optimal transport couplings and a learned continuous flow, as well as the exploration of counterfactual scenarios by selectively re-weighting interaction channels. The resulting outputs (shifts in lineage fate, changes in pathway usage, and differences in progression timing) offer interpretable readouts that can guide mechanistic hypotheses and help prioritize therapeutic strategies before experimental validation.

Table 5: Public longitudinal single-cell datasets. Each row lists an application area, a brief description, and representative studies/accessions (not exhaustive).

| Area | Dataset description ($\geq$3 timepoints) | References / accessions |
|---|---|---|
| Virology | PBMC/tissue scRNA-seq across acute, peak critical or challenge series (D0, D1–3, D7+). | Dengue virus: (Zanini et al., 2018) Influenza: (Arunachalam et al., 2021) |
| Neurology | Brain single-cell timecourses including immune infiltration and glial responses. | Brain organoids: (Camp et al., 2015) |
| Cardiology | Heart/aorta scRNA-seq after myocardial infarction (e.g., D1, D3, D7) or atherosclerosis progression (early $\rightarrow$ intermediate $\rightarrow$ late). | Post-MI hear: (Farbehi et al., 2019) Atherosclerosis: (Pan et al., 2020) |
| Immunology | Tissue + immune scRNA-seq across baseline $\rightarrow$ active disease $\rightarrow$ remission/recovery in model systems. | Lung: (Goldfarbmuren et al., 2020) |
| Development | Human iPSC/hPSC differentiation series (e.g., D0, D4, D8, D12/15), tracking lineage commitment and maturation. | Cardiomyocytes: (Strober et al., 2019) Blood cells: (Tusi et al., 2018) |
| Regeneration | Liver/kidney/muscle injury timecourses (e.g., 0h, 24h, 48h/96h; or 0d, 2d, 5d, 7d) capturing repair trajectories. | Liver injury: (Chen et al., 2023) |

## C Datasets

In addition to the synthetic dataset, we used 5 real-world scRNA datasets to showcase the effectiveness and limitations of our method. Details on the number of genes and the number of cells in each dataset can be found in Table 6.

Table 6: **Datasets used in our experiments.** Counts reflect the preprocessed objects used by `IADOT`. "Timestamps (h)" lists observed hours.

| Dataset | Reference | Timestamps | #Cells | #Genes |
|---|---|---|---|---|
| Tumour | – | 0, 8, 24, 168 (h) | 31,536 | 22,681 |
| V1 Cortex | (Hrvatin et al., 2018) | 0, 1, 4 (h) | 6,505 | 17,008 |
| Immune Stimulus | (Wierenga et al., 2022) | 0, 1, 2, 4, 6 (h) | 2,382 | 10,972 |
| Mouse embryo | (Moon et al., 2019) | 0, 6, 12, 18, 24 (d) | 18,203 | 17,789 |
| Macrophage Stimulus | (Shalek et al., 2014) | 0, 3, 5 (h) | 223 | 478 |

## C.1 SYNTHETIC EXAMPLE

In this section we detail the synthetic setup used in Section 5.1. We construct $\mathcal{D}_0$ as three 2D Gaussian clusters,

$$\mathcal{D}_0 = \bigcup_{k=0}^{2} \mathcal{S}_k, \qquad \mathcal{S}_k = \{X_i^{(k)}\}_{i=1}^{35}, \qquad X_i^{(k)} \overset{\text{i.i.d.}}{\sim} \mathcal{N}(\mu_k,\, 0.1\, I_2),$$

with centers $\mu_0 = (-2, 2)$, $\mu_1 = (0, 2)$, and $\mu_2 = (2, 2)$. The target snapshot $\mathcal{D}_1 = \bigcup_{k=0}^{2} \mathcal{S}_k'$ is obtained by translating each cluster via

$$T_0(x) = x + (4, -4), \quad T_1(x) = x + (0, -4), \quad T_2(x) = x + (-4, -4),$$

so that $\mathcal{S}_k' = \{T_k(X) : X \in \mathcal{S}_k\}$.

For structure, we define two-channel, directed relation tensors $G, G' \in \{0, 1\}^{105 \times 105 \times 2}$ over $\mathcal{D}_0$ and $\mathcal{D}_1$, respectively. Writing $G^{(c)}$ for channel $c$, we set

$$G_{ij}^{(1)} = \mathbf{1}\{X_i \in \mathcal{S}_1,\ X_j \in \mathcal{S}_0\}, \qquad G_{ij}^{(2)} = \mathbf{1}\{X_i \in \mathcal{S}_1,\ X_j \in \mathcal{S}_2\},$$

with $G'$ defined analogously on $\mathcal{D}_1$. Thus, channel 1 encodes $\mathcal{S}_1 \to \mathcal{S}_0$ and channel 2 encodes $\mathcal{S}_1 \to \mathcal{S}_2$.

## C.2 LUNG TUMOR

We use a scRNA-seq dataset to study rapid tumour progression driven by RAS–MYC signalling using a $Kras^{\text{G12D}}$ lung tumour model with tamoxifen-inducible MycER. Samples were collected at 0 h (vehicle), 8 h, 24 h ($n = 8$ biological replicates per condition; 0 h is time zero). Lungs from LSL-$Kras^{\text{G12D}}$ (Jackson et al., 2001) and LSL-$Rosa26^{\text{MIE/MIE}}$ (*MycERT2*) mice (Murphy et al., 2008) were dissociated to single cells, red blood cells removed, filtered (70 $\mu$m), and 6,000 cells per sample were loaded for 10x Chromium $3'$ v3 libraries. Libraries were sequenced on a NovaSeq 6000 and processed with Cell Ranger v6.1.1 against `mm10`. All animal work complied with institutional ethical regulations.

## C.3 V1 CORTEX—LIGHT STIMULATION

Adult (6–8 week) mice were dark-adapted for 7 days, then either euthanized in darkness (0h, control) or exposed to ambient light for 1h or 4h (Hrvatin et al., 2017). The visual cortex was profiled by scRNA-seq to capture early transcriptional responses to sensory input. We treat 0h as the source snapshot, 4h as the target snapshot, and use 1h as an intermediate timepoint for interpolation/validation. After filtering and subsampling we are left with 6505 cells.

## C.4 IMMUNE

To probe innate immune modulation, we use scRNA-seq of murine fetal liver–derived macrophages exposed to LPS with or without 24 h pre-treatment by docosahexaenoic acid (DHA, 25 $\mu$M) (Wierenga et al., 2022). Cells were collected at 0 h (vehicle), 1 h, and 4 h after LPS (20 ng/mL) and sequenced on the 10x Chromium platform. We use 0 h as source, 4 h as target, and 1 h for interpolation/validation; when comparing conditions, we stratify by DHA vs. vehicle and subsample to balance groups.

Table 7: **Software stack (key Python packages).**

| | |
|---|---|
| **Core scientific Python** | NumPy 2.2.6; SciPy 1.16.1; pandas 2.3.1; scikit–learn 1.7.1; numba 0.61.2; matplotlib 3.10.5; seaborn 0.13.2 |
| **Deep learning / training** | PyTorch 2.8.0; PyTorch Lightning 2.5.3; torchmetrics 1.8.1; Triton 3.4.0; Hydra–core 1.3.2; Omega-Conf 2.3.0 |
| **Optimal transport / geometry** | POT 0.9.5; GeomLoss 0.2.6; Graphtools 1.5.3 |
| **Single–cell analysis** | Scanpy 1.11.4; anndata 0.12.2; scVelo 0.3.3; harmonypy 0.0.10; UMAP–learn 0.5.9.post2; PHATE 1.0.11; igraph 0.11.9; leidenalg 0.10.2; networkx 3.5; OmniPath 1.0.12; pypath–omnipath 0.16.20 |

## C.5 EMBRYO DEVELOPMENT

In Section 5.6, we analyze a human embryoid body (EB) differentiation time course used in Moon et al. (2019), which profiles human embryonic stem cells differentiating toward germ layers over 27 days by scRNA-seq. We use the first (Day 0) and third (Day 12) snapshot to infer the cellular dynamics, reserving data at Day 6 for interpolation/validation.

## C.6 MACROPHAGE STIMULUS

To evaluate robustness across experimental platforms, we deliberately included datasets generated with multiple scRNA-seq technologies: 10x Chromium (droplet-based, whole-transcriptome) and BD Rhapsody (microwell-based, often targeted or lower-depth whole-transcriptome). This cross-platform design allows us to test whether our method generalizes despite differences in capture chemistry, library preparation, and typical read depth, which can affect UMI yield and the number of detected genes per cell. All datasets were processed through a consistent downstream pipeline to ensure comparability. We use a macrophage stimulus–response time series that profiled single-cell dynamics across three polarization states (M0, M1 via IFN$\gamma$, M2 via IL-4) responding to six immune ligands (LPS, poly(I:C), CpG, PCSK3) (Shalek et al., 2014). Cells were sampled at 0 h (baseline) and multiple post-stimulation time points (15/30 min, 1 h, 3 h, 5 h, 8 h), (BD Rhapsody). For our alignment tasks we treat 0 h as the source snapshot, 5 h as the target, and use 3 h as intermediate validation points. Experimental results for this dataset can be found in Section E.1.

# D Experimental details

In what follows, we provide details about our experiments presented in Section 5. Code will be released upon acceptance.

## D.1 SOFTWARE AND LIBRARIES USED

We provide in Table 7 the main Python packages we used.

## D.2 DATA PRE-PROCESSING

Raw scRNA-seq files for all datasets were converted to AnnData to standardize processing. We applied basic QC, removing cells with $< 300$ detected genes and genes expressed in $< 3$ cells. Counts were library-size normalized per cell (fixed total). We then selected the 2000 highly variable genes and computed a 20-component PCA on these features. Finally, we performed Harmony batch correction in PCA space (retaining both corrected and uncorrected embeddings for downstream analyses).

## D.3 CONSTRUCTING CCIS USING METACELLS

We detail how we construct CCIs using metacells in the ablation presented in Section 5.5. Without loss of generality and to keep the presentation simple (with matrix multiplications), we assume $K = 1$ (i.e., one LR pair) reducing the CCI tensors to matrices. Before constructing the CCI matrices, we cluster the cells in each snapshot using Leiden community detection on a $k$-nearest-neighbour (kNN) graph built from the PCA representations with Euclidean distances and $k = 10$. An example of the Leiden clustering with subsequent cell annotations is provided in Figure 7. We

select the resolution $\rho^\star$ by scanning a small grid of resolutions and choosing the value whose *median* cluster size is closest to a target of $n^\star = 40$ cells.

Let $S \in \mathbb{R}_{\geq 0}^{n \times g}$ be the membership matrix of the resulting $g$ clusters (rows sum to 1, are correspond to one-hot assignments). We obtain metacell-level activations by averaging the $s_{cg}$ within clusters and form the metacell CCI in $\mathbb{R}^{g \times g}$ similarly as in the setting with individual cells.

Having constructed the metacell CCI matrix $\bar{G}$, we lift it back to the cell level via

$$\tilde{G} = S\,(S^\top S)^{-1}\,\bar{G}\,(S^\top S)^{-1}\,S^\top,$$

This lifting operation ensures $S^\top \tilde{G} S = \bar{G}$. In contrast to $G$, the matrix $\tilde{G}$ is constrained to lie in the subspace $\{SMS^\top \mid M \in \mathbb{R}^{g \times g}\}$, i.e., cell–cell interactions in $\tilde{G}$ are entirely mediated by metacell–metacell interactions.

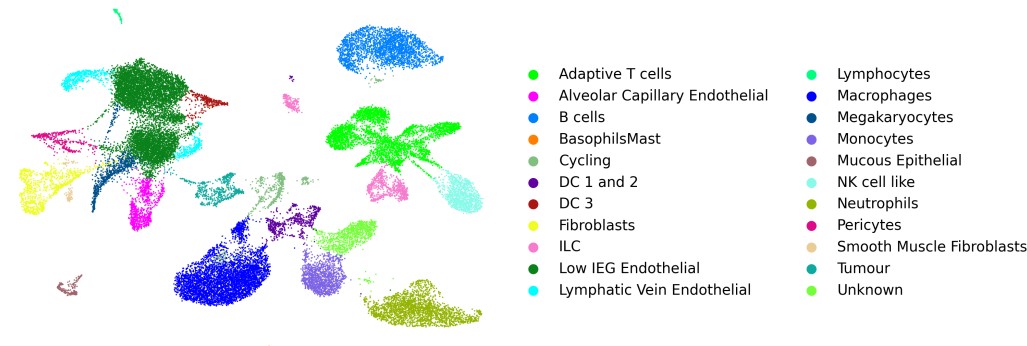

Figure 7: **Metacell construction example.** UMAP visualization of the single-cell RNA-seq data of the lung cancer dataset after Leiden clustering. Each point corresponds to an individual cell, colored by its assigned cluster and annotated with the corresponding cell type based on marker genes.

### D.4 OPTIMAL TRANSPORT SOLVER

We extend POT's (Flamary et al., 2024) conditional-gradient (Frank–Wolfe) solver to handle multi-channel interactions. Given structure tensors $C_1 \in \mathbb{R}^{n_s \times n_s \times d}$, $C_2 \in \mathbb{R}^{n_t \times n_t \times d}$, marginals $p, q$ (uniform by default), and a matrix $\Sigma \succeq 0 \in \mathbb{R}^{d \times d}$, we measure discrepancies with the Mahalanobis norm $\|x\|_\Sigma = \sqrt{x^\top \Sigma^{-1} x}$.

Let $\langle A, B \rangle = \sum_{i,j} A_{ij} B_{ij}$ and write $C^{(r)}$ for the $r$-th channel of a structure tensor $C$. The GW quadratic term is

$$\mathcal{Q}(\Gamma) = \sum_{i,k,j,l} \big\| C_1[i,k] - C_2[j,l] \big\|_\Sigma^2 \Gamma_{ij} \Gamma_{kl} = \langle \mathrm{constC}, \Gamma \rangle - \langle \mathcal{B}(\Gamma), \Gamma \rangle,$$

with

$$\mathrm{constC}_{ij} = \sum_k \|C_1[i,k]\|_\Sigma^2 p_k + \sum_l \|C_2[j,l]\|_\Sigma^2 q_l, \qquad \mathcal{B}(\Gamma) = \sum_{r=1}^d C_1^{(r)}\, \Gamma \,\big(C_2^{(r)}\big)^\top.$$

The gradient computed by the solver is

$$\nabla \mathcal{Q}(\Gamma) = 2\big(\mathrm{constC} - \mathcal{B}(\Gamma)\big) \tag{8}$$

We keep POT's CG loop, stopping criteria, and line-search options unchanged.

We minimize

$$\min_{\Gamma \in \Pi(p,q)} (1-\alpha)\,\langle M, \Gamma \rangle + \alpha \mathcal{Q}(\Gamma),$$

with the same CG loop, where this objective is linearized using the gradient in Equation (8).

When $d = 1$ (scalar edges), the method reduces to the original POT solver.

Table 8: Flow Matching hyperparameters.

| Category | Hyperparameter | Setting / Notes |
|---|---|---|
| Model | Architecture | Velocity MLP (hidden dim 64, depth 3, no dropout) |
| | Time conditioning | Sinusoidal embedding (dim 16), concatenated to inputs |
| Training schedule | Epochs | 500 |
| | Minibatch size | 128 (train loader), 2 048 for validation batches |
| | Optimizer | AdamW, lr $= 10^{-3}$, weight decay $10^{-4}$, betas $(0.9, 0.999)$ |

### D.5 NORMALIZATION

To balance the contributions of the feature term and the structure term in the objective described at Equation (5), we rescale the feature cost matrix $C$ and the CCI tensors $G^{(0)}$ and $G^{(1)}$. We first compute the two endpoint couplings by solving the feature-only ($\alpha = 0$) and structure-only ($\alpha = 1$) problems, yielding $\Gamma^\star_{\alpha=0}$ and $\Gamma^\star_{\alpha=1}$. We then define the scaling factors as follows:

$$\Delta \mathcal{F} := \mathcal{F}(\Gamma^\star_{\alpha=0}) - \mathcal{F}(\Gamma^\star_{\alpha=1}), \tag{9}$$

$$\Delta \mathcal{S} := \mathcal{S}(\Gamma^\star_{\alpha=1}) - \mathcal{S}(\Gamma^\star_{\alpha=0}), \tag{10}$$

$$\tag{11}$$

and rescale the feature cost matrix and the CCI tensors:

$$C \leftarrow \frac{C}{|\Delta \mathcal{F}|}, \tag{12}$$

$$G^{(0)} \leftarrow \frac{G^{(0)}}{\sqrt{\Delta \mathcal{S}}}, \qquad G^{(1)} \leftarrow \frac{G^{(1)}}{\sqrt{\Delta \mathcal{S}}}. \tag{13}$$

This places the terms on comparable scales so that $\alpha$ meaningfully reflects the feature/structure trade-off, and increasing $\alpha$ from 0 to 1 smoothly interpolates between the Kantorovich and the Gromov–Wasserstein problems.

### D.6 SELECTION OF LIGAND / RECEPTOR PAIRS

We apply LIANA's (Dimitrov et al., 2024b) consensus rank aggregation with `expr_prop` = 0.1 to obtain per–cell-type interaction scores. We retain interactions with `cellphone_pvals` $\leq 0.05$ and `lr_logfc` $\geq 0$, then keep ligand–receptor pairs whose `expr_prod` exceeds the median within that significant set. We require the same significance criteria in each snapshot. For every surviving pair, we aggregate LIANA results across significant edges to compute the mean expression product, average specificity ranks, counts of significant source→target edges, and the numbers of unique source and target cell types. We define coverage as `coverage` = `n_edges`$/N_{\text{sig edges}}$ and retain only pairs with $0.10 \leq$ `coverage` $\leq 0.40$ and at least two sources and two targets. We compute a standardized score $s = 0.6\, z(\text{mean\_expr}) + 0.4\, z(-\text{spec\_rank})$ and greedily select pairs in descending $s$ while preventing repeated ligands or receptors. We keep the top 10 pairs for each dataset.

### D.7 CONDITIONAL FLOW MATCHING HYPERPARAMETERS

We detail the hyperparameters used for the CFM stage in Table 8, which we kept fixed across the datasets. Given a $0.9/0.1$ train/val split, we keep the checkpoint that minimizes the validation loss over the run.

### D.8 BASELINES

**TrajectoryNet.** We use the implementation from the authors (Tong et al., 2020) available at https://github.com/KrishnaswamyLab/TrajectoryNet. We summarize the hyperparameters used in Table 9.

Table 9: TrajectoryNet hyperparameters.

| Category | Hyperparameter | Setting |
|---|---|---|
| Optimization | Training iterations | 1,000 |
| | Batch size | 1,000 |
| | Learning rate | $1 \times 10^{-3}$ |
| | Weight decay | $1 \times 10^{-5}$ |
| Model | Blocks / layer type | 1 block, concatsquash layers Hidden dimensions: 64-64-64 |
| | Activations | Softplus layers with tanh control |
| Regularization | $s_{\text{L2int}}$ | $1 \times 10^{-3}$ |
| | $k_{\text{top}}$ regularizer | $1 \times 10^{-2}$ |
| | Training noise | 0.1 |
| ODE solver | Time scale | 0.4 (five integration points) |
| | Solver | dopri5 |
| | Tolerances | rtol = atol = $1 \times 10^{-5}$ |

**Diffusion Schrodinger Bridges.** We use the implementation from the authors (De Bortoli et al., 2021) available at `https://github.com/JTT94/diffusion_schrodinger_bridge`. We summarize the hyperparameters used in Table 10.

**MIOFlow** We use the implementation from the authors (Huguet et al., 2022) available at `https://github.com/KrishnaswamyLab/MIOFlow`. We summarize the hyperparameters used in Table 11.

**Moscot** We use the implementation from the authors (Klein et al., 2025b) available at `https://github.com/theislab/moscot`. We summarize the hyperparameters used in Table 12.

**VGFM** We use the implementation from the authors (Wang et al., 2025) available at `https://github.com/DongyiWang-66/VGFM`. We summarize the hyperparameters used in Table 13.

**MFM** We use the implementation from the authors (Kapusniak et al., 2024) available at `https://github.com/kkapusniak/metric-flow-matching`. We summarize the hyperparameters used in Table 14.

**SF2M** We developed a custom implementation of the SF2M framework (Tong et al., 2023) to enable the integration of the `IADOT` structural prior into the simulation-free training objective. We summarize the hyperparameters used in Table 15.

**UOT-FM** To incorporate the interaction-aware coupling in an unbalanced setting, we utilized a custom implementation of UOT-FM based on the original formulation (Eyring et al., 2023). We summarize the hyperparameters used in Table 16.

Table 10: Diffusion Schrödinger Bridge (DSB) baseline hyperparameters.

| Category | Hyperparameter | Setting |
|---|---|---|
| Model | Score network | Encoder [16, 32] Decoder [64, 64, 64], dim =16 |
| Training schedule | IPF rounds | 10 outer IPF iterations |
| | Optimisation steps | 10 000 gradient updates |
| | Langevin steps | 12 steps per bridge trajectory |
| | Batch size | 128 |
| | Learning rate | $1 \times 10^{-4}$ |
| Regularisation | $\gamma$ schedule | $\gamma_{\min} = \gamma_{\max} = 10^{-3}$, linear spacing |
| | Mean matching | Enabled |
| | EMA | Disabled |

Table 11: MioFlow Baseline Hyperparameters.

| Category | Hyperparameter | Setting |
|---|---|---|
| Model Architecture | Network layers | $[64, 64, 64]$ |
| Optimization & Network Training | Learning rate (LR) | $1 \times 10^{-4}$ |
| | Total epochs ($N_{\text{epochs}}$) | 20 |
| | Local epochs per stage ($N_{\text{local}}$) | 5 |
| | Post-local epochs ($N_{\text{post\_local}}$) | 5 |
| | Sample size (Batch size) | 256 |
| | Number of batches per epoch ($N_{\text{batches}}$) | 100 |

Table 12: Moscot Baseline Hyperparameters.

| Category | Hyperparameter | Setting |
|---|---|---|
| Optimal Transport Parameters | Epsilon ($\epsilon$, regularization) | 0.001 |
| | $\tau_{\text{a}}$ (Source regularization) | 1.0 |
| | $\tau_{\text{b}}$ (Target regularization) | 1.0 |

Table 13: VGFM Baseline Hyperparameters.

| Category | Hyperparameter | Setting |
|---|---|---|
| Model Architecture | Hidden dimension (`hidden_dim`) | 64 |
| | Number of hidden layers (`n_hiddens`) | 3 |
| | Activation function | Tanh |
| Optimization | Pre-train epochs | 300 |
| | Training epochs | 50 |
| | Batch size | 256 |
| | Initial learning rate (`learning_rate1`) | $1 \times 10^{-3}$ |
| | Second learning rate (`learning_rate2`) | $1 \times 10^{-4}$ |
| | Stepsize (for solver) | 0.01 |

## D.9 LUNG CANCER DATA EXPERIMENT

For the experiment described in Section 5.4, We annotated the lung cancer dataset using canonical lineage and state markers (Table 17); an overview of the full dataset is shown in Fig. 7. Because whole-lung profiling dilutes treatment effects (the tumour comprises only a small fraction of total cells), we constructed a focused *tumour-niche* subset to increase sensitivity and interpretability. Concretely, we retained all tumour cells and subsampled an equal number of T cells, B cells, fibroblasts, and endothelial cells from the same specimens to form a minimal viable tumour microenvironment. We then reused the analysis pipeline described earlier with matched timepoints at 0 h, 8 h, and 24 h. The only modification was to the ligand–receptor (LR) library: for pathway-specific probes, we toggled custom LR pairs to mimic the presence or absence of a given ligand (e.g., EGFR) and quantified the resulting changes in inferred communication and downstream dynamics. Marker definitions are provided in Table 17, and a dot-plot confirming marker specificity and minimal cross-lineage leakage is shown in Fig. 8.

### D.9.1 TUMOUR PROGRESSION QUANTIFICATION USING HALLMARK GENE SETS

There is no single, universally accepted definition of tumour progression. Clinical assessments typically use lesion size, extent of metastasis, and histopathology. While we observe distinct cellular changes and invasion over our 24 h window, these measures are not applicable at single-cell resolution. Instead, we construct an approximate *tumour differentiation* score based on the Hallmarks of Cancer (Hanahan & Weinberg, 2011), using the MSigDB *Hallmark* gene sets (Liberzon et al., 2015).

Table 14: MFM hyperparameters.

| Category | Hyperparameter | Setting / Notes |
|---|---|---|
| **Velocity network (Flow Matching)** | | |
| Model | Architecture | Velocity MLP (hidden dim 64, depth 3, no dropout) |
| | Time conditioning | Sinusoidal embedding (dim 16), concatenated to inputs |
| Training | Epochs | 500 |
| | Batch size | 128 (train), 2 048 (validation) |
| | Optimizer | AdamW, lr $= 10^{-3}$, wd $= 10^{-4}$, $\beta = (0.9, 0.999)$ |
| | Gradient clipping | 1.0 |
| **GeoPath network (Riemannian Correction)** | | |
| Model | Architecture | GeoPath MLP (hidden dim 128, depth 3) |
| | Time conditioning | Concatenated directly (no embedding) |
| | Activation | SELU |
| Training | Epochs | 100 |
| | Batch size | 128 (train) |
| | Optimizer | Adam, lr $= 10^{-4}$, wd $= 0.0$, $\beta = (0.9, 0.999)$ |
| | Time sampler | Uniform |
| Metric | Type | LAND |
| | LAND parameters | $\gamma = 0.2$, $\rho = 10^{-3}$, $\alpha = 1.0$ |
| | Max samples | 4 096 for metric computation |

Table 15: SF2M hyperparameters.

| Category | Hyperparameter | Setting / Notes |
|---|---|---|
| Model | Architecture | Velocity and Score MLP (hidden dim 64, depth 3) |
| | Time conditioning | Sinusoidal embedding (dim 16), concat. to inputs |
| | Activation | SELU |
| Distribution parameters | $\sigma_{\text{bridge}}$ | 1.0 |
| | $\sigma_{\text{sample}}$ | 1.0 |
| Training schedule | Epochs | 500 |
| | Minibatch size | 128 (train loader), 2 048 for validation batches |
| | Optimizer | AdamW, |
| | lr | $10^{-3}$ |
| | weight decay | $10^{-4}$ |
| | betas | $(0.9, 0.999)$ |
| | Gradient clipping | 1.0 |

For each hallmark, we compute a per-cell score as the *median* expression across its member genes (chosen over the mean for robustness to sparsity and outliers). The overall progression score is then the mean across the 20 retained hallmarks. The full hallmark definitions are available in MSigDB (Liberzon et al., 2015); the selected hallmarks, their gene counts, and five example genes each are listed in Table 18. Hallmarks not applicable to our tumour context (e.g., hormonal signalling for breast/prostate, long-term metabolic programs) were excluded.

As a baseline check, we verify that tumour cells exhibit coherent changes along the selected hallmarks over $0\,\text{h} \to 24\,\text{h}$; see Fig. 9.

Table 16: UOT-FM hyperparameters.

| Category | Hyperparameter | Setting / Notes |
|---|---|---|
| Model | Architecture | Velocity MLP (hidden dim 64, depth 3, no dropout) |
| | Time conditioning | Sinusoidal embedding (dim 16), concatenated to inputs |
| Training schedule | Epochs | 500 |
| | Minibatch size | 128 (train loader), 2 048 for validation batches |
| | Optimizer | AdamW, $\mathrm{lr} = 10^{-3}$, weight decay $10^{-4}$, betas $(0.9, 0.999)$ |
| Optimal transport | Convergence tol. | $10^{-9}$ (relative and absolute) |
| | Marginal reg. | 1.0 |

| Cell Type | Positive Markers |
|---|---|
| Differentiated AT1 | RTKN2, AGER |
| AT1 | CLDN18 |
| Tumour (AT2) | SFTPD, LAMP3, SCGB3A2 |
| Mucous Epithelial | DNAH12, AZGP1 |
| Endothelial | SEMA3G |
| Low IEG Endothelial | CDH5 |
| Alveolar Capillary Endothelial | EDNRB, RPRML |
| Lymphatic Vein Endothelial | LYVE1, SELE, VWF |
| Fibroblasts | COL1A2, PDGFRA |
| Smooth Muscle Fibroblasts | ACTA2, LGR6 |
| Fibroblast Subset | DCN |
| Pericytes | CSPG4 |
| Megakaryocytes | PPBP, PF4 |
| Erythrocytes | ALAS2 |
| Lymphocytes | CCL21A |
| Cycling | TOP2A |
| Neutrophils | S100A9, RETNLG |
| Basophils & Mast cells | MCPT8, MS4A2 |
| Macrophages | MARCO |
| Monocytes | LY6I |
| DC 1 and 2 | CLEC9A, XCR1, C1QA, SIGLECH |
| DC 3 | FSCN1, IL12B |
| NK cell like | NCR1, EOMES, TBX21 |
| ILC | RORA, RORC, IL2RA |
| Adaptive T cells | FOXP3, CD4, CD8A |
| B cells | CD79A |

Table 17: Curated panel of positive marker genes used for per-cell scoring and assignment in the lung cancer dataset.

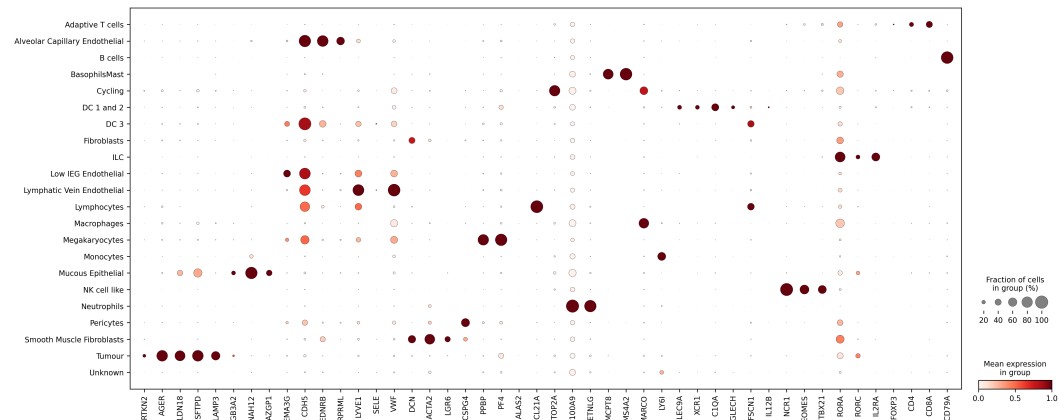

Figure 8: Dot-plot validation of curated marker genes across annotated cell types. Each column corresponds to a marker gene and each row to a cell-type labe. Dot size encodes the fraction of cells expressing that gene, while color intensity represents its standardized expression level.

Table 18: **Hallmark gene sets used for trajectory summarization.** We list each set's size and five randomly sampled member genes.

| Gene set | # Genes | Random gene examples (5) |
|---|---|---|
| Angiogenesis | 36 | TIMP1, POSTN, VTN, THBD, NRP1 |
| Apoptosis | 161 | ERBB2, IL1B, DPYD, NEDD9, MADD |
| DNA Repair | 150 | GTF2B, RAE1, ADCY6, POLA2, TAF1C |
| E2F Targets | 200 | MCM7, PCNA, MCM4, RFC2, GINS1 |
| Epithelial–Mesenchymal Transition | 200 | SPP1, GPX7, LOX, THBS1, SLC6A8 |
| G2M Checkpoint | 200 | RBM14, AMD1, CDC27, UCK2, NDC80 |
| Glycolysis | 200 | SPAG4, PKP2, SLC25A13, PRPS1, ZNF292 |
| Hypoxia | 200 | S100A4, CSRP2, DTNA, PIM1, TPST2 |
| KRAS Signaling v1 | 200 | FSHB, YPEL1, BARD1, SLC6A3, ATP6V1B1 |
| KRAS Signaling v2 | 200 | CIDEA, KIF5C, LAT2, PDCD1LG2, PIGR |
| MYC Targets v1 | 200 | RAD23B, USP1, NAP1L1, NDUFAB1, SNRPA1 |
| MYC Targets v2 | 58 | PRMT3, AIMP2, SRM, EXOSC5, SUPV3L1 |
| Myogenesis | 200 | EIF4A2, PDE4DIP, ANKRD2, EPHB3, ATP6AP1 |
| Notch Signaling | 32 | SKP1, MAML2, HES1, FBXW11, DTX1 |
| Oxidative Phosphorylation | 200 | NDUFS8, VDAC1, UQCRQ, NDUFB3, NDUFB2 |
| p53 Pathway | 200 | TNNI1, SLC35D1, BTG1, FDXR, JAG2 |
| Peroxisome | 104 | IDH2, FIS1, EPHX2, SLC23A2, SLC25A4 |
| Reactive Oxygen Species Pathway | 49 | PRNP, OXSR1, SOD1, PDLIM1, TXN |
| TNF$\alpha$ Signaling via NF$\kappa$B | 200 | DUSP2, CEBPB, OLR1, CCL20, IL1A |
| Xenobiotic Metabolism | 200 | SSR3, HACL1, ARPP19, AHCY, GSR |

## D.10 COMPUTATIONAL AND MEMORY COSTS

**Complexity of the full OT solver.** We solve Equation (5) with a custom conditional-gradient (Frank–Wolfe) solver detailed in Section D.4. Let $n_0$ and $n_1$ be the numbers of cells in the two snapshots and $K$ the number of ligand–receptor (LR) pairs (interaction channels).

Each Frank–Wolfe iteration consists of two main steps:

1. **Gradient computation.** This yields a per-iteration cost

$$\mathcal{O}\big(K\, n_0 n_1 (n_0 + n_1)\big).$$

since it requires performing the matrix multiplication of $C_1^{(r)}\Gamma$ and $\big(C_1^{(r)}\Gamma\big)\big(C_2^{(r)}\big)^\top$ for each channel $r$.

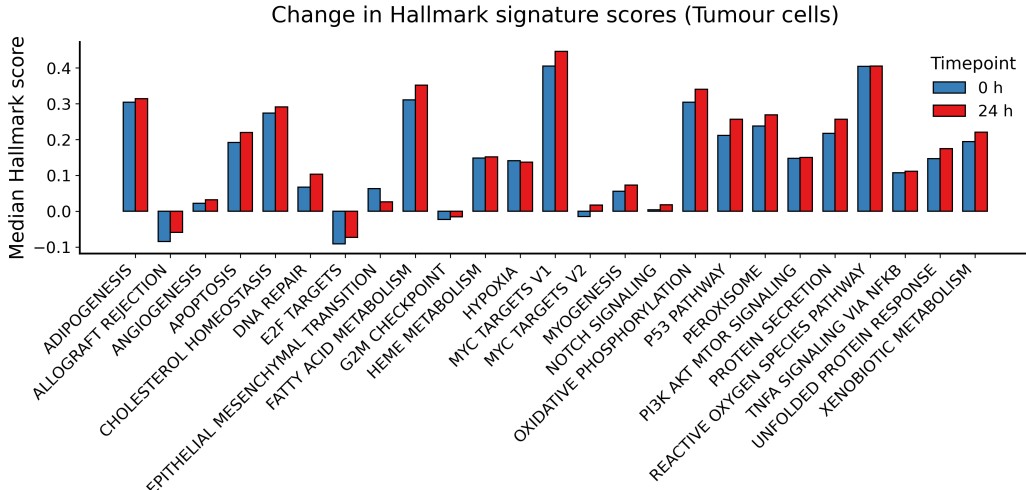

Figure 9: **Hallmark changes.** Changes in our dataset over 24 h following combined KRAS and MYC signalling across the 20 selected Hallmark gene sets.

Table 19: Wall-clock runtime (in seconds) of `IADOT`, decomposing into the OT part and the Flow matching part.

|        | Lung Tumour | V1 Light | Dendritic Stimulus |
| ------ | ----------- | -------- | ------------------ |
| OT [s] | 211.3       | 107.0    | 4.3                |
| FM [s] | 189.1       | 186.0    | 24.2               |

2. **Linear OT subproblem.** Given the linearized objective, we solve a linear OT problem over $\Pi(a, b)$ using POT's (Flamary et al., 2024) existing OT routine. Its complexity is

$$\mathcal{O}\big(\text{cost}_{\text{OT}}(n_0, n_1)\big),$$

(e.g. cubic in $n$ for a network-simplex LP, or $\mathcal{O}(T_{\text{Sinkhorn}} \, n_0 n_1)$ for entropic OT).

If $T_{\text{CG}}$ denotes the number of Frank–Wolfe iterations required to reach the desired tolerance, the total complexity of the IADOT OT stage is

$$\mathcal{O}\Big(T_{\text{CG}}\big[K \, n_0 n_1 (n_0 + n_1) + \text{cost}_{\text{OT}}(n_0, n_1)\big]\Big).$$

**Wall-clock runtimes.** We report the wall-clock runtimes (seconds) in Table 19, decomposing it into the OT part (finding the coupling $\Gamma^*$) and the Flow matching part (fitting the velocity model).

**Memory footprint of the OT stage.** The dominant memory costs come from: (i) the coupling $\Gamma \in \mathbb{R}^{n_0 \times n_1}$, (ii) the feature cost matrix $C \in \mathbb{R}^{n_0 \times n_1}$, (iii) the multi-channel structure tensors $C_1 \in \mathbb{R}^{n_0 \times n_0 \times K}$ and $C_2 \in \mathbb{R}^{n_1 \times n_1 \times K}$ (corresponding to the CCI tensors $G^{(0)}$ and $G^{(1)}$), and (iv) a small number of auxiliary matrices of size $n_0 \times n_1$ (e.g. constC, $B(\Gamma)$, and the gradient). Crucially, we never construct the full tensor of pairwise structure discrepancies in Equation (5). Instead, the structure term is implemented through the matrix products $C_1^{(r)} \Gamma \big(C_2^{(r)}\big)^\top$. As a result, the memory complexity of the OT solver scales as

$$\mathcal{O}\big(K(n_0^2 + n_1^2) + n_0 n_1\big),$$

Hence it is quadratic in the number of cells per snapshot and linear in the number of LR pairs $K$. In comparison, a feature-only OT solver ($\alpha = 0$) needs $C$ and $\Gamma$, with memory $\mathcal{O}(n_0 n_1)$.

**Using `IADOT` with large-scale datasets.** While the computational and memory cost remained reasonable across the datasets we used, for very large datasets it can be mitigated using standard scalable techniques that are orthogonal to our formulation:

- adding entropic regularization (Peyré & Cuturi, 2019) and using Sinkhorn-type solvers, which make the problem easier to optimize and reduce memory at the price of a small, controllable bias

- employing mini-batch optimization (Fatras et al., 2021), where the CCI prior is estimated from couplings computed on minibatches instead of the whole dataset

- constructing metacells, with details provided in Section D.3. Using metacells reduces the effective sample size. We evaluate this variant in Section 5.5.

These strategies preserve the form of the `IADOT` prior while substantially improving scalability for large scale datasets.

### D.11 EXTENDING `IADOT` TO THE UNBALANCED SETTING

As mentioned in Section 4.1, we can take into account cell proliferation or apoptosis between snapshots with an *unbalanced* OT formulation. Instead of searching the coupling over the space $\Pi(a, b)$ of couplings with marginals $a$ and $b$, we instead optimize an objective that relaxes this hard constraint. More specifically, we adopt a two-step procedure:

**Step 1: inferring non-uniform marginals via unbalanced OT.** We first ignore CCI structure and solve an unbalanced feature-only problem

$$\Gamma^{\mathrm{u}} \in \underset{\Gamma \in \mathbb{R}_{\geq 0}^{n_0 \times n_1}}{\arg\min} \left\{ \langle \Gamma, C \rangle + \lambda_0 \, \mathrm{KL}(\Gamma \mathbf{1} \, \| \, a) + \lambda_1 \, \mathrm{KL}(\Gamma^\top \mathbf{1} \, \| \, b) \right\}, \tag{14}$$

where $\mathrm{KL}(\cdot \| \cdot)$ denotes the Kullback–Leibler divergence and $\mathbf{1}$ is the all-ones vector. The penalty terms are soft constraints on the marginals of $\Gamma$, allowing deviations from $(a, b)$ that capture net cell birth or death between snapshots. From the optimal coupling $\Gamma^{\mathrm{u}}$ we extract the reweighted marginals

$$\tilde{a} = \Gamma^{\mathrm{u}} \mathbf{1}, \qquad \tilde{b} = (\Gamma^{\mathrm{u}})^\top \mathbf{1},$$

which are renormalized to sum to one. We note that this step is $\alpha$- agnostic, which allows to keep fixed marginals across the different $\alpha$. In practice, we use a solver based on L-BFGS-B implemented in (Flamary et al., 2024).

**Step 2: FGW with frozen unbalanced marginals.** In a second step, we fix $\tilde{a}$ and $\tilde{b}$ and solve the interaction-aware FGW problem of Section 4.1 with these new marginals:

$$\min_{\Gamma \in \Pi(\tilde{a}, \tilde{b})} (1 - \alpha) \, \mathcal{F}(\Gamma) + \alpha \, \mathcal{S}(\Gamma), \tag{15}$$

where $\mathcal{F}$ and $\mathcal{S}$ are defined as in Equation (5). This second step preserves the multi-LR-pair CCI structure while respecting the unequal total mass at the two snapshots inferred in Step 1.

### D.12 COMBINING `IADOT` WITH MFM

Here we detail how our CCI-based prior can be combined with Metric Flow Matching (MFM) (Kapusniak et al., 2024). MFM generalizes Conditional Flow Matching by learning interpolants $x_{t,\eta}$ that approximate geodesics of a data–dependent Riemannian metric $g$ on the ambient space. Given a coupling $q$ between $p_0$ and $p_1$, MFM first trains a network $\phi_{t,\eta}$ to minimize the geodesic loss

$$L_g(\eta) = \mathbb{E}_{(x_0, x_1) \sim q, \, t} \big[ \dot{x}_{t,\eta}^\top G(x_{t,\eta}; \mathcal{D}) \, \dot{x}_{t,\eta} \big],$$

where $G(\cdot; \mathcal{D})$ is the coordinate representation of the metric and $x_{t,\eta} = (1 - t)x_0 + tx_1 + t(1 - t)\phi_{t,\eta}(x_0, x_1)$.

In our experiments in Section 5.3 and Section E.4, we instantiate MFM with the LAND metric $g_{\mathrm{LAND}}$ (Arvanitidis et al., 2016).

Once the interpolant parameters $\eta^\star$ have been fitted via $L_g$, we can optimize the following MFM velocity–regression loss using the coupling found with `IADOT`:

$$\mathcal{L}_{\mathrm{MFM}}(\theta, \eta) = \mathbb{E}_{\substack{(X,Y)\sim\Pi \\ t\sim\mathrm{Unif}[0,1]}} \left[ \left\| v_\theta(Z_{t,\eta}, t) \;-\; u_{t,\eta}(Z_{t,\eta} \mid X, Y) \right\|^2_{g(Z_{t,\eta})} \right] \tag{16}$$

$$= \mathbb{E}_{\substack{(X,Y)\sim\Pi \\ t\sim\mathrm{Unif}[0,1]}} \left[ \left\| v_\theta(Z_{t,\eta}, t) \;-\; \dot{x}_{t,\eta}(X, Y) \right\|^2_{g(Z_{t,\eta})} \right], \tag{17}$$

where $\| \cdot \|_{g(Z_{t,\eta})}$ is the norm induced by the Riemannian metric $g$ at $Z_{t,\eta}$, i.e.

$$\|w\|^2_{g(Z_{t,\eta})} = w^\top G(Z_{t,\eta}; \mathcal{D}) \, w.$$

### D.13 COMBINING IADOT WITH SF2M

Here we describe how our CCI-based prior can be combined with SF2M (Tong et al., 2023) to obtain a Schrödinger-bridge–type dynamics that uses the IADOT coupling.

SF2M learns a drift $v_\theta$ and score $s_\theta$ by regressing to the conditional flow and score of a mixture of Brownian bridges between endpoints $(x_0, x_1)$. For a single bridge with diffusion $\sigma$, the conditional marginal at time $t \in (0, 1)$ is

$$p_t(x \mid x_0, x_1) = \mathcal{N}\big(x; \mu_t(x_0, x_1), \sigma^2 t(1-t) I_d\big), \qquad \mu_t(x_0, x_1) = (1-t)x_0 + t x_1,$$

with closed-form drift $u_t^\circ(x \mid x_0, x_1)$ and score $\nabla_x \log p_t(x \mid x_0, x_1)$ given in (Tong et al., 2024).

To combine SF2M with IADOT, we simply instantiate the endpoint coupling with the IADOT coupling $\Pi$ instead of the entropic OT plan. Training triples are then sampled as

$$t \sim \mathrm{Unif}[0,1], \qquad (X, Y) \sim \Pi, \qquad Z_t \sim p_t(\cdot \mid X, Y),$$

where $p_t(\cdot \mid X, Y)$ is the Brownian-bridge marginal above with $\mu_t(X, Y) = (1-t)X + tY$. The SF2M objective specialized to IADOT is

$$\mathcal{L}_{\mathrm{SF^2M}}(\theta) = \mathbb{E}_{\substack{(X,Y)\sim\Pi \\ t\sim\mathrm{Unif}[0,1] \\ Z_t\sim p_t(\cdot|X,Y)}} \left[ \left\| v_\theta(t, Z_t) \;-\; u_t^\circ(Z_t \mid X, Y) \right\|^2_2 \right. \tag{18}$$

$$\left. + \lambda(t)^2 \left\| s_\theta(t, Z_t) \;-\; \nabla_z \log p_t(Z_t \mid X, Y) \right\|^2_2 \right], \tag{19}$$

where $\lambda(t) = 2\sqrt{t(1-t)}/\sigma$ is the time-dependent weighting.

## E Additional results

### E.1 STIMULUS DATASETS

We reproduce the experimental setup described in Section 5.2 and Section 5.3 with the macrophase stimulus-response dataset (Section C.6). We report the results in Figure 10 and Table 20, which are consistent with the findings on the other datasets.

### E.2 SENSITIVITY OF COUPLINGS TO CATALOG EDITS

The experiment presented in Section 5.4 involved perturbing the LR catalog by removing specific LR pairs. In Table 21, we show how the coupling changes, by computing the fraction of source cells whose target argmax differs between "active" vs. "inactive" LR libraries for each pathway.

### E.3 COMPARING CELL INTERACTION TYPES

We further examined how different classes of molecular interactions influence the resulting transport couplings. Using our automated selection procedure (Section D.6), we identified a top-ranking set of 10 ligand-receptor pairs for each of the datasets. We contrasted this against a matched set of 10 canonical long-range soluble cytokines and growth factors: (CXCL12-CXCR4, VEGFA-KDR,

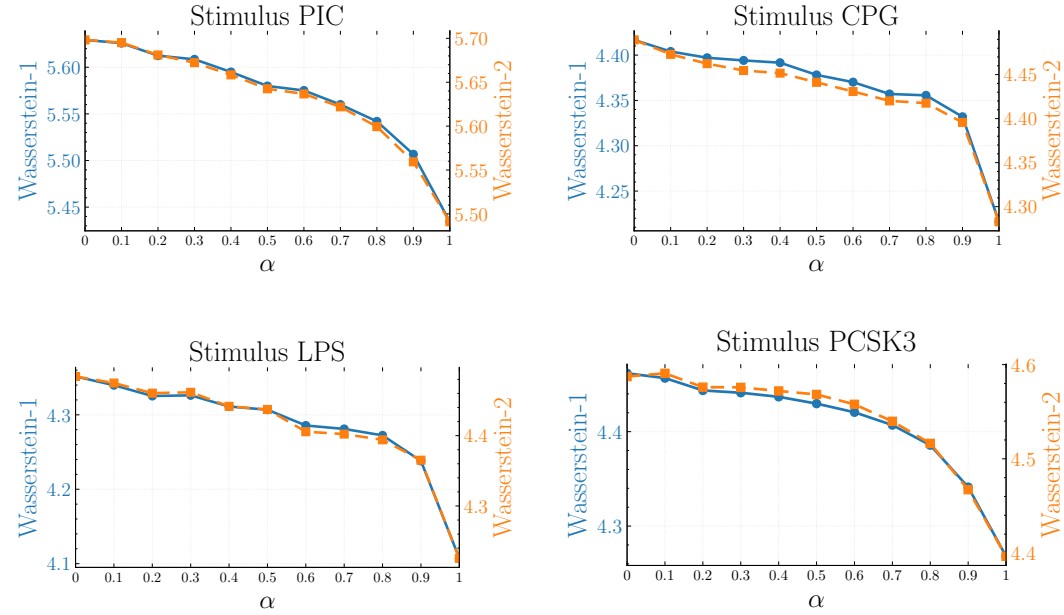

Figure 10: **Interpolation** results for the macrophage stimulus datasets.

Table 20: **Interpolation error for continuous time dynamics (lower is better).** IADOT with varying structure weight $\alpha$ vs. baselines for the macrophase stimulus datasets. We report mean±std over 5 runs.

| Method | $\alpha$ | Stimulus PIC | | Stimulus CPG | | Stimulus LPS | | Stimulus PCSK3 | |
|---|---|---|---|---|---|---|---|---|---|
| | | $W_1$ | $W_2$ | $W_1$ | $W_2$ | $W_1$ | $W_2$ | $W_1$ | $W_2$ |
| TrajectoryNet | — | 5.628 (0.055) | 5.930 (0.049) | 5.361 (0.085) | 5.826 (0.080) | 5.087 (0.109) | 5.589 (0.078) | 5.033 (0.051) | 5.434 (0.051) |
| DSB | — | 5.796 (0.574) | 5.833 (0.571) | 4.500 (0.128) | 4.594 (0.114) | 4.685 (0.533) | 4.815 (0.528) | 4.648 (0.328) | 4.749 (0.324) |
| OT-CFM | — | 5.544 (0.038) | 5.614 (0.036) | 4.430 (0.019) | 4.512 (0.020) | 4.434 (0.052) | 4.582 (0.058) | 4.423 (0.020) | 4.551 (0.018) |
| OT-MFM | — | 5.501 (0.020) | 5.566 (0.022) | 4.464 (0.013) | 4.556 (0.015) | 4.440 (0.033) | 4.592 (0.038) | 4.384 (0.008) | 4.505 (0.008) |
| UOT-FM | — | 5.414 (0.027) | 5.492 (0.024) | 4.572 (0.033) | 4.733 (0.045) | 4.570 (0.122) | 4.785 (0.143) | 4.332 (0.007) | **4.483 (0.008)** |
| SF2M | — | 6.132 (0.059) | 6.212 (0.058) | 4.959 (0.044) | 5.053 (0.043) | 4.930 (0.051) | 5.057 (0.054) | 5.048 (0.031) | 5.155 (0.032) |
| VGFM | — | 9.796 (0.658) | 9.863 (0.683) | 8.242 (0.192) | 8.362 (0.205) | 8.026 (0.135) | 8.158 (0.1469) | 8.205 (0.1101) | 8.297 (0.110) |
| MIOFlow | — | 9.365 (0.382) | 9.440 (0.404) | 16899 (22805) | 22007 (31264) | 7.807 (0.038) | 7.927 (0.034) | 8.179 (0.150) | 8.281 (0.166) |
| Moscot | — | 7.213 (0.000) | 7.244 (0.0000) | 6.983 (0.000) | 7.087 (0.000) | 7.663 (0.000) | 7.786 (0.000) | 6.734 (0.000) | 6.832 (0.000) |
| IADOT +SF2M | 0.5 | 6.109 (0.081) | 6.193 (0.086) | 4.948 (0.066) | 5.027 (0.063) | 4.846 (0.046) | 4.973 (0.046) | 4.960 (0.058) | 5.084 (0.058) |
| | 1 | 6.110 (0.062) | 6.197 (0.063) | 4.951 (0.056) | 5.037 (0.058) | 4.874 (0.068) | 5.016 (0.065) | 5.016 (0.029) | 5.173 (0.039) |
| IADOT +MFM | 0.5 | 5.483 (0.012) | 5.544 (0.014) | 4.448 (0.007) | 4.527 (0.006) | 4.442 (0.035) | 4.589 (0.041) | 4.386 (0.016) | 4.520 (0.013) |
| | 1 | 5.376 (0.021) | **5.440 (0.022)** | 4.460 (0.079) | 4.543 (0.082) | 4.477 (0.050) | 4.641 (0.062) | 4.329 (0.023) | 4.507 (0.029) |
| IADOT +UOT-FM | 0.5 | 5.355 (0.021) | 5.451 (0.018) | 4.544 (0.033) | 4.700 (0.035) | 4.489 (0.054) | 4.692 (0.061) | 4.343 (0.025) | 4.522 (0.033) |
| | 1 | **5.346 (0.024)** | 5.487 (0.038) | 4.547 (0.035) | 4.755 (0.040) | 4.489 (0.038) | 4.716 (0.041) | **4.325 (0.038)** | 4.531 (0.038) |
| IADOT +CFM | 0.5 | 5.490 (0.018) | 5.555 (0.019) | **4.427 (0.021)** | **4.502 (0.025)** | **4.380 (0.021)** | **4.517 (0.021)** | 4.396 (0.023) | 4.531 (0.019) |
| | 1 | 5.446 (0.018) | 5.512 (0.016) | 4.440 (0.041) | 4.518 (0.044) | 4.431 (0.126) | 4.577 (0.141) | 4.352 (0.020) | 4.530 (0.033) |

Table 21: **Coupling changes (argmax) at $\alpha = 1.0$.** Fraction of source cells whose target argmax differs between "active" vs. "inactive" LR libraries for each pathway; $N=2195$ source cells. Targeted pathways (EGFR/ALK/MET) show large shifts, while cardio–renal controls (RAAS, Vasopressin, Natriuretic) show little or moderate effect, as expected.

| Pathway / System | Coupling changed (count / $N$) | Percent |
|---|---|---|
| EGFR (targeted) | 2071/2195 | 94.35% |
| ALK (targeted) | 2164/2195 | 98.59% |
| MET (targeted) | 2154/2195 | 98.13% |
| RAAS (control) | 0/2195 | 0.00% |
| Vasopressin (control) | 0/2195 | 0.00% |
| Natriuretic (control) | 1582/2195 | 72.07% |

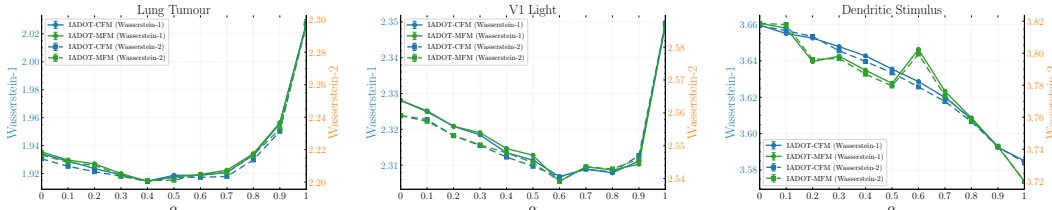

Figure 11: **Incorporating the CCI prior with MFM.** We plot the $W_1$ and $W_2$ distances between the interpolated and empirical $t_1$ snapshots as $\alpha$ varies.

CCL5-CCR5, TGFB1-TGFBR2, IL6-IL6R, EGF-EGFR, TNF-TNFRSF1A, IGF1-IGF1R, CSF1-CSF1R, IFNG-IFNGR1). As shown in Table 22, the cytokine pairs exhibit slightly higher Wasserstein ($W_1$ and $W_2$) distances compared to the results obtained previously with our selection procedure. This suggests that the specific interaction modes we keep have more informative topological constraints on the transport map than generic diffusive signaling, effectively recovering structure-aware couplings that reflect the physical tissue architecture.

Table 22: **Wasserstein distances for pure structural alignment ($\alpha = 1$).** Comparison of interpolation performance using generic Long Range priors versus our automated selection procedure. Lower values indicate better alignment.

| Dataset | Interaction Prior | $\mathcal{W}_1 \downarrow$ | $\mathcal{W}_2 \downarrow$ |
|---|---|---|---|
| **V1 Light** | Long range | 2.42 | 2.63 |
| | Dataset-specific | **2.35** | **2.59** |
| **Immune** | Long range | **3.58** | **3.73** |
| | Dataset-specific | 3.59 | **3.73** |
| **Lung Cancer** | Long range | 2.10 | 2.33 |
| | Dataset-specific | **2.02** | **2.30** |

### E.4    COMBINING IADOT WITH OTHER PRIORS

A key advantage of IADOT is its modularity: the CCI-derived prior only depends on the CCI tensors $(G^{(0)}, G^{(1)})$ and on a coupling $\Gamma$, and is therefore largely orthogonal to how $\Gamma$ is obtained. As a consequence, the CCI prior can be combined with a wide range of existing priors or architectural choices for trajectory inference. Here, we illustrate this flexibility by extending IADOT to two settings: (i) unbalanced OT, which explicitly accounts for cell birth and death between snapshots, and (ii) metric flow matching, which replaces the standard Euclidean flow-matching objective with a geometry-aware variant. Details on both of these implementations can be found in Section D.11 and Section D.12.

We reproduce the experiment in Section 5.2 with these IADOT variants, and report the results in Figure 11 and Figure 12. We notice the following:

- $\alpha > 0$ **remains optimal.** For all datasets and the two IADOT variants, the best $W_1/W_2$ values occur at a non-zero structure weight $\alpha$, mirroring the behavior observed in Section 5.2.
- **Complementary to other priors.** The fact that $\alpha > 0$ remains optimal shows that adding the CCI prior on top of MFM or UOT-FM yields consistent improvements over the corresponding feature-only baselines, highlighting that IADOT's gains are not tied to a specific OT or flow-matching objective, but rather come from the biological prior.

### E.5    SENSITIVITY ANALYSIS ON THE LR EXPRESSIONS

In this section, we study the sensitivity of IADOT to measurement noise in the LR expressions. We inject this noise in LR genes expressions by adding zero-mean Gaussian noise to the gene

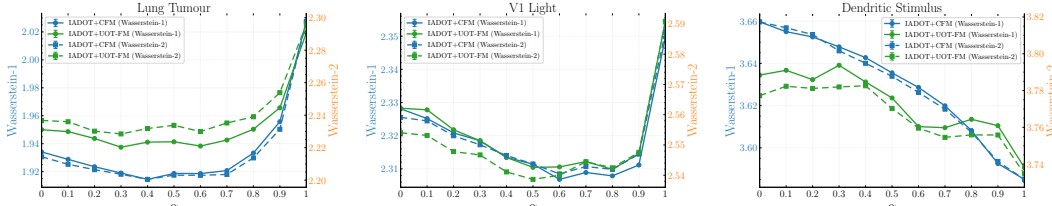

Figure 12: **Incorporating the CCI prior with UOT-CFM.** We plot the $W_1$ and $W_2$ distances between the interpolated and empirical $t_1$ snapshots as $\alpha$ varies.

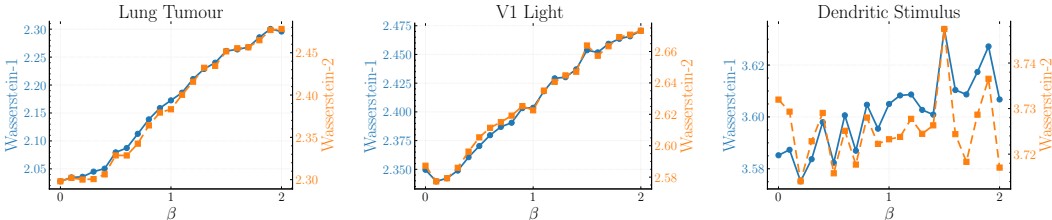

Figure 13: **Perturbation of the LR expressions with Gaussian noise.** We plot the $W_1$ and $W_2$ distances between the interpolated and empirical $t_1$ snapshots as the scaling factor of the noise $\beta$ increases.

expressions before applying the Hill transform and clipping below by 0, i.e. we define $\tilde{x}_{cg} = \max(0, x_{cg} + \epsilon_{cg})$ where $\epsilon_{cg} \sim \mathcal{N}(0, \sigma_g^2)$. $\sigma_g^2$ denotes a gene-specific noise variance, defined as $\sigma_g = \beta \hat{\sigma}_g$, with $\hat{\sigma}_g$ the empirical standard deviation of $\{x_{cg} \mid c \in [n]\}$ (to take into account per-gene variance) and $\beta$ a scaling factor. From these perturbed expressions, we compute the activations $\tilde{s}_{cg}$ and we construct the CCI tensors with the entries $\tilde{q}_{i \to j}^{(p_k)} = \tilde{s}_{i\ell_k} \tilde{s}_{jr_k}$ and obtain the couplings by solving Equation (5).

We report the results in Figure 13, where we sweep $\beta$ for different values across the interval $[0, 2]$, with $\alpha = 1$.

As the noise scale $\beta$ increases, both $W_1$ and $W_2$ gradually deteriorate across all three datasets. This non-zero sensitivity is expected and desirable: if the CCI prior was irrelevant, corrupting the LR expressions would leave the interpolation error unchanged. Instead, adding noise worsens alignment, showing the benefits of the prior. Furthermore, the performance is relatively robust to small level of noises for the Lung tumour and V1 Light datasets. Interestingly, for the V1 Light dataset, we see that $\beta \in \{0.1, 0.2\}$ improves the results upon $\beta = 0$, which we attribute to a small regularization / denoising effect. Adding a small amount of centered Gaussian noise before the Hill transform and clipping makes low-intensity ligand or receptor expressions become zero while leaving strongly expressed pairs essentially unchanged. The results are noisier for the Dendritic Stimulus dataset, which we attribute to the smaller size of the dataset.

### E.6 SENSITIVITY WITH RESPECT TO $K_g$ AND $h_g$

In this section, we conduct a sensitivity analysis on the hyperparameters $K_g$ and $h_g$, used to define the interaction scores in Section 4.1 as $s_{cg} = x_{cg}^{h_g} / (x_{cg}^{h_g} + K_g^{h_g})$. We consider different values of the percentile level $p \in \{80, 90, 99\}$ (with $K_g = Q_g(p)$ denoting the $p$-th percentile of $\{x_{cg} \mid c \in [n]\}$) and $h_g \in \{1, 2, 4\}$, for $\alpha = 0.5$. We report the interpolation results in Figure 14. We observe that the performance is largely insensitive to the specific choice of these parameters. This stability justifies the use of standard default values (90th percentile and $h_g = 1$) across our experiments without the need for extensive per-dataset tuning.

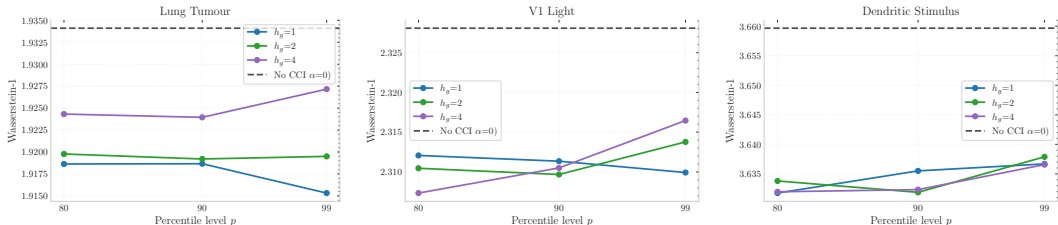

Figure 14: **Sensitivity analysis on the hyperparameters of the Hill transform.** We plot the $W_1$ distances between the interpolated and empirical $t_1$ snapshots for different values of $K_g$ and $h_g$.

### E.7 PATH CURVATURE ANALYSIS

To empirically demonstrate that `IADOT` learns non-linear interaction effects, we measure the average path length ratio (displacement divided by path length) of the inferred trajectories:

$$S(z_0, v_\theta) = \frac{\|z_1 - z_0\|_2}{\int_0^1 \|v_\theta(z_t, t)\|_2 \, dt} \tag{20}$$

where $z_t = z_0 + \int_0^t v_\theta(z_t, t)dt$ for $t \in [0, 1]$ and $z_0$ denotes the initial point.

A ratio of $1.0$ indicates a straight line, while values $< 1.0$ indicate curvature.

As shown in Table 23, increasing the interaction weight $\alpha$ leads to significantly higher curvature (lower ratios), confirming that incorporating interactions prevents the model from simply learning independent straight lines.

Table 23: Path length ratio comparison across different datasets.

| $\alpha$ | Lung Tumour | Dendritic Stimulus | V1 Light |
|---|---|---|---|
| 0 | $0.974 \pm 0.001$ | $0.982 \pm 0.002$ | $0.954 \pm 0.004$ |
| 0.5 | $0.952 \pm 0.002$ | $0.979 \pm 0.003$ | $0.907 \pm 0.011$ |
| 1 | $0.862 \pm 0.011$ | $0.969 \pm 0.005$ | $0.635 \pm 0.009$ |

# F  Theoretical analysis

## F.1  SYNTHETIC SETUP

In this section, we provide a theoretical guarantee for the synthetic setup of Section C.1.

**Theorem 1.** *Let $\mathcal{D}_0$ and $\mathcal{D}_1$ be the source and target datasets defined by the synthetic clusters, and let $G^{(0)}, G^{(1)}$ be the associated directed interaction tensors.*

*Consider the two candidate couplings:*

  1. *$\Gamma_{GT}$: The transport plan corresponding to the true translation vectors (preserving cluster identity).*

  2. *$\Gamma_{FO}$: The transport plan corresponding to the feature-only map.*

*As in Section D.5, define the (unnormalized) feature and structure gaps between these two couplings as*

$$\Delta\mathcal{F} := \mathcal{F}(\Gamma_{FO}) - \mathcal{F}(\Gamma_{GT}), \qquad \Delta\mathcal{S} := \mathcal{S}(\Gamma_{FO}) - \mathcal{S}(\Gamma_{GT}),$$

*and the corresponding normalized feature and structure terms*

$$\tilde{\mathcal{F}}(\Gamma) := \frac{\mathcal{F}(\Gamma)}{|\Delta\mathcal{F}|}, \qquad \tilde{\mathcal{S}}(\Gamma) := \frac{\mathcal{S}(\Gamma)}{\Delta\mathcal{S}}.$$

*Let*

$$J(\Gamma, \alpha) = (1 - \alpha)\,\tilde{\mathcal{F}}(\Gamma) + \alpha\,\tilde{\mathcal{S}}(\Gamma)$$

*be the normalized FGW objective function.*

*Let $N$ denote the size of each cluster. Then, for sufficiently large $N$, there exists a critical threshold $\alpha^* \in (0, 1)$ such that for all $\alpha > \alpha^*$, the ground truth coupling strictly minimizes the objective relative to the feature-only alternative: $J(\Gamma_{GT}, \alpha) < J(\Gamma_{FO}, \alpha)$.*

*Proof.* Let the source measure be $\mu$ and the target measure be $\nu$, with the variance of the Normal distributions set to $\sigma^2 = 0.1$. The centroids are located at $\mu_0 = (-2, 2), \mu_1 = (0, 2), \mu_2 = (2, 2)$ and $\mu_0' = (2, -2), \mu_1' = (0, -2), \mu_2' = (-2, -2)$. The interaction tensors $G^{(0)}$ and $G^{(1)}$ encode directed edges from the middle cluster ($k = 1$) to the left ($k = 0$) via Channel 1, and to the right ($k = 2$) via Channel 2. In what follows, we first compute the *unnormalized* feature and structure costs. We then incorporate the normalization scheme of Section D.5 in the final threshold derivation.

**1. Analysis of the feature cost $\mathcal{F}$**

For the ground truth coupling $\Gamma_{GT}$, each cluster $k$ maps to its true image $\mu_k'$. The cost is the mean squared norm of the translation vectors $v_0 = (4, -4)$, $v_1 = (0, -4)$, and $v_2 = (-4, -4)$:

$$\mathcal{F}(\Gamma_{GT}) = \frac{1}{3}\sum_{k=0}^{2}||v_k||^2 = \frac{1}{3}(32 + 16 + 32) = \frac{80}{3}. \tag{21}$$

For the feature-only coupling $\Gamma_{FO}$, $\mathcal{S}_0$ maps to $\mathcal{S}_2'$, $\mathcal{S}_1$ to $\mathcal{S}_1'$, and $\mathcal{S}_2$ to $\mathcal{S}_0'$. In the finite sample regime with $N$ points, the optimal transport cost between two empirical Gaussian distributions with identical covariance matrices converges to the squared Euclidean distance between their means. We denote the finite-sample deviation by $\delta_N$:

$$\mathcal{F}(\Gamma_{FO}) = \frac{1}{3}\left(||\mu_0 - \mu_2'||^2 + ||\mu_1 - \mu_1'||^2 + ||\mu_2 - \mu_0'||^2\right) + \delta_N. \tag{22}$$

Hence:

$$\mathcal{F}(\Gamma_{FO}) = 16 + \delta_N. \tag{23}$$

The term $\delta_N$ represents the error between the empirical measures and their population counterparts. For distributions in dimension $d = 2$, this error decays at a rate of $\delta_N = O(N^{-1/2})$ (Fournier & Guillin, 2015). Provided $N$ is sufficiently large, standard OT ($\alpha = 0$) prefers the incorrect mapping since $16 < \frac{80}{3}$.

**2. Analysis of structure cost $\mathcal{S}$**

The structure cost is the Gromov-Wasserstein cost:

$$\mathcal{S}(\Gamma) = \sum_{i,k} \sum_{j,l} ||G_{ik}^{(0)} - G_{jl}^{(1)}||^2 \Gamma_{ij} \Gamma_{kl}. \tag{24}$$

Since $\Gamma_{GT}$ maps every source cluster $k$ to the target cluster with the same index $k$, and $G^{(1)}$ is defined to preserve the index-based structure of $G^{(0)}$, we have:

$$\mathcal{S}(\Gamma_{GT}) = 0. \tag{25}$$

For $\Gamma_{FO}$, the mapping permutes indices as $\pi(0) = 2, \pi(1) = 1, \pi(2) = 0$. We evaluate the cost for the two active interactions in $G^{(0)}$:

- **Edge** $1 \to 0$ **(Channel 1):** The source relation is $[1, 0]^\top$ and the target relation (from $\pi(1) = 1$ to $\pi(0) = 2$) is Channel 2 ($[0, 1]^\top$), which yields a squared difference of 2 with mass weight $1/9$.

- **Edge** $1 \to 2$ **(Channel 2):** The source relation is $[0, 1]^\top$ and the target relation (from $\pi(1) = 1$ to $\pi(2) = 0$) is Channel 1 ($[1, 0]^\top$), which yields a squared difference of 2 with mass weight $1/9$.

The total structure cost is:

$$\mathcal{S}(\Gamma_{FO}) = \frac{1}{9} \times 2 + \frac{1}{9} \times 2 = \frac{4}{9}. \tag{26}$$

**3. Threshold derivation with normalization**

We now incorporate the normalization scheme of Section D.5. Define the (unnormalized) feature and structure gaps between the two couplings as

$$\Delta\mathcal{F} := \mathcal{F}(\Gamma_{FO}) - \mathcal{F}(\Gamma_{GT}), \qquad \Delta\mathcal{S} := \mathcal{S}(\Gamma_{FO}) - \mathcal{S}(\Gamma_{GT}). \tag{27}$$

From the computations above,

$$\Delta\mathcal{F} = 16 + \delta_N - \frac{80}{3}, \qquad \Delta\mathcal{S} = \frac{4}{9}. \tag{28}$$

For sufficiently large $N$, we have $\mathcal{F}(\Gamma_{GT}) > \mathcal{F}(\Gamma_{FO})$, so $|\Delta\mathcal{F}| = \mathcal{F}(\Gamma_{GT}) - \mathcal{F}(\Gamma_{FO}) > 0$ and the normalization is well-defined. Normalizing, we get:

$$\tilde{\mathcal{F}}(\Gamma) = \frac{\mathcal{F}(\Gamma)}{|\Delta\mathcal{F}|}, \qquad \tilde{\mathcal{S}}(\Gamma) = \frac{\mathcal{S}(\Gamma)}{\Delta\mathcal{S}}. \tag{29}$$

The normalized FGW objective can therefore be written as

$$J(\Gamma, \alpha) = (1 - \alpha)\tilde{\mathcal{F}}(\Gamma) + \alpha\tilde{\mathcal{S}}(\Gamma). \tag{30}$$

For the two couplings of interest, we obtain

$$\tilde{\mathcal{S}}(\Gamma_{GT}) = \frac{\mathcal{S}(\Gamma_{GT})}{\Delta\mathcal{S}} = 0, \qquad \tilde{\mathcal{S}}(\Gamma_{FO}) = \frac{\mathcal{S}(\Gamma_{FO})}{\Delta\mathcal{S}} = 1, \tag{31}$$

and

$$\tilde{\mathcal{F}}(\Gamma_{GT}) - \tilde{\mathcal{F}}(\Gamma_{FO}) = \frac{\mathcal{F}(\Gamma_{GT}) - \mathcal{F}(\Gamma_{FO})}{|\Delta\mathcal{F}|} = \frac{|\Delta\mathcal{F}|}{|\Delta\mathcal{F}|} = 1. \tag{32}$$

We seek $\alpha$ such that $J(\Gamma_{GT}, \alpha) < J(\Gamma_{FO}, \alpha)$ under this normalization, i.e.

$$(1 - \alpha)\tilde{\mathcal{F}}(\Gamma_{GT}) < (1 - \alpha)\tilde{\mathcal{F}}(\Gamma_{FO}) + \alpha. \tag{33}$$

Using $\tilde{\mathcal{F}}(\Gamma_{GT}) - \tilde{\mathcal{F}}(\Gamma_{FO}) = 1$, this inequality becomes

$$(1 - \alpha) < \alpha \iff \alpha > \frac{1}{2}. \tag{34}$$

Hence, under the normalization of Section D.5 and in the asymptotic regime, a critical threshold $\alpha^* = 1/2$ exists above which the ground truth coupling strictly improves the normalized objective relative to the feature-only alternative: $J(\Gamma_{GT}, \alpha) < J(\Gamma_{FO}, \alpha)$ for all $\alpha > 1/2$. $\qquad\square$

**Remarks.** In theory, $\alpha^* = 0.5$ comes from an idealized analysis of the normalized objective that only compares the feature-only and structure-only couplings in the *population limit*. Finite-sample effects, approximate normalization, and the existence of many 'almost-correct' couplings break the symmetry of the idealized setting and make the optimal $\alpha$ slightly bigger than $0.5$.

Second, Theorem 1 shows that the ground truth coupling $\Gamma_{GT}$ is better than $\Gamma_{FO}$ at $\alpha = 1$. However, it is not the *only* one. The interaction tensors $G^{(0)}$ and $G^{(1)}$ are constant for all points within a cluster. Therefore, the structure cost $\mathcal{S}(\Gamma)$ depends only on which clusters are matched, not on how individual points are mapped within them.

Any coupling that correctly maps source clusters to their corresponding target clusters yields a structure cost of $0$. This includes the ground truth coupling $\Gamma_{GT}$, but also any coupling that correctly matches clusters while randomly permuting points inside them. This explains the results observed in Figure 3 : at $\alpha = 1$, the solver returns a solution that is structurally perfect but fails to recover the exact point-to-point correspondence.

## F.2   DYNAMIC INTERPRETATION OF IADOT

We provide a dynamic viewpoint on IADOT, showing that it can be seen as the solution of a joint static-dynamic energy minimization problem combining kinetic energy in expression space and a structure-preserving term.

As before, let
$$\Pi(a, b) := \left\{ \Gamma \in \mathbb{R}_+^{n_0 \times n_1} : \Gamma \mathbf{1}_{n_1} = a,\ \Gamma^\top \mathbf{1}_{n_0} = b \right\}.$$
We further consider the common choice of feature cost
$$C_{ij} = \|x_i - y_j\|^2, \qquad 1 \le i \le n_0,\ 1 \le j \le n_1. \tag{35}$$

**Admissible processes for a fixed coupling.** Let $\Gamma \in \Pi(a, b)$ and define the associated joint law on endpoints
$$\Pi_\Gamma := \sum_{i=1}^{n_0} \sum_{j=1}^{n_1} \Gamma_{ij}\, \delta_{(x_i, y_j)}. \tag{36}$$
In the balanced case $\sum_{i,j} \Gamma_{ij} = 1$, so $\Pi_\Gamma$ is a probability measure with marginals $\rho_0, \rho_1$.

We consider continuous-time processes $(X_t)_{t \in [0,1]}$ taking values in $\mathbb{R}^d$ and satisfying:

- $X_.$ has almost surely absolutely continuous paths
- the joint law of its endpoints is $(X_0, X_1) \sim \Pi_\Gamma$

We write $\mathcal{A}(\Pi_\Gamma)$ for the class of all such processes. For any $X_. \in \mathcal{A}(\Pi_\Gamma)$, define the kinetic energy as:
$$\mathcal{K}(X_.) := \mathbb{E}\left[ \int_0^1 \|\dot{X}_t\|^2\, dt \right]. \tag{37}$$

The following lemma is standard but we include it for completeness.

**Lemma 1.** *Let $x, y \in \mathbb{R}^d$ and let $\gamma : [0, 1] \to \mathbb{R}^d$ be absolutely continuous with $\gamma(0) = x$, $\gamma(1) = y$. Then*
$$\int_0^1 \|\dot{\gamma}(t)\|^2\, dt \ge \|y - x\|^2, \tag{38}$$
*with equality if and only if $\gamma(t) = (1 - t)x + ty$ for all $t \in [0, 1]$.*

*Proof.* By Cauchy–Schwarz inequality,
$$\left\| \int_0^1 \dot{\gamma}(t)\, dt \right\|^2 \le \int_0^1 \|\dot{\gamma}(t)\|^2\, dt, \tag{39}$$
with equality if and only if $\dot{\gamma}(t)$ is constant in $t$. Since $\gamma(1) - \gamma(0) = y - x$, this yields
$$\|y - x\|^2 = \left\| \int_0^1 \dot{\gamma}(t)\, dt \right\|^2 \le \int_0^1 \|\dot{\gamma}(t)\|^2\, dt, \tag{40}$$

Equality holds if and only if $\dot{\gamma}(t) = y - x$, i.e. $\gamma(t) = (1-t)x + ty$. ☐

**Proposition 1.** *Let $\Gamma \in \Pi(a, b)$ and $\Pi_\Gamma$ be as above. Consider the admissible class $\mathcal{A}(\Pi_\Gamma)$ and the kinetic energy $\mathcal{K}$. Then:*

*1. The energy $\mathcal{K}(X.)$ is minimized over $\mathcal{A}(\Pi_\Gamma)$ by the process*

$$X_t^{\mathrm{lin}} := (1-t)X + tY, \qquad (X, Y) \sim \Pi_\Gamma. \tag{41}$$

*2. The minimal value of the kinetic energy is*

$$\inf_{X. \in \mathcal{A}(\Pi_\Gamma)} \mathcal{K}(X.) = \mathbb{E}_{(X,Y) \sim \Pi_\Gamma}\big[\|X - Y\|^2\big] = \sum_{i,j} \Gamma_{ij} C_{ij}. \tag{42}$$

*Proof.* Any $X. \in \mathcal{A}(\Pi_\Gamma)$ satisfies $(X_0, X_1) \sim \Pi_\Gamma$. Condition on the endpoints:

$$\mathcal{K}(X.) = \mathbb{E}_{(X,Y) \sim \Pi_\Gamma}\left[\mathbb{E}\Big[\int_0^1 \|\dot{X}_t\|^2 \, dt \,\Big|\, (X_0, X_1) = (X, Y)\Big]\right]. \tag{43}$$

For each fixed pair $(X, Y) = (x, y)$, Lemma 1 shows that the conditional energy is minimized by the straight-line path $t \mapsto (1-t)x + ty$, with minimal value $\|x - y\|^2$. Thus the global minimizer over $\mathcal{A}(\Pi_\Gamma)$ is the straight-line process $X_t^{\mathrm{lin}}$, and

$$\inf_{X. \in \mathcal{A}(\Pi_\Gamma)} \mathcal{K}(X.) = \mathbb{E}_{(X,Y) \sim \Pi_\Gamma}\Big[\|X - Y\|^2\Big] = \sum_{i,j} \Gamma_{ij} \|x_i - y_j\|^2 = \sum_{i,j} \Gamma_{ij} C_{ij}. \tag{44}$$

☐

**Joint static–dynamic energy and reduction to FGW.** We can view `IADOT` as minimizing over both couplings and dynamics the joint energy functional

$$\mathcal{E}_\alpha(\Gamma, X.) := (1-\alpha)\,\mathcal{K}(X.) + \alpha\,S(\Gamma), \tag{45}$$

subject to $\Gamma \in \Pi(a, b)$ and $X. \in \mathcal{A}(\Pi_\Gamma)$.

**Proposition 2.** *Fix $\alpha \in [0, 1]$. Consider the optimization problem*

$$\inf_{\Gamma \in \Pi(a,b)} \inf_{X. \in \mathcal{A}(\Pi_\Gamma)} \mathcal{E}_\alpha(\Gamma, X.) = \inf_{\Gamma \in \Pi(a,b)} \inf_{X. \in \mathcal{A}(\Pi_\Gamma)} \Big[(1-\alpha)\mathcal{K}(X.) + \alpha S(\Gamma)\Big]. \tag{46}$$

*Then:*

*1. For any fixed $\Gamma$, the inner infimum over $X. \in \mathcal{A}(\Pi_\Gamma)$ is attained by the straight-line process $X_t^{\mathrm{lin}} = (1-t)X + tY$, $(X, Y) \sim \Pi_\Gamma$, and*

$$\inf_{X. \in \mathcal{A}(\Pi_\Gamma)} \mathcal{E}_\alpha(\Gamma, X.) = (1-\alpha) \sum_{i,j} \Gamma_{ij} C_{ij} + \alpha S(\Gamma). \tag{47}$$

*2. Consequently, the joint static–dynamic problem reduces to the purely static FGW problem*

$$\inf_{\Gamma \in \Pi(a,b)} \inf_{X. \in \mathcal{A}(\Pi_\Gamma)} \mathcal{E}_\alpha(\Gamma, X.) = \inf_{\Gamma \in \Pi(a,b)} \Big[(1-\alpha)\,\langle\Gamma, C\rangle_F + \alpha S(\Gamma)\Big], \tag{48}$$

*whose minimizers are exactly the FGW-optimal couplings used by `IADOT`.*

*Proof.* Point (1) follows directly from Proposition 1: for any $\Gamma$,

$$\inf_{X. \in \mathcal{A}(\Pi_\Gamma)} \mathcal{E}_\alpha(\Gamma, X.) = (1-\alpha) \inf_{X. \in \mathcal{A}(\Pi_\Gamma)} \mathcal{K}(X.) + \alpha S(\Gamma) = (1-\alpha) \sum_{i,j} \Gamma_{ij} C_{ij} + \alpha S(\Gamma). \tag{49}$$

Taking the infimum over $\Gamma \in \Pi(a, b)$ yields (2), which coincides with the static FGW objective. ☐

Intuitively, Proposition 2 shows that `IADOT` does not use linear interpolations between matched cells as a heuristic, but as the *unique* minimal-action choice once the coupling $\Gamma^\star$ is fixed. The static FGW step therefore selects an interaction-aware coupling that trades off feature displacement and CCI preservation, and the subsequent dynamic step realizes this coupling by approximating the lowest-kinetic-energy flow in expression space. When $\alpha = 0$, this recovers the classical OT–CFM (Tong et al., 2024)/ Benamou-Brenier (Benamou & Brenier, 2000) interpolation.

### F.3 CONNECTION TO THE VELOCITY FIELD LEARNT WITH CFM.

In practice, IADOT does not explicitly construct the process $X_t^{\mathrm{lin}}$ but instead uses CFM to learn a time–dependent vector field $v_\theta$ that generates the same probability path.

In the infinite–capacity and optimization limit, the minimizer $v^\star$ of $\mathcal{L}_{\mathrm{CFM}}$ coincides with the velocity field of $X_t^{\mathrm{lin}}$ constructed in Section F.2, in the sense that

$$v^\star(z, t) = \mathbb{E}[\, Y - X \mid Z_t = z \,],$$

and the ODE

$$\dot{z}_t = v^\star(z_t, t)$$

generates exactly the probability path $\{\rho_t\}_{t \in [0,1]}$ induced by $\Gamma^\star$.

We can relate $v^\star$ to the kinetic energy of the straight–line process, following a similar technique as in (Lipman et al., 2024). For a time–dependent vector field $w : \mathbb{R}^d \times [0,1] \to \mathbb{R}^d$ that generates $\{\rho_t\}$, define its kinetic energy along this path by

$$\mathcal{K}_{\mathrm{Eul}}(w) := \int_0^1 \mathbb{E}_{Z_t \sim \rho_t}\Big[\, \|w(Z_t, t)\|_2^2 \,\Big]\, dt.$$

Using the formula of $v^\star$ above and Jensen's inequality, we obtain

$$\mathcal{K}_{\mathrm{Eul}}(v^\star) = \int_0^1 \mathbb{E}\Big[\, \big\|\mathbb{E}[\, Y - X \mid Z_t \,]\big\|_2^2 \,\Big]\, dt$$

$$\leq \int_0^1 \mathbb{E}\Big[\, \mathbb{E}\big[\|Y - X\|_2^2 \mid Z_t\big] \,\Big]\, dt$$

$$= \int_0^1 \mathbb{E}\Big[\, \|Y - X\|_2^2 \,\Big]\, dt$$

$$= \mathbb{E}_{(X,Y) \sim \Pi}\Big[\, \|Y - X\|_2^2 \,\Big].$$

By Proposition 1 we have

$$\mathbb{E}_{(X,Y) \sim \Pi_{\Gamma^\star}}\Big[\, \|Y - X\|_2^2 \,\Big] = \sum_{i,j} \Gamma_{ij}^\star C_{ij} = K\big(X_\cdot^{\mathrm{lin}}\big),$$

Hence

$$\mathcal{K}_{\mathrm{Eul}}(v^\star) \;\leq\; \sum_{i,j} \Gamma_{ij}^\star C_{ij} = K\big(X_\cdot^{\mathrm{lin}}\big).$$

In other words, for a fixed coupling $\Gamma$, the feature term

$$F(\Gamma) = \langle \Gamma, C \rangle_F = \sum_{i,j} \Gamma_{ij} C_{ij}$$

provides an explicit upper bound on the kinetic energy of the velocity field recovered by CFM from the corresponding straight–line dynamics. Combined with the joint static–dynamic formulation in Equation (45), this shows that the IADOT objective

$$(1 - \alpha)\, \langle \Gamma, C \rangle_F + \alpha\, S(\Gamma)$$

can be viewed as selecting a coupling that balances CCI preservation with a surrogate upper bound on the kinetic energy of the flow that CFM learns from that coupling.

## G LLM usage

We used large language models (LLMs) to assist with improving the clarity of writing and refining the formatting of tables and figures. LLMs were not used for research ideation, experimental design, analysis, or any substantive contributions that would merit authorship.

