# OpenReview forum: "Learning Cellular Dynamics with Cell–Cell Interaction–Aware Optimal Transport"
_ICLR.cc/2026/Conference — Submitted to ICLR 2026_

### Official Review · Reviewer_ja6u · 2025-10-27

**Soundness:** 2
**Presentation:** 3
**Contribution:** 2
**Rating:** 2
**Confidence:** 4

**Summary:**

This paper proposes IADOT, a two-stage pipeline for inferring cellular dynamics from scRNA-seq snapshots by (i) computing a structure-aware cross-snapshot coupling with a Fused Gromov–Wasserstein (FGW) objective that combines expression distances with a cell–cell interaction (CCI) prior derived from ligand–receptor (LR) expression, and (ii) fitting a time-dependent velocity field with Conditional Flow Matching (CFM) induced by the coupling. Empirically, the authors report lower interpolation errors (W1/W2) on several datasets and demonstrate in-silico LR catalog edits that decrease a tumour progression proxy.

**Strengths:**

1. Well-presented formulation. The paper is explicit about (a) constructing CCI tensors from LR expressions and (b) inserting them into an FGW objective; then converting a coupling into a continuous flow via CFM. This makes the pipeline replicable.
2. Simple prior to enabling sensitivity tests. Because structure enters as an explicit prior (CCI tensor, α), the paper can perform catalog-edit counterfactuals and quantify effects on a progression proxy.
3. This problem is important.

**Weaknesses:**

1. **Scope and novelty**. This method can be viewed as substituting the feature-only OT coupling commonly used in CFM-style trajectories with an FGW coupling that incorporates a CCI term, and then fitting dynamics via standard CFM on the resulting affine interpolants. The paper does not seem to introduce a new dynamic objective beyond this substitution, and the theoretical discussion primarily shows that CFM reproduces the coupling-induced linear path rather than establishing guarantees for transporting the empirical endpoints under a broader dynamic optimality principle for FGWOT. Making explicit what is (and is not) guaranteed would strengthen the positioning. From this perspective, the technical contributions are incremental. “Interaction-aware” is implemented as a generic FGW structure term, not a CCI-specific dynamic model. The framework turns LR information into static multi-channel adjacency and then optimizes a generic FGW objective; this idea is not specific to CCIs—any typed graph prior could be dropped in. Similar ideas have also been presented in Moscot and many spatial OT alignment methods.
2. **Static CCI assumption** The method constructs snapshot-wise static tensors and encourages their cross-snapshot persistence, yet cellular neighborhoods and communication partners typically reconfigure between times. The static-graph prior may bias trajectories when transient interactions matter.
3. **Sensitivity to CCI false positives.** Single-cell LR-based CCIs may contain a high rate of false positives. The sensitivity analysis (Table 3) shows that shuffling CCIs or using random LR catalogs could change interpolation performance, which may suggest some dependence on the specified CCI structure and could be influenced by the CCI false positives.
4. **Computational cost and parameter selection.** The method introduces several hyperparameters—most notably the trade-off weight $\alpha$ between feature and structure terms, and Hill-function parameters $K_g$, $h_g$—whose values are chosen empirically by grid search on interpolation metrics. The paper does not provide principled guidance or sensitivity analysis for these settings. Moreover, the FGW solver and subsequent CFM training add considerable runtime and memory overhead.
5. **Lack of unbalanced and stochastic modeling.** The proposed framework assumes balanced mass transport and deterministic dynamics, yet biological systems often involve cell proliferation, death, and stochastic transcriptional noise. Ignoring unbalanced or stochastic components limits the model’s biological realism and comparability to Schrödinger Bridge or unbalanced OT (UOT) formulations, which explicitly capture mass creation or diffusion effects.

Klein, Dominik, et al. "Mapping cells through time and space with moscot." Nature 638.8052 (2025): 1065-1075.

**Questions:**

1. **Clarity of theoretical contribution** The dynamic formulation appears to reuse the standard Conditional Flow Matching (CFM) objective with an FGW coupling that includes CCI information. Could the authors clarify what theoretical advances this introduces beyond substituting the coupling step? In particular, does the proposed FGWOT formulation guarantee valid optimal transport of endpoint distributions? If so, could the authors formally state and prove it?
2. **Specificity of “interaction-aware” modeling** Since the FGW structure term could accept any typed graph prior, to what extent is the method specific to cell–cell interactions rather than general structured alignment?
3. **Static CCI assumption** The model constructs snapshot-specific static CCI tensors and enforces persistence across time, but cellular communication networks typically reconfigure dynamically. Could the authors discuss how this static assumption affects trajectories when transient or time-varying interactions are present?
4. **Sensitivity to CCI noise** Given that single-cell LR-based CCIs often contain false positives, Table 3 suggests that interpolation quality changes when CCIs are shuffled or randomized. How robust is the method to such noise in the LR catalog, and could uncertainty propagation or perturbation analysis be added to quantify sensitivity?
5. **Computational cost and hyperparameter robustness** The approach relies on several empirically tuned parameters (e.g., $\alpha$, $K_g$, $h_g$) and involves a computationally intensive FGW solver. Could the authors report runtime, memory scaling, and parameter sensitivity, or discuss practical guidelines for applying the method to larger atlas-level datasets?
6. **Perturbation design** The in-silico perturbation experiment modifies only the CCI matrix (by removing selected LR pairs) while keeping expression data fixed. Is this design biologically justified? Would jointly perturbing expression or signaling outputs yield different dynamics, and how robust are the current results to that modeling choice?

My comments are based on my current understanding. At this stage, I find it difficult to be more positive, but I would be happy to reconsider my evaluation if the authors provide a thorough and convincing rebuttal addressing these points.

Only language polishing was assisted by an LLM; the review’s analysis and conclusions are the reviewer’s own.

---

> ### Author Response · Authors · 2025-11-24
> **Response (1/5)**
>
> Thank you for your constructive feedback and for acknowledging the importance of the problem tackled by our work. In what follows, we provide a point-by-point response to your commentary, which has been of great assistance in strengthening the manuscript.
>
> ---
> # [P1] Scope and novelty
>
> We appreciate the opportunity to clarify the contributions of $\texttt{IADOT}$. While we utilize the mathematical frameworks of Fused Gromov-Wasserstein (FGW) and Conditional Flow Matching (CFM), we respectfully disagree that our contribution is incremental. Our novelty lies in **(1) a domain-specific construction** of the structural prior that is distinct from generic graph regularization, **(2) a multi-channel optimization** scheme required to handle this specific structure, and **(3) a modular design** that solves the identifiability problem across various dynamic models.
>
> **1. Novelty lies in the prior construction**
>
> The reviewer suggests the "interaction-aware" term is generic. We argue that our formulation is strictly specific to the biological domain and distinct from standard graph Laplacians or kNN graphs used in other FGW applications.
>
> * **Inferred, directed, typed structure:** Unlike methods that rely on **external** priors like physical coordinates (e.g. spatial OT methods [1]) or **undirected** geometric proximity (e.g. standard MOSCOT usage [2]), $\texttt{IADOT}$ infers a **directed** and **multi-channel** interaction tensor strictly from gene expression data. This allows trajectory inference in standard scRNA-seq data where no external structure is provided (e.g. spatial coordinates).
> * **Biological specificity:** The prior is not just "any graph." Our sensitivity analysis confirms that the performance gains are driven by the specific biological semantics of the Ligand-Receptor (LR) networks. As shown in `Section 5.5` and `Table 3`, replacing the curated LR catalog with random or shuffled interactions degrades performance. This proves that $\texttt{IADOT}$ leverages specific signaling biology, not just generic regularization.
>
> **2. Technical differentiation from standard FGW methods**
>
> We emphasize that $\texttt{IADOT}$ is not a vanilla application of FGW. The biological complexity of cell-cell interactions necessitates specific algorithmic adaptations that differentiate it from existing baselines.
>
> * **Multi-channel solver:** Standard single-cell FGW implementations typically treat cells as nodes in a single-layer, undirected kNN graph. In contrast, to capture the complexity of signaling, $\texttt{IADOT}$ operates on **multi-typed** interaction tensors (where $G_{ij}$ is a vector in $\mathbb{R}^K$, not a scalar). This required us to implement a customized **multi-dimensional FGW solver** to handle vector-valued edge weights, which we detail in `Appendix D.4`.
> * **Empirical superiority:** This technical distinction translates to performance. We have performed an additional experiment comparing $\texttt{IADOT}$ with MOSCOT (without spatial data, hence which uses a kNN graph), with results presented in `Table 2`, and we see that $\texttt{IADOT}$ consistently outperforms MOSCOT in interpolation tasks across the different datasets, indicating that our directed, multi-channel prior provides signal that generic graph alignment misses.
>
> **3. Modularity of the dynamic objective**
>
> The choice to not introduce a new flow matching loss is a deliberate design choice to ensure **modularity**.
>
> * **Identifiability via coupling:** The core challenge in scRNA-seq trajectory inference is the lack of ground-truth alignment [3]. We posit that the "optimality" of the trajectory is critically influenced the boundary conditions (the coupling) using biological priors, rather than enforcing specific constraints on the flow field itself.
> * **IADOT is plug-and-play:** By decoupling the prior learning (the coupling) from the dynamics learning (the flow), $\texttt{IADOT}$ acts as a flexible plug-and-play prior. To demonstrate this point, we conducted additional experiments, showing in `Table 2` and `Appendix E.4` that $\texttt{IADOT}$ improves performance when combined with various state-of-the-art dynamics models, including MFM [4], UOT-FM [5], and SF2M [6].

---

> ### Author Response · Authors · 2025-11-24
> **Response (2/5)**
>
> **To summarize:**  we establish that $\texttt{IADOT}$ tackles a specific identifiability issue through a novel, domain-specific prior.  **As noted in [3], "the adaptive selection of robust cost functions remains a crucial area for future research"** and our work answers this call by deriving transport costs from signaling biology rather than generic geometry.
>
> **Actions taken:**
> * **Clarified novelty:** We have revised `Section 3` to explicitly contrast our multi-channel, directed construction with standard undirected graph methods used in related work.
> * **Extended baselines:** We conducted experiments with additional baselines; the manuscript now contains **9 baselines** and **4 $\texttt{IADOT}$ variants** (`Table 2`) specifically highlighting the improvement over generic FGW-based approaches like MOSCOT.
>
> **References**
>
> [1] Cang, Zixuan, and Qing Nie. "Inferring spatial and signaling relationships between cells from single cell transcriptomic data."
>
> [2] Klein, Dominik, et al. "Mapping cells through time and space with moscot."
>
> [3] Bunne, Charlotte, et al. "Optimal transport for single-cell and spatial omics."
>
> [4] Kapusniak, Kacper, et al. "Metric flow matching for smooth interpolations on the data manifold."
>
> [5] Eyring, Luca, et al. "Unbalancedness in neural monge maps improves unpaired domain translation."
>
> [6] Tong, Alexander, et al. "Simulation-free schrodinger bridges via score and flow matching."
>
>
> ---
> # [P2] Theoretical guarantees
>
> Thank you for the question regarding theoretical guarantees. We address this along two axes: the validity of the boundary conditions and the recoverability of the ground truth under structural assumptions.
>
> **1. Validity of the OT plan (marginal constraints)**
>
> Our formulation guarantees valid marginals by design. As defined in `Section 4.1` and `Equation (5)`, the $\texttt{IADOT}$ objective minimizes the cost subject to the strict constraint that the coupling $\Gamma$ belongs to the transport polytope $\Pi(a,b)$. This polytope is defined as the set of all non-negative matrices whose row and column sums match the empirical source $a$ and target $b$ distributions, respectively.
>
> Because the optimization is strictly constrained to $\Pi(a,b)$, any optimal solution $\Gamma^*$ respects the source and target empirical measures exactly. Consequently, the continuous probability path $\{\rho_t\}_{t \in [0,1]}$ constructed via Conditional Flow Matching satisfies the boundary conditions $\rho_0$ and $\rho_1$ by construction.
>
> **2. Structural identifiability and recovery**
>
> Beyond boundary satisfaction, we provide a guarantee, for the synthetic setup in `Section 5.1`, that $\texttt{IADOT}$ assings a lower cost to the correct coupling, compared to the feature-only coupling when the underlying ground truth preserves interaction structure and for sufficiently big $\alpha$.
> * **Formal proof:** We provide a formal proof in `Appendix F` for the synthetic setup described in `Section 5.1`. We prove that there exists a critical threshold $\alpha^\ast$ such that for all $\alpha > \alpha^\ast$, the ground-truth transport map achieves a lower cost than the feature-only ($\alpha=0$) coupling.
> * **Empirical validation:** This theoretical result underpins the empirical success shown in `Figure 3`, where $\texttt{IADOT}$ successfully recovers the ground-truth diagonal trajectory that feature-only OT ($\alpha=0$) fails to identify, as evidenced by the Hits@1 metric.
>
> **Action taken:** We have clarified these guarantees in the manuscript, specifically pointing to `Appendix F` for the full proof of the identifiability theorem.

---

> ### Author Response · Authors · 2025-11-24
> **Response (3/5)**
>
> # [P3] Static CCI assumption
> Thank you for raising the important point that cellular neighborhoods and communication partners typically reconfigure over time. We agree that assuming rigid structural persistence can be risky; however, $\texttt{IADOT}$ is explicitly designed to treat interaction persistence as a **tunable inductive bias** rather than a hard constraint. We address the impact of transient interactions through the following observations:
>
> **1. Soft constraint via $\alpha$**
> The parameter $\alpha$ allows the model to balance feature similarity (standard transcriptomic evolution) with structural persistence in the FGW objective. It does *not* enforce a rigid graph match.
> * **Empirical evidence:** As shown in the interpolation results (`Figure 4`), we observe a "U-shaped" curve for the Lung Tumour and V1 Cortex datasets, where an intermediate $\alpha$ (e.g., $\alpha \approx 0.5-0.7$) yields optimal performance. This demonstrates that the model successfully leverages *partial* structural information without enforcing a rigid structure ($\alpha=1$) that might bias trajectories.
> * **Selection:** In practice, $\alpha$ can be selected by computing interpolation metrics on held-out timepoints (`Section 5.2`), meaning that it can be selected in a **dataset-dependent** way.
>
> **2. Datasets with rapid remodeling**
> We explicitly tested $\texttt{IADOT}$ on a system where the "static assumption" fails due to rapid remodeling: the developing mouse embryo dataset (`Section 5.6`).
> In this dataset, tissue composition changes drastically over long intervals (6 days), making the CCI structure largely uninformative. As shown in `Figure 6`, $\texttt{IADOT}$’s performance does not degrade catastrophically; rather, the interpolation error curve is flat or best at $\alpha = 0$. This confirms that when the static CCI assumption does not hold, $\texttt{IADOT}$ naturally defaults to the feature-only coupling ($\alpha = 0$), ignoring the uninformative structural prior.
>
>
> ---
> # [P4] Sensitivity to CCI noise and false positives
>
> Thank you for your comment regarding the CCI and false positives. We agree that inferring CCIs from single-cell data is prone to false positives due to dropout and the lack of spatial context. We have addressed this through rigorous preprocessing and extensive sensitivity analyses in the appendix. We structure our response below to distinguish between **sensitivity to biological signal** (which is desired) and robustness to noise.
>
> **1. Sensitivity to structure (`Table 3`) demonstrates specificity.**
> You correctly note that `Table 3` shows a performance drop when using random LR catalogs or shuffled CCIs. We argue this is a necessary property of a structure-aware method. If $\texttt{IADOT}$ performed equally well with a shuffled CCI (which effectively simulates false positives) as it did with the original CCIs, it would imply the prior is uninformative. The performance gap confirms that $\texttt{IADOT}$ is successfully leveraging the specific biological signal in the true catalog. Importantly, while "Random LR" performs worse than the true catalog, it does not lead to catastrophic failure. This indicates that while false positives dilute the benefit of the prior, they do not break the underlying transport formulation.
>
> **2. Additional perturbation analysis  (`Appendix E.5`).**
>  Following your suggestion, we have performed an additional perturbation analysis in `Appendix E.5`.
> * **Setup:** For LR genes, we add zero-mean Gaussian noise to the gene expressions before applying the Hill transform and clip below by $0$, i.e. we define $\tilde{x}\_{cg} = \max(0,x\_{cg}+\epsilon\_{cg})$ where $\epsilon\_{cg} \sim \mathcal{N}(0,\sigma\_g^2)$.
> * **Results:** `Figure 13` demonstrates that as the noise scale increases, the interpolation error gradually increases across all three datasets. Again, this shows the specificity of our biological prior: no sensitivity with respect to noise would have meant that this prior is irrelevant. Furthermore, the performance is relatively robust to small level of noises for the Lung tumour and V1 Light datasets. Interestingly, for the V1 Light dataset, we see that small noise scaling improves the results, which we attribute to a small regularization / denoising effect. Adding a small amount of centered Gaussian noise before the Hill transform and clipping  pushes low-intensity ligand or receptor expressions to zero while leaving strongly expressed pairs essentially unchanged.

---

> ### Author Response · Authors · 2025-11-24
> **Response (4/5)**
>
> **3. Mitigating false positives with our LR selection (`Appendix E.3`).**
> To proactively mitigate false positives before optimization, we employ a strict construction pipeline detailed in `Appendix D.6` to choose the LR catalog. Rather than using all co-expressions, we utilize perform consensus rank aggregation, and filter interactions by both significance ($p < 0.05$) and expression magnitude thresholds. To validate that this selection matters and to test robustness to "functional" false positives,we compared these optimized tissue-specific priors against a "generic" list of canonical long-range interactions in an *additional experiment* conducted in `Appendix E.3`. `Table 22` shows that using the generic list (which effectively acts as a catalog with high false positives relative to the specific tissue) yields higher Wasserstein distances than our LR catalog, showing the importance of a robust LR selection procedure to mitigate false positives.
>
> **Actions taken:** (i) We conducted an additional perturbation analysis on the LR expressions in `Appendix E.5`. (ii) We compared our LR selection procedure to a generic LR catalog in `Appendix E.3`
>
>
> ---
> # [P5] Computational and memory overhead
>
>
> We appreciate your feedback regarding the potential overhead of our method. We have added a dedicated section (`Appendix D.10`) to the revised manuscript to provide a **detailed analysis** of the computational complexity and wall-clock runtimes, which we briefly summarize here.
>
> **Wall-clock runtime.** We report wall-clock runtimes in `Table 19`, decomposing the total time into the  OT step and the Flow Matching step. This table demonstrates that $\texttt{IADOT}$ runs in the order of **minutes** on standard hardware, rather than adding prohibitive overhead.
>
> **Memory complexity and implementation.** We have clarified in `Appendix D.10` that our implementation manages memory efficiently by avoiding the materialization of the full pairwise structure tensor. Instead of constructing a tensor of size $N \times N \times K$, we implement the structure term via matrix products (computing $C_{1}^{(r)}\Gamma(C_{2}^{(r)})^{\top}$). Consequently, the memory complexity scales as $\mathcal{O}(K(n_{0}^{2}+n_{1}^{2})+n_{0}n_{1})$. While quadratic in cell number (as is *typical* for feature-only OT), it is linear in the number of interaction channels $K$.
>
> **Scalability.** For very large datasets where computational quadratic scaling is a bottleneck, we discuss mitigation strategies in `Appendix D.10`: i) use entropy-regularized OT objectives to accelerate convergence [1] ii) solve the OT problem at the minibatch level [2] instead of solving it at the dataset level iii) Use metacells to reduce the effective sample size, as explained in `Appendix D.3` and evaluated in `Section 5.5`.
>
> **Action taken:** We have added a detailed discussion on computation cost, wall-clock runtime, memory overhead and strategies to reduce them in `Appendix D.10`.
>
> **References**
>
> [1] Peyré, Gabriel, and Marco Cuturi. "Computational optimal transport: With applications to data science."
>
> [2] Tong, Alexander, et al. "Improving and generalizing flow-based generative models with minibatch optimal transport."
>
> ---
> # [P6] Unbalanced and stochastic modelling
> Thank you for mentioning extensions of $\texttt{IADOT}$ to the unbalanced and stochastic settings. Following your suggestions, we have formalized extensions of $\texttt{IADOT}$ to both of these settings and have conducted additional experiments with these extensions.
>
> **Unbalanced mass transport.** We have extended $\texttt{IADOT}$ to the unbalanced setting to explicitly account for cell birth and death. We implemented a two-step procedure detailed in `Appendix D.11`: first, we infer non-uniform marginals using a feature-only unbalanced OT formulation; then, we optimize the interaction-aware FGW objective using these reweighted marginals.
>
> **Results.** This extension, denoted as $\texttt{IADOT + UOT-FM}$, consistently outperforms the feature-only UOT-FM baseline as can be seen in `Table 2`. This confirms that our interaction prior improves alignment even when mass conservation constraints are relaxed.

---

> ### Author Response · Authors · 2025-11-24
> **Response (5/5)**
>
> **Stochastic bridges.** To address the concern regarding deterministic dynamics, we demonstrate that $\texttt{IADOT}$ integrates seamlessly with stochastic frameworks. We demonstrate this with SF2M [1]. As detailed in `Appendix D.13`, we integrated $\texttt{IADOT}$ with SF2M by using the structure-aware coupling $\Pi$ to sample training triples for the Brownian bridge mixture, an extension we denote as $\texttt{IADOT+SF2M}$.
>
> **Results:** We report interpolation results in `Table 2`. We find that adding $\texttt{IADOT}$'s interaction prior improves performance over the standard SF2M baseline. This demonstrates that $\texttt{IADOT}$ is compatible with stochastic formulations.
>
> **Action taken:** We have formulated and conducted additional experiment with extensions of $\texttt{IADOT}$ to (i) the unbalanced setting (ii) stochastic dynamics, showing that it consistently improves trajectory inference. Results are shown in `Table 2`.
>
>
> **Reference**
>
> [1] Tong, Alexander, et al. "Simulation-free schrodinger bridges via score and flow matching."
>
> ---
> # [P7] Hill function sensitivity analysis
>
> To answer your point on the sensitivity with respect to the Hill transform hyperparameters, we conducted an additional sensitivity analysis on the hyperparameters $K_g$ and $h_g$, used to define the interaction scores $s_{cg}=x_{cg}^{h_g}/(x_{cg}^{h_g}+K_g^{h_g})$.
> We consider different values of the percentile level $p \in \\{80, 90, 99\\}$ (with $K_g = Q_g(p)$ denoting the $p$-th percentile of $\\{ x_{cg} \mid c \in [n]\\}$) and $h_g \in \\{1,2,4\\}$, for $\alpha = 1$. We report the interpolation results in `Figure 14` in `Appendix E.6`. We observe that the performance is largely insensitive to the specific choice of these parameters. This stability justifies the use of standard default values ($90$th percentile and $h_g=1$) across our experiments without the need for extensive per-dataset tuning.
>
> **Action taken:** We performed an additional sensitivity analysis on the Hill function parameters, in `Appendix E.6`.
>
> ---
>
> # [P8] Perturbation design and biological justification
>
> We appreciate the reviewer’s question regarding the biological justification of our in-silico perturbation design (`Section 5.4`). We argue that modifying the CCI prior while holding expression fixed is the methodologically correct approach for modeling acute interventions.
>
> **1. Biological justification.**
> Our design models an **acute pharmacological blockade** (e.g., antibody therapy). Biologically, receptor inhibition occurs immediately, whereas transcriptional adaptation follows a lag phase [1]. By holding initial expression fixed, we accurately model the cell state at the exact moment of intervention. "Jointly perturbing" expression would simulate a post-adaptation state, hence assuming the outcome the model is designed to predict.
>
> **2. Separating cell state from dynamics**
> $\texttt{IADOT}$ distinguishes between **cell state** (gene expression) and the **rule of evolution** (the transport coupling). Removing an LR pair from the CCI matrix effectively removes the "structural reward" for that signaling axis. The model is thus forced to infer a *new* trajectory conditional on the signal being unavailable. This isolates the causal role of communication without confounding the analysis with arbitrary changes to the starting population.
>
> **3. Robustness and specificity**
> We empirically validated that this design yields specific, biologically plausible shifts rather than random noise:
> * **Specificity:** Ablating relevant tumor pathways (EGFR, ALK, MET) significantly altered trajectories (up to 15.5% shift in hallmark scores).
> * **Robustness:** Ablating irrelevant control pathways (RAAS, Vasopressin) resulted in **no change**.
> * **Coupling stability:** Our analysis in `Appendix E.2` confirms that for targeted pathways, >94% of cells shifted their optimal coupling, whereas for controls, 0% shifted. This confirms the results are driven by specific pathway topology, not model instability.
>
> **Reference**
>
> [1] Lee et al., "Structural basis of checkpoint blockade by monoclonal antibodies in cancer immunotherapy,"
>
> ---
>
> ### Thank you for your feedback
>
> Thank you for your detailed and constructive feedback, which has greatly helped us to refine our manuscript. We believe the revisions, including the new baselines and their $\texttt{IADOT}$ variants, the clarification on biological validity *substantially improve* the clarity and rigor of our submission. We are confident that $\texttt{IADOT}$, by tackling the **crucial question of: "What constitutes a good cost function for inferring biological transport maps?"**, provides a practical step toward more biology-aligned methods in scRNA analysis and welcome continued dialogue.

---

> > ### Comment · Reviewer_ja6u · 2025-11-25
> >
> > I appreciate the authors’ very thorough experimental efforts, and I believe substantial time and work have been devoted to this revision. I especially appreciate the inclusion of new baselines, the additional sensitivity analyses, and the expanded set of experiments designed to address my and other reviewers' earlier questions. In recognition of these efforts, I am willing to raise my score accordingly.
> >
> > However, several aspects still do not fully address my concerns, particularly regarding the theoretical justification of FGW–FM in relation to the underlying optimal transport formulation. The analysis provided does not resolve my core concern. When considering cell–cell interactions, although CFM can yield non-linear trajectories, the assumption that the intermediate path between two states can be treated as a linear interpolation requires stronger justification. In OT-CFM, this is supported by an independent particle-system formulation and an established equivalence to the BB dynamic optimal transport form, which is not the case here. Without a theoretical guarantee of what problem is actually being solved in this setting, the explanation remains unconvincing to me. In addition, my concerns regarding the assumption of static cell–cell communications remain. In my view, this represents a fundamental difficulty in the problem considered here and is not satisfactorily resolved. Meanwhile, while I appreciate the authors’ clarifications, they do not largely change my current view of the contribution’s novelty. So I am sorry that I am unable to be more positive at this stage, and I wish the authors all the best moving forward.

---

> ### Author Response · Authors · 2025-11-25
> **Response (1/3)**
>
> Thank you for acknowledging our additional experiments and analyses, and for the willingness to increase the score. We appreciate your continued engagement with our work. Below, we address the remaining concerns, in particular the theoretical justification of FGW–CFM and the assumption of static cell–cell communications.
>
> ---
>
> ## [P1] Theory
>
> We appreciate the opportunity to make the theory clearer. We clarify below, and refer to the new `App. F.2–F.3` for full details.
>
>
> ### 1. Why we use linear interpolations in IADOT
>
> In `App. F.2` we give a dynamic formulation of IADOT that we summarize briefly now. This formulation allows us to clarify: 1) why we use linear interpolations 2) what implicit objective IADOT is solving.
>
> Let $\Gamma \in \Pi(a,b)$ be a coupling and $(X,Y)\sim \Pi_\Gamma$ denote the endpoints. Consider all continuous-time processes $(X_t)\_{t\in[0,1]}$ with absolutely continuous paths and $(X\_0,X\_1)\sim \Pi\_\Gamma$, and define the kinetic energy $K(X\_{\bullet}) := \mathbb{E}\Big[\int\_0^1 \||\dot X\_t\||^2  dt\Big].$
>
> 1. **Linear interpolations minimize the kinetic energy.** For a fixed $\Gamma$, the unique minimizer of $K(X_{\bullet})$ is the linear bridge $X_t^{\text{lin}} = (1-t)X + tY,$ with minimal value $K(X_{\bullet}^{\text{lin}}) = \mathbb{E}\||X-Y\||^2 = \sum_{i,j} \Gamma_{ij} C_{ij}$ where $C_{ij} = \||x_i - y_j\||^2$. Thus, **once a coupling $\Gamma$ is fixed and a quadratic kinetic energy is used, linear interpolations are not a modeling choice, but the unique minimum-energy trajectories consistent with that $\Gamma$**.
>
> 2. **Joint static–dynamic problem.** We introduce the energy $E_\alpha(\Gamma, X_{\bullet}) = (1-\alpha)K(X_{\bullet}) + \alpha S(\Gamma)$ where $S(\Gamma)$ is exactly the structure alignment cost defined in `Section 4.1`. We show
>
> $$
> \inf_{\Gamma, X_{\bullet}} E_\alpha(\Gamma,X_{\bullet})
> = \inf_{\Gamma\in\Pi(a,b)} \big[(1-\alpha)\langle \Gamma, C\rangle_F + \alpha S(\Gamma)\big]
> $$
>
> Therefore, **optimizing the static FGW objective is equivalent to minimizing, over all couplings and all absolutely continuous processes consistent with those couplings, a joint energy consisting of kinetic energy in expression space plus a CCI misalignment penalty**.
> The minimizers of this joint problem are precisely:
> - an FGW-optimal coupling $\Gamma^\ast$
> - its associated **linear interpolations** $X_t^{lin} = (1-t)X + tY, (X,Y) \sim \Pi_{\Gamma^\ast}$
>
> This gives a precise answer to why we use linear interpolations and “what problem is being solved”: IADOT finds a coupling and a collection of trajectories that jointly minimize “kinetic energy + structure cost” under the constraints induced by the FGW formulation.
>
>
>
> 3. **Connection to the learned velocity field (`App. F.3`).** Adopting linear interpolations (as motivated by the two points above), the ground-truth minimizer of the CFM objective is  $v^\ast(z,t) =  \mathbb{E}[Y-X \mid Z_t = z]$. We show in `App F.3` that its kinetic energy $K_{\mathrm{Eul}}(v^\star) := \int_0^1 \mathbb{E}_{Z_t}\big[\||v^\star(Z_t,t)\||^2\big]dt$ is **upper-bounded** by the feature cost $\langle \Gamma^\star, C\rangle_F$ of the optimal coupling $\Gamma^\star$.
>
> **To summarize:** Once we fix a coupling $\Gamma$ and work with quadratic kinetic energy, linear interpolations are not an arbitrary modeling choice: they are the only admissible paths that minimize kinetic energy between the endpoints. Hence, IADOT optimizes a well-defined joint objective
>
> $$
> \min_{\Gamma, X_{\bullet}} \ (1-\alpha)\\text{kinetic energy}(X_{\bullet}) + \alpha\\text{Structure cost}(\Gamma).
> $$
>
>
> Finally, CFM learns a velocity field whose kinetic energy is controlled by this objective. We hope that this clarifies “what problem is being solved.”
>
> ---
>
> ### 2. Why linear interpolations do not preclude interactions in IADOT
>
> **Linear interpolations don't cross in OT-CFM.** In OT-CFM and under mild regularity assumptions, the coupling comes from a Monge map $T = \nabla\phi$ where $\phi$ is convex, so trajectories $t \mapsto (1-t)x + tT(x)$ are non-crossing (the proof is given in [1]). This is why the velocity field can be interpreted as modeling independent particles.
>
> IADOT differs in two key aspects:
>
> 1. **General coupling.**  There is in general no Monge map $T$ for the FGW problem, and therefore the linear interpolations can cross.
>
> 2. **Velocity field is a mixture of bridges.**  The velocity field  $v^\star(z,t) = \mathbb{E}[Y-X \mid Z_t = z]$ is an average over all bridges whose straight line passes through $z$. Therefore this mixes drifts from multiple cell pairs, which implicitly models interactions.
>
> In practice, higher values of $\alpha$ are more likely to induce crossing paths. We show this *empirically (`App. E.7`)*: increasing $\alpha$ systematically increases the path-length ratio decreases, confirming that the learned paths are not straight OT geodesics.
>
> [1] Tong, Alexander, et al. "Simulation-free schrodinger bridges via score and flow matching."

---

> ### Author Response · Authors · 2025-11-25
> **Response (2/3)**
>
> ### 3. Why we do not model CCIs explicitly along the path
> There are two main reasons for this:
> 1. **CCIs are only defined at observed times.** Our CCI tensors are defined only for observed cells at discrete time points. Any “interaction along the path” would involve synthetic intermediate cells and synthetic CCIs that are not measured and cannot be uniquely inferred from snapshot data alone.
>
> 2. **Identifiability.** Allowing arbitrary time-dependent interaction forces (e.g. mean field interaction kernels) introduces many extra degrees of freedom. From snapshot data, many different force fields would generate essentially the same marginal distributions at observed times, so these forces would be non-identifiable. In contrast, encoding CCIs in the structure term $S(\Gamma)$ at the endpoints yields a well-posed regularization of the coupling and shapes the velocity field through $\Gamma^*$.
>
> For these reasons, our approach is to encode interactions in the endpoint structure term $S(\Gamma)$ and let this structural prior propagate through the coupling and the induced velocity field.
>
> ---
>
> ### 4. On a Benamou–Brenier-style theorem for GW/FGW
>
>
>
> To the best of our knowledge, a BB-style dynamic formulation whose value exactly matches the GW/FGW distance does not currently exist for GW/FGW, and extending the BB framework from classical OT to the non-linear GW/FGW setting is an open problem. This limitation is **not specific to IADOT**, it affects all methods based on GW/FGW (including Moscot [1]). While the original BB formulation for linear OT is simple to state, the underlying theorem is technically quite involved, and extending it to the GW/FGW case would be highly nontrivial. A rigorous development of such a formulation therefore falls outside the scope of this paper, although we view it as a very interesting direction for future work.
>
> [1] Klein, Dominik, et al. "Mapping cells through time and space with moscot." Nature 638.8052 (2025): 1065-1075.
>
>
> ---
>
> ## [P2] Static CCI
>
> A key point is that our “static CCI” assumption is not a biological claim. We do not posit that cell–cell communication is literally fixed across time. Rather, this plays the same conceptual role as the smoothness priors used in linear OT-based trajectory inference: methods built on the **principle of least action** in Wasserstein space assume that cell states follow cost-minimizing paths between snapshots, even though gene expression is of course dynamic. In IADOT, our assumption of slowly varying CCI is of the same nature, but uses a different choice of geometry (CCIs).
>
>
>  In ill-posed inverse problems (like trajectory inference from snapshots), the validity of a prior is determined by its utility. As shown in our experiments, this specific topological prior recovers trajectories that are more accurate (lower interpolation error) and biologically consistent (correct perturbation response) than standard OT.
>
>  We do not claim that it holds everywhere: we explicitly show in `Section 5.6` a regime where it breaks down (snapshots days apart) where the best performance is attained near $\alpha=0$, i.e., IADOT falls back to standard OT. Similarly, in standard OT, the principle of least action assumption is known to fail for large time gaps or abrupt transitions (for example, see [1]: *"We also observed that the predictive performance for CellOT drops when perturbations are too strong (the cell distributions before and after perturbations are very different)"*.
>
>
> Thus, we fully agree that dynamics inferred from destructive snapshots remain fundamentally underdetermined. Our goal is to provide a biologically interpretable, optional CCI prior whose influence is controlled by the data rather than imposed as a hard assumption, and we empirically showed on $7$ datasets that it outperforms $9$ baselines and improves performance across $4$ methods when used as a plug-in.
>
>
> [1] Bunne, Charlotte, et al. "Learning single-cell perturbation responses using neural optimal transport."

---

> ### Author Response · Authors · 2025-11-25
> **Response (3/3)**
>
> ## [P3] Novelty
>
> We fully understand that assessments of “novelty” can differ. Our goal with this work is less to introduce a new mathematical framework than to address what Bunne et al. [1] explicitly identify as an *open problem*: “the adaptive selection of robust cost functions remains a crucial area for future research” for OT-based cell trajectory inference.
> Our contribution is to pursue this in the underexplored direction of ligand–receptor–mediated cell–cell interactions, by (i) introducing a directed, typed, multi-channel CCI prior in a vector-valued FGW objective, (ii) adapting the solver to this structure, and (iii) demonstrating that the resulting coupling can be used as a plug-and-play component that consistently improves interpolation across CFM, Metric Flow Matching, and unbalanced OT baselines, with ablations showing that biologically meaningful CCIs (not generic structure) drive the gains.
>
> Even if one views the use of FGW and/or CFM themselves as methodologically incremental, we believe that systematically studying how biologically grounded interaction priors should enter OT-based trajectory inference (and showing that they can improve identifiability and downstream dynamics across multiple settings) addresses a problem that the community has explicitly flagged as both *important* and *unresolved*.
>
>
> [1] Bunne, Charlotte, et al. "Optimal transport for single-cell and spatial omics."

---

### Official Review · Reviewer_QVyK · 2025-10-28

**Soundness:** 2
**Presentation:** 2
**Contribution:** 2
**Rating:** 4
**Confidence:** 4

**Summary:**

This paper presents IADOT, a method for inferring cellular dynamics from scRNA-seq snapshots by incorporating cell-cell interaction (CCI) networks into the computation of OT couplings. The authors propose a multi-channel Fused Gromov-Wasserstein (FGW) objective to find a coupling that balances gene expression similarity and the preservation of CCI structure. This coupling is then used to train a continuous-time velocity field via Conditional Flow Matching (CFM). IADOT is benchmarked on synthetic data and multiple real scRNA-seq datasets, indicating that incorporating CCI information improves the accuracy of trajectory inference.

**Strengths:**

- Inferring trajectories from snapshot data is important in single cell biology.
- The paper moves beyond the common and limiting assumption of treating cells as independent particles in OT-based trajectory inference.
- The paper is well written in general, with comprehensive analysis under different experimental settings.

**Weaknesses:**

- From a technical perspective, this paper provides limited novelty in the OT formulation. **The application of GW/FGW-OT in single-cell trajectory inference is not new.** Apart from several works the authors already cited, moslin[1] also uses FGW-OT in the trajectory inference task. The difference lies in the specific biological priors, not the methodology itself.
- Directly using CFM to construct dynamic trajectories is also not novel. Here the dynamic paths are constructed using linear interpolation. However, if the population of cells is truly considered as an interacting system, then the trajectory of one cell will be influenced by others. Thus, the linear interpolation is questionable, as **it still considers each cell as an independent particle by enforcing straight lines.** Moreover, as the constraints of CCI patterns are only applied at endpoints, it is also questionable **whether these patterns can be preserved along the whole interpolated trajectories.**
- From a biological perspective, the CCI matrix is constructed from dissociated single-cell data, which removes the spatial context. In biological tissues, cells typically communicate within a local neighborhood. By ignoring spatial proximity, the method is likely to infer a large number of **false positive interactions** between cells that would never be in contact in vivo. This raises concerns about whether the CCI prior being used is biologically meaningful.
- From an empirical perspective, the improvement of incorporating CCI information in real world datasets is marginal, and seems to be **sensitive to different $\alpha$ across different datasets.** The authors should provide a clear guideline for selecting $\alpha$ on new datasets and report the computational cost required for this tuning process.
- The connection between results in Section 5.2 and Section 5.3 is unclear. Are 'V1 Light' in Figure 4 (which is not mentioned in Table 6) and 'V1 Cortex' in Table 2 the same dataset? If so, the results seem to be inconsistent, as $\alpha=1$ yields the best performance in Table 2, while it exhibits the worst performance in Figure 4. If not, I recommend keeping the benchmarking datasets the same throughout the paper.
- The authors should consider more recent flow-matching baselines, such as Metric FM[2].
- The definitions of different properties in Table 1 should be introduced to ensure a fair comparison of different methods.

[1] Lange, Marius, et al. "Mapping lineage-traced cells across time points with moslin." Genome Biology 25.1 (2024): 277.

[2] Kapusniak, Kacper, et al. "Metric flow matching for smooth interpolations on the data manifold." Advances in Neural Information Processing Systems 37 (2024): 135011-135042.

**Questions:**

Following the above weaknesses, the authors should make the following revisions before it can be considered for a favorable evaluation:
 - Justify the usage of linear interpolation to generate dynamic trajectories of interacting cells.
 - Justify the biological validity of the constructed CCI matrix, particularly the issue of potential false positives arising from the lack of spatial constraints.
 - Provide guidelines and computational cost for selecting $\alpha$.
 - Clarify the connection between Figure 4 and Table 2.
 - Incorporate more recent baselines.
 - Clarify the proposed properties in Table 1.


The reviewer wrote this review. LLM was only utilized to polish the review for grammatical accuracy and clarity.

---

> ### Author Response · Authors · 2025-11-24
> **Response (1/5)**
>
> Thank you for your constructive feedback and for acknowledging the importance of the problem tackled of our work as well as the limitations of existing approaches. The following sections address each point you raised, detailing how your feedback has led to improving the quality of our submission.
>
> ---
>
> # [P1] Novelty
>
> We appreciate the opportunity to clarify the contributions of $\texttt{IADOT}$. While we utilize the mathematical frameworks of Fused Gromov-Wasserstein (FGW) and Conditional Flow Matching (CFM), we argue that  that our contribution is not incremental. Our novelty lies in **(1) a domain-specific construction** of the structural prior that is distinct from generic graph regularization, **(2) a multi-channel optimization** scheme required to handle this specific structure, and **(3) a modular design** that solves the identifiability problem across various dynamic models.
>
> **1. Novelty lies in the prior construction**
>
> While we agree that "the application of GW/FGW-OT in single-cell trajectory inference is not new", we argue that our formulation is strictly specific to the biological domain and distinct from standard graph structures used in other FGW applications.
>
> * **Inferred, directed, typed structure:** Unlike methods that rely on **external** priors like physical coordinates or lineage trees (such as moslin [1]) or **undirected** geometric proximity (e.g. standard MOSCOT usage [2]) $\texttt{IADOT}$ infers a **directed** and **multi-channel** interaction tensor strictly from gene expression data. This allows trajectory inference in standard scRNA-seq data where no external structure is provided (e.g. unlike moslin which assumes access to lineage trees data).
> * **Biological specificity:**  Our sensitivity analysis confirms that the performance gains are driven by the specific biological semantics of the Ligand-Receptor (LR) interactions. As shown in `Section 5.5` and `Table 3`, replacing the curated LR catalog with random or shuffled interactions degrades performance. This proves $\texttt{IADOT}$ leverages specific signaling biology, not just generic regularization.
>
> **2. Technical differentiation from standard FGW methods**
>
> We emphasize that $\texttt{IADOT}$ is not a vanilla application of FGW. The biological complexity of cell-cell interactions necessitates specific algorithmic adaptations that differentiate it from existing baselines.
>
> * **Multi-channel solver:** Standard single-cell FGW implementations typically treat cells as nodes in a single-layer, undirected kNN graph. In contrast, to capture the complexity of signaling, $\texttt{IADOT}$ operates on **multi-typed** interaction tensors (where $G_{ij}$ is a vector in $\mathbb{R}^K$, not a scalar). This required us to implement a customized **multi-dimensional FGW solver** to handle vector-valued edge weights, which we detail in `Appendix D.4`.
> * **Empirical superiority:** This technical distinction translates to performance. We have performed an additional experiment comparing $\texttt{IADOT}$ with MOSCOT (without spatial data, hence which uses a kNN graph), with results presented in `Table 2`, and we see that $\texttt{IADOT}$ consistently outperforms MOSCOT in interpolation tasks across the different datasets, indicating that our directed, multi-channel prior provides signal that generic graph alignment misses.
>
> **3. Modularity of the dynamic objective**
>
> The choice to not introduce a new flow matching loss is a deliberate design choice to ensure **modularity**.
>
> * **Identifiability via coupling:** The core challenge in scRNA-seq trajectory inference is the lack of ground-truth alignment [3]. We posit that the "optimality" of the trajectory is critically influenced the boundary conditions (the coupling) using biological priors, rather than enforcing specific constraints on the flow field itself.
> * **IADOT is plug-and-play:** By decoupling the prior learning (the coupling) from the dynamics learning (the flow), $\texttt{IADOT}$ acts as a flexible plug-and-play prior. To demonstrate this point, we conducted additional experiments, showing in `Table 2` and `Appendix E.4` that $\texttt{IADOT}$ improves performance when combined with various state-of-the-art dynamics models, including MFM [4], UOT-FM [5], and SF2M [6].

---

> ### Author Response · Authors · 2025-11-24
> **Response (2/5)**
>
> **To summarize:**  we establish that $\texttt{IADOT}$ tackles a specific identifiability issue through a novel, domain-specific prior. **As noted in [3], "the adaptive selection of robust cost functions remains a crucial area for future research"** and our work answers this call by deriving transport costs from signaling biology rather than generic geometry.
>
> **Actions taken:**
> * **Clarified novelty:** We have revised `Section 3` to explicitly contrast our multi-channel, directed construction with standard undirected graph methods used in related work, as well as methods assuming external data (e.g. moslin).
> * **Extended baselines:** We conducted experiments with additional baselines; the manuscript now contains **9 baselines** and **4 $\texttt{IADOT}$ variants** (`Table 2`).
>
> **References**
>
> [1] Lange, Marius, et al. "Mapping lineage-traced cells across time points with moslin."
>
> [2] Klein, Dominik, et al. "Mapping cells through time and space with moscot."
>
> [3] Bunne, Charlotte, et al. "Optimal transport for single-cell and spatial omics."
>
> [4] Kapusniak, Kacper, et al. "Metric flow matching for smooth interpolations on the data manifold."
>
> [5] Eyring, Luca, et al. "Unbalancedness in neural monge maps improves unpaired domain translation."
>
> [6] Tong, Alexander, et al. "Simulation-free schrodinger bridges via score and flow matching."
>
> ---
>
> # [P2] Justification of linear interpolation in CFM
>
> We agree that biological systems are interacting and non-linear, and that treating cells as independent particles moving in straight lines is biologically implausible.
>
> However, we want to clarify that **the linear interpolation in Conditional Flow Matching (CFM) is only an auxiliary training signal, not the final inferred trajectory.** The resulting vector field becomes non-linear and interaction-dependent because it learns the global transport map dictated by the coupling. We justify the usage of linear interpolation with the following points:
>
> **1. Kinetic energy** As shown in `App F.2`, optimizing the FGW objective is equivalent to minimizing, over all couplings and all absolutely continuous processes consistent with those couplings, a joint energy consisting of kinetic energy in expression space plus a CCI misalignment penalty.
> The minimizers of this problem are precisely:
> - an FGW-optimal coupling $\Gamma^\ast$
> - its associated **linear interpolations** $X_t^{lin} = (1-t)X + tY, (X,Y) \sim \Pi_{\Gamma^\ast}$
>
> **2. The learned vector field is non-linear and interaction-dependent**
> While CFM uses linear paths *during training* (connecting source $X$ to target $Y$), the **learned vector field** $v_{\theta}(z, t)$ approximates the posterior expectation of these drifts over the entire coupling $\Pi$:
> $$v\_\theta(z, t) \approx \mathbb{E}_{(X,Y) \sim \Pi}[Y - X \mid Z_t = z]$$
>
> Because the expectation is taken with respect to the interaction-aware coupling $\Pi$, the learned field $v^*$ can average conflicting linear directions from the training data, leading to curved, non-linear trajectories.
>
> **Curvature analysis.** To empirically demonstrate that $\texttt{IADOT}$ learns non-linear interaction effects, we conducted an additional experiment measuring the average **path length ratio** (displacement divided by path length) of the inferred trajectories:
> $$S(z\_0,v_\theta) = \frac{\||z\_1 - z\_0\||_2}{\int_0^1 \||v\_{\theta}(z\_t, t)\||_2 \, dt}$$  where $z\_t = z\_0 + \int\_0^t v\_{\theta}(z\_t,t)dt$ for $t\in [0,1]$ and $z\_0$ denotes the initial point.
>
> A ratio of $1$ indicates a straight line, while values $<1$ indicate curvature.
>
> As shown in the table below, increasing $\alpha$ leads to significantly higher curvature (lower ratios), confirming that **incorporating interactions prevents the model from simply learning independent straight lines.**
>
> | $\alpha$ | Lung Tumour | Dendritic Stimulus | V1 Light |
> | :--- | :--- | :--- | :--- |
> | 0 | 0.974 ± 0.001 | 0.982 ± 0.002 | 0.954 ± 0.004 |
> | 0.5 | 0.952 ± 0.002 | 0.979 ± 0.003 | 0.907 ± 0.011 |
> | 1| 0.862 ± 0.011 | 0.969 ± 0.005 | 0.635 ± 0.009 |
>
> **3. Non-linear interpolation schemes**
> Finally, to address the concern that linear interpolation might be insufficient for specific manifolds, we highlight that $\texttt{IADOT}$ is modular: the CCI prior can be applied to **non-linear** interpolations in flow matching objectives. We evaluated $\texttt{IADOT}$ with **Metric Flow Matching (MFM)** [1] (geodesics on the data manifold).
> As shown in `Table 2` (the $\texttt{IADOT + MFM}$ variant), $\texttt{IADOT}$ consistently improves performance ($\alpha > 0$ is optimal) even when combined with these non-linear interpolations.
>
> **Actions taken:** (i) We have included the curvature analysis in a new `Appendix E.7`. (ii) We have conducted an additional experiment in `Table 2` with the $\texttt{IADOT + MFM}$ variant to allow for non linear interpolations.
>
> **Reference**
>
> [1] Kapusniak, Kacper, et al. "Metric flow matching for smooth interpolations on the data manifold."

---

> ### Author Response · Authors · 2025-11-24
> **Response (3/5)**
>
> # [P3] Biological validity of CCI without spatial context
>
>
> Thank you for your comment regarding the CCI and false positives. We agree that inferring CCIs from single-cell data is prone to false positives due to dropout and the lack of spatial context. We have addressed this through rigorous preprocessing and extensive sensitivity analyses in the appendix. We structure our response below to distinguish between **sensitivity to biological signal** (which is desired) and robustness to noise.
>
> **1. Sensitivity to structure (`Table 3`) demonstrates specificity.**
> In `Table 3` , we see a performance drop when using random LR catalogs or shuffled CCIs. We argue this is a necessary property of a structure-aware method. If $\texttt{IADOT}$ performed equally well with a shuffled CCI (which effectively simulates false positives) as it did with the original CCIs, it would imply the prior is uninformative. The performance gap confirms that $\texttt{IADOT}$ is successfully leveraging the specific biological signal in the true catalog. Importantly, while "Random LR" performs worse than the true catalog, it does not lead to catastrophic failure. This indicates that while false positives dilute the benefit of the prior, they do not break the underlying transport formulation.
>
> **2. Additional perturbation analysis  (`Appendix E.5`).**
>  We have performed an additional perturbation analysis in `Appendix E.5`.
> * **Setup:** For LR genes, we add zero-mean Gaussian noise to the gene expressions before applying the Hill transform and clip below by $0$, i.e. we define $\tilde{x}\_{cg} = \max(0,x\_{cg}+\epsilon\_{cg})$ where $\epsilon\_{cg} \sim \mathcal{N}(0,\sigma\_g^2)$.
> * **Results:** `Figure 13` demonstrates that as the noise scale increases, the interpolation error increases gradually across all three datasets. Again, this shows the specificity of our biological prior: no sensitivity with respect to noise would have meant that this prior is irrelevant. Furthermore, the performance is relatively robust to small level of noises for the Lung tumour and V1 Light datasets. Interestingly, for the V1 Light dataset, we see that small noise scaling improves the results, which we attribute to a small regularization / denoising effect. Adding a small amount of centered Gaussian noise before the Hill transform and clipping  pushes low-intensity ligand or receptor expressions to zero while leaving strongly expressed pairs essentially unchanged.
>
> **3. Mitigating false positives with our LR selection (`Appendix E.3`).**
> To proactively mitigate false positives before optimization, we employ a strict construction pipeline detailed in `Appendix D.6` to choose the LR catalog. Rather than using all co-expressions, we utilize perform consensus rank aggregation, and filter interactions by both significance ($p < 0.05$) and expression magnitude thresholds. To validate that this selection matters and to test robustness to "functional" false positives,we compared these optimized tissue-specific priors against a "generic" list of canonical long-range interactions in an additional experiment conducted in `Appendix E.3`. `Table 22` shows that using the generic list (which effectively acts as a catalog with high false positives relative to the specific tissue) yields higher Wasserstein distances than our LR catalog, showing the importance of a robust LR selection procedure to mitigate false positives.
>
> **4. IADOT does not enforce a hard constraint.**
> $\texttt{IADOT}$ incorporates the CCI prior in a soft way via the parameter $\alpha$ in the FGW objective. `Figure 4` validates this flexibility, displaying a U-shaped error curve where intermediate weights ($\alpha$≈0.5−0.7) yield optimal transport plans. By calibrating $\alpha$ on held-out timepoints, $\texttt{IADOT}$ automatically adapts to dataset-specific noise. For example, in the embryo dataset (`Section 5.6`) where the CCI prior is uninformative, the optimal $\alpha$ is $\approx 0$.
>
> **Actions taken:** (i) We conducted an additional perturbation analysis on the LR expressions in `Appendix E.5` where we add Gaussian noise to the LR expressions (ii) We compared our automatic LR selection procedure to a generic LR catalog in `Appendix E.3`.

---

> ### Author Response · Authors · 2025-11-24
> **Response (4/5)**
>
> # [P4] Selecting $\alpha$ and computational cost
>
>
> Thank you for the practical question regarding tuning $\alpha$.   Our recommended strategy for selecting $\alpha$ depends on the number of snapshots in the dataset:
>
> **For datasets with more than two timepoints (as in our manuscript):** we recommend the interpolation validation strategy detailed in `Section 5.2`, where an intermediate snapshot is held out and $\alpha$ is selected to minimize the Wasserstein distance between the predicted and empirical distributions.
>
> **For datasets with only 2 snapshots:** we recommend to select $\alpha$ based on biological heuristics, such as maximizing label transfer accuracy (e.g. selecting the coupling that preserves cell types the most). We also recommend downweighting the structural term ($\alpha \to 0$) when dealing with long temporal intervals (e.g., 6 days) where massive tissue remodeling makes the prior uninformative, as we illustrate in our analysis of the Embryo dataset in `Section 5.6`.
>
> **Computational cost.** To demonstrate the feasibility of the tuning process based on interpolation metrics, we report the wall-clock runtimes in `Table 19` and `Appendix D.10`. For example, on the Lung Tumor dataset, the OT solver requires approximately $211$ seconds. Consequently, a typical grid search over $10$ distinct $\alpha$ values requires approximately $35$ minutes on a standard hardware. Given that finding the coupling is only performed *once*, we consider this computational cost negligible relative to the practical value of improving trajectory inference.
>
> **Action taken:** We have added a detailed discussion on computational cost in `Appendix D.10`.
>
>
> ---
>
> # [P5] Clarification of dataset naming and result consistency
>
> We apologize for the inconsistent naming. We confirm that **'V1 Light' (Figure 4) and 'V1 Cortex' (Table 2) refer to the same dataset**; we have standardized the name to 'V1 Light' throughout the revised manuscript.
>
> Regarding the difference in optimal $\alpha$, this results from the mathematical distinction between the two evaluations:
>
> **Figure 4.** The results in Figure 4 are computed using the affine interpolant $\rho_t$ derived directly from the optimal coupling $\Gamma^*$, i.e. $\rho_{t} = \text{Law}((1-t)X + tY)$.
>
>
> **Table 2.** In Table 2, we evaluate the continuous flow $v_\theta$ learned via CFM. The CFM objective trains the network to regress the expected conditional drift, $v\_\theta(z, t) \approx \mathbb{E}\_{(X,Y) \sim \Pi}[Y - X \mid Z\_t = z]$. We then use $v\_\theta$ to obtain the pushforward of $\rho\_0$, denoted $\hat{\rho}\_t$, which is compared to the observed snapshot at time $t$. In practice, $\hat{\rho}\_t$ is not guaranteed to be equal to $\rho\_{t}$ for two reasons: 1) $v\_\theta$ is parameterized as a neural network and might not reach a global optimum because of optimization considerations 2) even if $v\_\theta$ was optimal, the pushforward $\hat{\rho}_t$ is computed by integrating the trajectories of the $n_0$ source cells. It is therefore strictly an empirical distribution of size $n_0$, while the affine interpolant $\rho_t$ represents the full distribution of the coupling (where one cell can split into multiple paths). This explains why the results in `Table 2` are not identical than in `Figure 4` for the V1 Light dataset.
>
> **Action taken:** Standardized the name to 'V1 Light' throughout the revised manuscript
>
>
> ---
>
> # [P6] Additional baselines
>
> Thank you for suggesting us to compare $\texttt{IADOT}$ with additional baselines. Following this, we have performed the following new experiments:
>
>
> **New baselines.** Following your recommendation, we have *significantly* expanded our evaluation in `Table 2` and `Section 5.3` to include baselines with:
> - Geometric priors: MIOFlow [1] and Metric Flow Matching (MFM) [2].
> - Unbalanced mass: VGFM [3] and UOT-FM [4].
> - Stochastic bridges: Simulation-Free Score Matching (SF2M) [5]
> - other OT-dynamics: MOSCOT [6].
>
> **New IADOT variants.** Furthermore, to demonstrate the modularity of $\texttt{IADOT}$, we extended MFM (`Appendix D.12`), UOT-FM (`Appendix D.11`), and SF2M (`Appendix D.13`) to incorporate the IADOT prior.

---

> ### Author Response · Authors · 2025-11-24
> **Response (5/5)**
>
> **Results.** As reported in `Table 2`:
> - $\texttt{IADOT+CFM}$ consistently outperforms the newly added baselines across datasets.
> - The $\texttt{IADOT}$ prior improves the other frameworks: $\texttt{IADOT+MFM}$, $\texttt{IADOT+UOT-FM}$, $\texttt{IADOT+SF2M}$ yield consistently lower errors than their vanilla counterparts (MFM, UOT-FM, SF2M), confirming that the interaction prior is orthogonal to the priors considered in these works.
>
> **Action taken:** We conducted experiments with additional baselines; the manuscript now contains **9 baselines** and **4 $\texttt{IADOT}$ variants** (`Table 2`).
>
> **References**
>
> [1] Huguet, Guillaume, et al. "Manifold interpolating optimal-transport flows for trajectory inference."
>
> [2] Kapusniak, Kacper, et al. "Metric flow matching for smooth interpolations on the data manifold."
>
> [3] Wang, Dongyi, et al. "Joint Velocity-Growth Flow Matching for Single-Cell Dynamics Modeling."
>
> [4] Eyring, Luca, et al. "Unbalancedness in neural monge maps improves unpaired domain translation."
>
> [5] Tong, Alexander, et al. "Simulation-free schrodinger bridges via score and flow matching."
>
> [6] Klein, Dominik, et al. "Mapping cells through time and space with moscot."
>
> ---
>
> # [P7] Clarification of `Table 1` properties
>
> We fully agree that explicit definitions are essential for ensuring a fair comparison across methods. We have now provided explicit and detailed definitions for all five properties listed in `Table 1` (Dynamic, Trajectories, In-silico Perturbation, Structure-Aware, and scRNA Data Sufficient) in `Appendix A` for transparent comparison.
>
> ---
>
> # [P8] Significance of the improvements
>
> We agree with the reviewer that the observed quantitative reductions in interpolation error on most real-world datasets are not huge. However, the true significance of $\texttt{IADOT}$ lies in consistency and biological plausibility, rather than massive error reduction.
>
> - **Consistency across methods:** While the magnitude of the gains is not massive, the positive effect is highly consistent across intermediate structural weight ($\alpha$≈0.5) and across diverse flow-matching techniques. This holds true for deterministic (CFM, MFM), unbalanced (UOT-FM), and stochastic (SF2M) models, as evidenced in our revised `Table 2`. This suggests the prior is a robust and generalizable signal. Furthermore, it is biologically grounded, as we construct the CCI tensors using LR expressions.
>
>
> - **Stricter data setting:** Unlike methods that rely on extensive auxiliary data (e.g., spatial coordinates or lineage trees), $\texttt{IADOT}$ operates effectively using standard scRNA-seq alone. In this stricter setting, we believe that extracting a consistent, reliable gain from an inferred biological prior represents a significant methodological contribution.
>
>
> ---
>
>
> ### Thank you for your feedback
>
>
> Thank you for your detailed and constructive feedback, which has greatly helped us to refine our manuscript. We believe the revisions, including the new baselines and their $\texttt{IADOT}$ variants, the clarification on biological validity *substantially improve* the clarity and rigor of our submission. We are confident that $\texttt{IADOT}$, by tackling the **crucial question of: "What constitutes a good cost function for inferring biological transport maps?"**, provides a practical step toward more biology-aligned methods in scRNA analysis and welcome continued dialogue.

---

### Official Review · Reviewer_SwiP · 2025-10-29

**Soundness:** 3
**Presentation:** 3
**Contribution:** 2
**Rating:** 6
**Confidence:** 3

**Summary:**

This paper introduces **IADOT (Interaction-Aware Dynamic Optimal Transport)**, a framework that incorporates cell–cell interaction (CCI) structure into optimal transport–based trajectory inference for single-cell RNA sequencing (scRNA-seq) data. The method extends Fused Gromov–Wasserstein optimal transport by integrating *directed, typed ligand–receptor networks* as a structural prior, encouraging couplings that preserve biologically meaningful communication patterns across timepoints. Using these structure-aware couplings, the authors then learn continuous-time dynamics via Conditional Flow Matching (CFM).

Empirically, IADOT is evaluated on synthetic and real scRNA-seq datasets across different tissues, demonstrating that incorporating CCIs improves both cross-snapshot alignment and temporal interpolation compared to feature-only OT-CFM. The authors also perform in-silico perturbation analyses, showing biologically consistent trajectory shifts under ligand–receptor catalog edits. Overall, results support that interaction-aware priors can refine dynamic modelling of cellular systems.

**Strengths:**

1. **Well-motivated problem** - The paper tackles a significant and well-motivated problem. Standard OT-based trajectory inference methods often treat cells as independent data points, ignoring the rich biological context of cell-cell communication, which is a known driver of cellular dynamics.
2. **Clear validation of hypothesis** - The central hypothesis, that incorporating a CCI-based inductive bias can improve alignment over lack of this prior, is clearly validated.

**Weaknesses:**

1. **Limited technical novelty** - The core components: Fused Gromov-Wasserstein (FGW) and Conditional Flow Matching (CFM) are existing methods. The main contribution is the application of FGW to use directed, multi-typed LR interaction tensors*as the structural prior. While effective, this is a relatively straightforward extension of prior work.
2. **Omission of unbalanced optimal transport** - The paper's formulation relies on standard, balanced OT, which assumes conserved mass and does not account for cell proliferation or death. The authors acknowledge this limitation in the discussion, but given the incremental nature of the technical contribution, incorporating an unbalanced OT (UOT) formulation (e.g., as in UOT-FM [1] or VGFM [4]) seems within scope and would have significantly strengthened the paper.
3. **Incomplete positioning and outdated baselines** -  The paper's related work misses several relevant, recent methods that also incorporate various inductive biases into OT/flow-based trajectory models. For instance, the authors do not compare against methods that model geometric biases (MIOFlow [2], Metric FM [3]) or explicitly model cellular growth (VGFM [4]). The chosen baselines, TrajectoryNet and DSB, are now outdated given newer OT-based dynamic models that incorporate unbalanced or geometric regularization. A paper would benefit from positioning and comparison to more recent and competitive methods [1, 2, 3, 4] to observe the impact of different priors.

[1] UOT-FM (Eyring et al., 2024)
[2] MIOFlow (Huguet et al., 2022)
[3] Metric FM (Kapusniak et al., 2024)
[4] VGFM (Wang et al., 2025)

**Questions:**

1.  Given the rapid development in flow-based generative models for single-cell dynamics, why were more recent and relevant baselines (e.g., UOT-FM [1], MIOFlow [2], Metric FM [3], VGFM [4]) omitted in favour of older methods like TrajectoryNet and DSB?
2.  The authors correctly identify that the balanced OT formulation is a limitation for modeling systems with proliferation and death. Could the authors comment on the technical difficulty of extending IADOT to an unbalanced setting (e.g., using an unbalanced FGW objective)?

---

> ### Author Response · Authors · 2025-11-24
> **Response (1/3)**
>
> Thank you for your thoughtful analysis and positive evaluation of our work which recognises the biological motivation and clarity of our empirical validation. In what follows, we provide answers to each of the points you raised, which have helped us significantly improve the quality of our work.
>
> ---
>
> # [P1] Novelty and technical contribution
>
> We appreciate the opportunity to clarify the contributions of $\texttt{IADOT}$. While we utilize the mathematical frameworks of Fused Gromov-Wasserstein (FGW) and Conditional Flow Matching (CFM), we argue that  that our contributions is not a straightforward extension of prior work. Our novelty lies in **(1) a domain-specific construction** of the structural prior that is distinct from generic graph regularization, **(2) a multi-channel optimization** scheme required to handle this specific structure, and **(3) a modular design** that solves the identifiability problem across various dynamic models.
>
> **1. Novelty lies in the prior construction**
>
> While we agree that "Fused Gromov-Wasserstein (FGW) and Conditional Flow Matching (CFM) are existing methods", we argue that our formulation is strictly specific to the biological domain and distinct from standard graph structures used in other FGW applications.
>
> * **Inferred, directed, typed structure:** Unlike methods that rely on **external** priors like physical coordinates or lineage trees (such as moslin [1]) or **undirected** geometric proximity (e.g. standard MOSCOT usage [2]) $\texttt{IADOT}$ infers a **directed** and **multi-channel** interaction tensor strictly from gene expression data. This allows trajectory inference in standard scRNA-seq data where no external structure is provided (e.g. unlike moslin which assumes access to lineage trees data).
> * **Biological specificity:**  Our sensitivity analysis confirms that the performance gains are driven by the specific biological semantics of the Ligand-Receptor (LR) interactions. As shown in `Section 5.5` and `Table 3`, replacing the curated LR catalog with random or shuffled interactions degrades performance. This proves $\texttt{IADOT}$ leverages specific signaling biology, not just generic regularization.
>
> **2. Technical differentiation from standard FGW methods**
>
> We emphasize that $\texttt{IADOT}$ is not a vanilla application of FGW. The biological complexity of cell-cell interactions necessitates specific algorithmic adaptations that differentiate it from existing baselines.
>
> * **Multi-channel solver:** Standard single-cell FGW implementations typically treat cells as nodes in a single-layer, undirected kNN graph. In contrast, to capture the complexity of signaling, $\texttt{IADOT}$ operates on **multi-typed** interaction tensors (where $G_{ij}$ is a vector in $\mathbb{R}^K$, not a scalar). This required us to implement a customized **multi-dimensional FGW solver** to handle vector-valued edge weights, which we detail in `Appendix D.4`.
> * **Empirical superiority:** This technical distinction translates to performance. We have performed an additional experiment comparing $\texttt{IADOT}$ with MOSCOT (without spatial data, hence which uses a kNN graph), with results presented in `Table 2`, and we see that $\texttt{IADOT}$ consistently outperforms MOSCOT in interpolation tasks across the different datasets, indicating that our directed, multi-channel prior provides signal that generic graph alignment misses.
>
> **3. Modularity of the dynamic objective**
>
> The choice to not introduce a new flow matching loss is a deliberate design choice to ensure **modularity**.
>
> * **Identifiability via coupling:** The core challenge in scRNA-seq trajectory inference is the lack of ground-truth alignment [3]. We posit that the "optimality" of the trajectory is critically influenced the boundary conditions (the coupling) using biological priors, rather than enforcing specific constraints on the flow field itself.
> * **IADOT is plug-and-play:** By decoupling the prior learning (the coupling) from the dynamics learning (the flow), $\texttt{IADOT}$ acts as a flexible plug-and-play prior. To demonstrate this point, we conducted additional experiments, showing in `Table 2` and `Appendix E.4` that $\texttt{IADOT}$ improves performance when combined with various state-of-the-art dynamics models, including MFM [4], UOT-FM [5] (see our response **[P2]** for more details), and SF2M [6].

---

> ### Author Response · Authors · 2025-11-24
> **Response (2/3)**
>
> **To summarize:**  we establish that $\texttt{IADOT}$ tackles a specific identifiability issue through a novel, domain-specific prior.  **As noted in [3], "the adaptive selection of robust cost functions remains a crucial area for future research"** and our work answers this call by deriving transport costs from signaling biology rather than generic geometry.
>
>
> **Actions taken:**
> * **Clarified novelty:** We have revised `Section 3` to explicitly contrast our multi-channel, directed construction with standard undirected graph methods used in related work, as well as methods assuming external data.
> * **Extended baselines:** We conducted experiments with additional baselines; the manuscript now contains **9 baselines** and **4 $\texttt{IADOT}$ variants** (`Table 2`). See our response **[P3]** for more details.
>
> **References**
>
> [1] Lange, Marius, et al. "Mapping lineage-traced cells across time points with moslin."
>
> [2] Klein, Dominik, et al. "Mapping cells through time and space with moscot."
>
> [3] Bunne, Charlotte, et al. "Optimal transport for single-cell and spatial omics."
>
> [4] Kapusniak, Kacper, et al. "Metric flow matching for smooth interpolations on the data manifold."
>
> [5] Eyring, Luca, et al. "Unbalancedness in neural monge maps improves unpaired domain translation."
>
> [6] Tong, Alexander, et al. "Simulation-free schrodinger bridges via score and flow matching."
>
>
> ---
>
> # [P2] Unbalanced optimal transport
> We prioritized the balanced setting in our original manuscript to rigorously isolate the specific gain provided by our interaction-aware prior, avoiding potential confounding effects. However, we agree that an unbalanced formulation is important to capture realistic dynamics.
>
>
> **Additional experiments.** Following your suggestion, we conducted additional experiments presented in our revised manuscript:
> - we compare the balanced $\texttt{IADOT+CFM}$ with VGFM [1] and UOT-FM [2], two baselines that assume unbalancedness of mass.
> - we extend $\texttt{IADOT}$ to the unbalanced setting, with details provided in `Appendix D.11`. We denote this variant as $\texttt{IADOT+UOT-FM}$.
>
>
> **Results.** We report the results in `Table 2`, which highlight three key findings. First, $\texttt{IADOT+CFM}$ outperforms VGFM on all the datasets, and UOT-FM on $2$ datasets out of $3$, in continuous-time trajectory inference. Second, we demonstrate that $\texttt{IADOT}$ serves as an effective modular prior: specifically, $\texttt{IADOT+UOT-FM}$ yields lower transport errors than the feature-only UOT-FM baseline. Finally, these gains are explained by the underlying cross-snapshot interpolation results in `Figure 12`, where increasing the structure weight $\alpha$ leads to better alignments (exhibiting either a U-shaped curve or a monotonic improvement).
>
> **Action taken:** We have formulated and conducted additional experiments with an unbalanced extension of $\texttt{IADOT}$, see `Appendix D.11` for details.
>
> **References**
>
> [1] Wang, Dongyi, et al. "Joint Velocity-Growth Flow Matching for Single-Cell Dynamics Modeling."
>
> [2] Eyring, Luca, et al. "Unbalancedness in neural monge maps improves unpaired domain translation."
>
> ---
>
> # [P3] Baseline comparisons
> Thank you for highlighting relevant recent methods. We agree that comparing against varied inductive biases strengthens the positioning of our work.
>
> **New baselines.** Following your recommendation, we have *significantly* expanded our evaluation in `Table 2` and `Section 5.3` to include baselines with:
> - Geometric priors: MIOFlow [1] and Metric Flow Matching (MFM) [2].
> - Unbalanced mass: VGFM [3] and UOT-FM [4] (see **[P2]** for details).
> - Stochastic bridges: Simulation-Free Score Matching (SF2M) [5]
> - other OT-dynamics: MOSCOT [6].
>
> **New IADOT extensions.** Furthermore, to demonstrate the modularity of IADOT, we extended MFM (`Appendix D.12`) and SF2M (`Appendix D.13`) to incorporate the $\texttt{IADOT}$ prior, complementing the extension of $\texttt{IADOT}$ to the unbalanced setting discussed in **[P2]**.

---

> ### Author Response · Authors · 2025-11-24
> **Response (3/3)**
>
> **Results.** As reported in `Table 2`:
> - $\texttt{IADOT+CFM}$ consistently outperforms the newly added baselines across datasets.
> - The $\texttt{IADOT}$ prior improves the other frameworks: $\texttt{IADOT+MFM}$, $\texttt{IADOT+UOT-FM}$, $\texttt{IADOT+SF2M}$ yield consistently lower errors than their vanilla counterparts (MFM, UOT-FM, SF2M), confirming that the interaction prior is orthogonal to the priors considered in these works.
>
> **Action taken:** We conducted experiments with additional baselines; the manuscript now contains **9 baselines** and **4 $\texttt{IADOT}$ variants**.
>
> **References**
>
> [1] Huguet, Guillaume, et al. "Manifold interpolating optimal-transport flows for trajectory inference."
>
> [2] Kapusniak, Kacper, et al. "Metric flow matching for smooth interpolations on the data manifold."
>
> [3] Wang, Dongyi, et al. "Joint Velocity-Growth Flow Matching for Single-Cell Dynamics Modeling."
>
> [4] Eyring, Luca, et al. "Unbalancedness in neural monge maps improves unpaired domain translation."
>
> [5] Tong, Alexander, et al. "Simulation-free schrodinger bridges via score and flow matching."
>
> [6] Klein, Dominik, et al. "Mapping cells through time and space with moscot."
>
> ---
>
> ### Thank you for your feedback
>
> Thank you for your detailed and constructive feedback. We believe the extensive revisions undertaken, including the integration of the additional baselines and the empirical validation of our CCI prior in the unbalanced setting, have significantly strengthened the paper's technical rigor and positioning. We are confident that $\texttt{IADOT}$, by tackling the **crucial question of: "What constitutes a good cost function for inferring biological transport maps?"**, provides a practical step toward more biology-aligned methods in scRNA analysis and welcome continued dialogue.

---

> > ### Comment · Reviewer_SwiP · 2025-11-27
> >
> > Thank you for the detailed response and the novelty clarifications. While I appreciate the new baselines, some results appear suspiciously high (e.g., MIOFlow) or show no improvement over CFM. Could you explain the intuition behind these results in this application? It would also be helpful if you could clarify the hyperparameter selection process for these methods.

---

> ### Author Response · Authors · 2025-12-02
>
> Thank you for your continued engagement and for acknowledging the clarification of our novelty. We answer below your follow-up questions regarding hyperparameter selection and baseline performance intuition.
>
> **1. Hyperparameter selection.** To ensure a rigorous and fair comparison, we prioritized architectural consistency across all methods. As detailed in `Appendix D.8` and `Tables 11–16`, we implemented all baselines using their official code releases.
>
> For baselines augmented with $\texttt{IADOT}$ (i.e. UOT-FM, MFM, SF2M, and OT-CFM), we utilized a consistent network architecture (3 hidden layers, 64 units), the exact same one used in [1].
>
> In our rebuttal, MIOFlow and VGFM  used default settings (2 layers, 128 units). Prompted by your comment, we have re-run both MIOFlow and VGFM using the standardized architecture (3 layers, 64 units).
>
> **Results.** This change improved MIOFlow’s stability on the smaller "Stimulus" datasets (reducing catastrophic failure from 3 datasets to 1), but it barely changes the results on the other datasets (in `Table 2`). Conversely, VGFM saw a slight performance decrease, likely due to reduced capacity.
>
> **Action taken:** We have updated the results in `Table 2` and`Table 20` to reflect these standardized runs. We believe this provides the fairest possible comparison.
>
> **2. Intuition on baseline performance.** You noted that OT-CFM performs strongly and that MIOFlow occasionally yields high error rates. We attribute these results to the fundamental differences in how these models approach trajectory learning:
>
> **Stability of CFM vs Neural ODEs**: Methods like MIOFlow and TrajectoryNet are based on Neural ODEs, which require integrating a vector field during training. This is known to suffer from numerical instability and be hard to optimize. The "suspiciously high" error on the smaller Stimulus datasets (`Table 20`) aligns with the known tendency of Neural ODEs to overfit or fail to converge when data is scarce or the manifold is disjoint. On the contrary, CFM is much more stable as it optimizes a simulation-free regression objective, leading to the robust performance and lower variance observed in our results.
>
> **Suitability of inductive biases (MFM, SF2M, UOT-FM)**: More complex methods impose specific geometric or stochastic priors that are *not universally beneficial*. For instance, MFM relies on learning a Riemannian metric; on sparse scRNA-seq data, this task is often ill-posed, causing the model to enforce distorted paths where a simple Euclidean interpolation (OT-CFM) would be safer. Similarly, SF2M assumes dynamics are diffusive (Schrödinger Bridges), which can introduce unnecessary variance in datasets driven by strong signaling. The success of $\texttt{IADOT}$ confirms that the most effective inductive bias for these datasets is not a generic geometric constraint, but a domain-specific biological prior (cell-cell communication).
>
>
> **References.**
>
> [1] Tong, Alexander, et al. "Improving and generalizing flow-based generative models with minibatch optimal transport." Transactions on Machine Learning Research (2024).

---

### Official Review · Reviewer_W1vz · 2025-11-03

**Soundness:** 2
**Presentation:** 3
**Contribution:** 3
**Rating:** 4
**Confidence:** 4

**Summary:**

The paper proposes IADOT, an interaction-aware optimal transport (OT) coupling that augments the Fused Gromov–Wasserstein (FGW) objective with cell–cell interaction (CCI) priors derived from ligand–receptor (LR) expression. The resulting coupling is used within a conditional flow matching (CFM) algorithm to learn continuous-time single-cell dynamics. Experiments span synthetic data and several scRNA-seq datasets.

**Strengths:**

* **Motivation**: The problem of solving trajectory inference in single-cell data and learning evolving dynamics in cells is strongly motivated in introduction and through main text. Authors highlight need for this line of work and challenges such as noise, sparsity, ill-one-to-one mappings due to unbalanced distributions and beyond.
* **Quality**: The need for a structure-aware interaction coupling is argued clearly, and the FGW extension is conceptually coherent.
* **Interpretability**: IADOT uses real-world biological priors such as ligand-receptor expressions to construct its CCI network

**Weaknesses:**

* **Baselines**: The authors propose formulating trajectory inference problem as learning dynamics in interacting subsystems. This is very similar to the work done in [1] and [2] where where similar setting of cell-cell interactions is considered, as well as some more recent work on modeling single-cell dynamics such as [3]. These should at least be discussed in Related Work and, ideally, included as empirical baselines (*see references in the questions section*).
* **Quality of LR expressions**: Quality of coupling depends on the quality of fed LR expressions. I would suggest authors to perform sensitivity analysis to test this relationship.

**Questions:**

* What is the computational cost of IADOT vs. OT-based coupling and other baselines?
* In section 5.6 authors demonstrate that CCI persistence assumption fails to outperform OT-based methods due to rapidly changing development. In which cases is IADOT applicable? It would be good to provide analysis on whether [1] and [2] have similar limitations when modeling embryo data
* Have you tried applying IADOT in stochastic setting by constructing a stochastic bridge given the noise levels in single-cell data?
* Have you tried combining IADOT with other FM paradigms such as [3] instead of optimal transport coupling?

**References**

[1] Atanackovic, Lazar, et al. "Meta flow matching: Integrating vector fields on the wasserstein manifold." arXiv preprint arXiv:2408.14608 (2024).

[2] Sakalyan, Kristiyan, et al. “Modeling Microenvironment Trajectories on Spatial Transcriptomics with NicheFlow.” The Thirty-Ninth Annual Conference on Neural Information Processing Systems (2025)

[3] Kapusniak, Kacper, et al. "Metric flow matching for smooth interpolations on the data manifold." Advances in Neural Information Processing Systems 37 (2024): 135011-135042.

---

> ### Author Response · Authors · 2025-11-24
> **Response (1/4)**
>
> Thank you for your constructive feedback and for acknowledging the strong motivation and conceptual coherence of our work. In what follows, we provide answers to each of the points you raised, which have greatly helped us refine our manuscript.
>
> ___
>
> # [P1] Relation to baselines
>
> Thank you for pointing us to [1–3]. These works are indeed related in that they also model the evolution of single-cell populations and, in the case of [1, 2], reason about interacting cells. We have revised the Related Work section to position $\texttt{IADOT}$ with respect to them and clarify how their assumptions differ from ours.
>
> **Meta Flow Matching [1].**  Meta Flow Matching learns a vector field that is *amortized over a family of source–target distributions* by encoding each initial population with a graph neural network and training on $N$ pairs $\\{(p_0^{(i)}, p_1^{(i)}) \\}_{i=1}^{N}$. Hence this work assumes that *multiple* datasets are available for training. In contrast, $\texttt{IADOT}$ only assumes access to a *single* time-resolved scRNA-seq dataset. In this setting tackled by $\texttt{IADOT}$, Meta Flow Matching effectively reduces to a standard conditional flow matching (CFM) pipeline, which we already include in `Table 2`.
>
> **NicheFlow [2].**  NicheFlow is designed for *spatial transcriptomics*, assuming acces to data comprising both spatial coordinates and gene expression. Thus, NicheFlow *cannot* be applied to our scRNA-seq datasets. $\texttt{IADOT}$, in contrast, does *not* require access to external data such as spatial coordinates or histology images.
>
> **Metric Flow Matching [3].**  Metric Flow Matching augments CFM by learning geodesic interpolants under a data-induced Riemannian metric, enforcing that trajectories remain near the data manifold. This geometric prior is *orthogonal* to $\texttt{IADOT}$’s contribution, which constructs a *coupling* between snapshots via a cell-cell interaction-based Fused Gromov–Wasserstein objective. Following your comment, we have therefore:
> - **Included Metric Flow Matching as a baseline** in our trajectory inference experiments, see `Table 2`for the results
> - **Plugged $\texttt{IADOT}$ into MFM, an extension we denote as $\texttt{IADOT + MFM}$**: we detail in `Appendix D.12` how we achieve this in practice (keeping the CCI-derived coupling $\Gamma^\star$ and replacing the Euclidean CFM loss with the MFM objective). We report results in `Table 2` and in `Fig 11`.
>
> **Results.** In `Fig 11`, we see that the best interpolation for  $\texttt{IADOT + MFM}$ (derived from the couplings) consistently occurs at a non-zero structure weight $\alpha^* > 0$.
> In `Table 2`, we find that the velocity field obtained with $\texttt{IADOT+MFM}$ yields interpolation errors almost always lower than vanilla MFM across datasets. These findings indicate that the CCI prior is **complementary** to the geometry-aware flow-matching objective introduced by MFM and leads to better trajectory inference.
>
> **Other extensions of IADOT.** We further demonstrate the flexibility of $\texttt{IADOT}$ by considering and implementing other extensions: the unbalanced setting (with UOT-FM [4], leading to $\texttt{IADOT + UOT-FM}$) and a setting with stochastic bridges (with SF2M [5], leading to $\texttt{IADOT + SF2M}$, see our response **[P5]** below). The results for these additional extensions can be found in `Table 2`, showing that **our biological prior consistently improves trajectory inference**.
>
> **Actions taken.**  (i) Discussed [1–3] and their differences with $\texttt{IADOT}$ in `Section 3` (Related Work).   (ii) Conducted additional experiments with MFM and the $\texttt{IADOT+MFM}$ variant (`Table 2`, `Fig 11`). (iii) Conducted additional experiments with other baselines and $\texttt{IADOT}$ extensions (`Table 2`). Our manuscript now includes **$9$ baselines** and **$4$ IADOT variants**.
>
>
> **References**
>
> [1] Atanackovic, Lazar, et al. "Meta Flow Matching: Integrating Vector Fields on the Wasserstein Manifold."
>
> [2] Sakalyan, Kristiyan, et al. "Modeling microenvironment trajectories on spatial transcriptomics with nicheflow."
>
> [3] Kapusniak, Kacper, et al. "Metric flow matching for smooth interpolations on the data manifold."
>
> [4] Eyring, Luca, et al. "Unbalancedness in neural monge maps improves unpaired domain translation."
>
> [5] Tong, Alexander, et al. "Simulation-free schrodinger bridges via score and flow matching."

---

> ### Author Response · Authors · 2025-11-24
> **Response (2/4)**
>
> # [P2] Quality of LR expressions
> We appreciate your comment on the impact of the quality of the ligand-receptor (LR) expressions on the learnt couplings. We agree that the quality of LR information is important, since the cell-cell interaction (CCI) tensors depend on LR expression.
>
> **Perturbations to the CCIs.** Our current submission already includes a sensitivity analysis in `Sec. 5.5`, `Table 3`, where we vary the LR-based construction and re-evaluate the couplings. Specifically, we compare the default LR setup to variants which perturb the CCI: 1) random permutation of the entries, 2) random LR catalog. As reported in`Table 3`, both perturbations result in significantly higher interpolation errors. This confirms that our constructed CCI tensors capture a biologically meaningful prior. Otherwise, random or permuted inputs would have yielded comparable performance.
>
> **Corrupting the LR expressions.** Following your suggestion, we examined measurement noise in LR expression by conducting an additional experiment described in `Appendix E.5`. Concretely, for LR genes, we add zero-mean Gaussian noise to the gene expressions before applying the Hill transform and clip below by $0$, i.e. we define $\tilde{x}\_{cg} = \max(0,x\_{cg}+\epsilon\_{cg})$ where $\epsilon\_{cg} \sim \mathcal{N}(0,\sigma\_g^2)$. As shown in `Fig 13`, interpolation error increases as $\sigma\_g$ increases. This non-zero sensitivity is expected and desirable: if the CCI prior was irrelevant, corrupting the LR expressions would leave the interpolation error unchanged. Furthermore, the performance is relatively robust to small level of noises for the Lung tumour and V1 Light datasets. Interestingly, for the V1 Light dataset, we see that small noise scaling improves the results, which we attribute to a small regularization / denoising effect. Adding a small amount of centered Gaussian noise before the Hill transform and clipping  pushes low-intensity ligand or receptor expressions to zero while leaving strongly expressed pairs essentially unchanged.
>
> **Changing the LR catalog.** We also conducted an additional experiment where we use a set of generic cytokines to construct the CCI tensors, instead of our LR selection procedure. The results, presented in `Appendix E.3`and `Table 22`, demonstrate that our LR selection procedure imposes more informative constraints on the transport map than generic diffusive signaling.
>
> **Actions taken:** (i) Conducted an additional sensitivity analysis on the CCIs in `Appendix E.5` (ii) Compared our automatic LR selection procedure with a specific choice of LR catalog in `Appendix E.3`.
>
> ---
>
> # [P3] Computational cost
>
>  Thank you for your question regarding the computation cost of $\texttt{IADOT}$. We have accordingly revised the manuscript with a *detailed discussion* about the computational cost associated with $\texttt{IADOT}$ in `Appendix D.10`. The cost consists of:
>
> - **the OT coupling step:** $\texttt{IADOT}$ uses a conditional-gradient solver, extending it to multi–LR-pair CCIs keeps the same leading-order complexity as standard FGW optimization on $n_0$ and $n_1$ cells, up to a small constant factor linear in the number $K$ of LR pairs. For the modest $K$ used in our experiments, the overhead compared to feature-only OT is limited.
>
> - **the velocity field regression:** The CFM stage (architecture, optimizer, and schedule) is similar to the OT-CFM baseline, only the supervision coupling $\Gamma^\star$ changes. Thus, the cost of learning the continuous dynamics is essentially the same as for the FM-based methods.
>
> **Wall-clock runtimes.** In our experiments run on standard hardware, the OT step involves a reasonable computational cost on real-world datasets (e.g. less than $212$ seconds for the Lung tumour dataset), see `Table 19`.
> While this is reasonable considering that the OT step is only performed once, we propose several ways to reduce it when applying $\texttt{IADOT}$ to larger-scale datasets: i) use entropy-regularized OT objectives to accelerate convergence [1] ii) solve the OT problem at the minibatch level [2] instead of solving it at the dataset level iii) Use metacells to reduce the effective sample size, as explained in `Appendix D.3` and evaluated in `Section 5.5`.
>
>
> **Action taken:** We added a detailed discussion about the computational cost in `Appendix D.10`.
>
>
> **References**
>
> [1] Peyré, Gabriel, and Marco Cuturi. "Computational optimal transport: With applications to data science."
>
> [2] Tong, Alexander, et al. "Improving and generalizing flow-based generative models with minibatch optimal transport."

---

> ### Author Response · Authors · 2025-11-24
> **Response (3/4)**
>
> # [P4] Applicability of IADOT
>
>
>
> We clarify below (i) in which regimes $\texttt{IADOT}$ is most applicable, (ii) how to interpret the embryo experiment in `Section 5.6`, and (iii) how these limitations relate to [1] and [2].
>
> **When is IADOT applicable?** $\texttt{IADOT}$ introduces a *prior* that favors transport plans preserving ligand–receptor–derived CCI structure between adjacent snapshots. In practice, we find $\texttt{IADOT}$ particularly useful when:
>
> - **Time gaps are short-to-moderate** (hours) and **cell-type composition overlaps** substantially across snapshots (e.g., tumour progression, immune stimulation, cortex activation datasets in `Section 5.2` and `Section 5.3`).
> - In these settings, choosing some **$\alpha > 0$** (non-zero structure weight) consistently improves both (a) interpolation error and (b) downstream dynamics relative to feature-only OT, as shown in `Fig. 4` and `Table 2`.
>
> Thus, $\texttt{IADOT}$ is intended for regimes where CCI structure provides extra disambiguating signal on top of expression geometry, and our experiments show that this is common across several realistic scRNA-seq time courses.
>
> **Rapidly remodeling systems**
>
> The human embryoid body dataset is deliberately chosen as a *counter-example* where the CCI persistence assumption is weak:
>
> - Snapshots are **days apart** and the system undergoes **strong remodeling**, with large changes in tissue composition and function.
> - In this case, the ligand–receptor–based CCI patterns at $D\_{0}$ and $D\_{12}$ are no longer transferable. Consequently, the interpolation curve in `Fig. 6`is essentially flat with respect to $\alpha$, and performance is best at $\alpha \approx 0$, where $\texttt{IADOT}$ effectively reduces to standard OT+CFM.
>
>
> More generally, because we designed the CCI prior to act as **soft constraint** via the tunable parameter $\alpha$, the optimal $\alpha^\ast$, which is obtained with our interpolation-based selection protocol, is naturally **dataset-dependent**. We recover $\alpha^\ast > 0$ when the CCI prior is helpful and $\alpha\^\ast \approx 0$ when it is not. Therefore, **$\texttt{IADOT}$ is always applicable**; in the worst case, it behaves like standard OT-based methods, and in the best case, it exploits persistent CCI structure to improve alignment and dynamics.
>
> **Do Meta Flow Matching and NicheFlow have similar limitations on embryo-like data?**
>
> All three methods (ours, [1], [2]) operate on destructive snapshots and rely on a smooth population-level evolution. None can fully resolve identifiability issues when timepoints are extremely far apart and intermediate states are missing.
>
> - **Meta Flow Matching (Meta FM).** It does **not** consider an explicit ligand–receptor CCI persistence prior and only uses standard feature-based OT/FM at the population level. Therefore:
>   - Meta FM would *not* suffer specifically from a mismatch of LR-based CCI structure as $\texttt{IADOT}$ does when $\alpha$ is large on the embryoid body data ( Meta FM is equivalent to $\texttt{IADOT}$ with $\alpha = 0$ in this experiment).
>   - However, it **still shares the fundamental limitation** that, with very sparse time sampling and drastic changes in cell composition (as in the embryo dataset), many transport maps are compatible with the observed marginals and hence does not overcome the identifiability issue.
>
> - **NicheFlow.** NicheFlow's structural prior is geometric: it assumes that local niches and spatial architecture evolve smoothly across consecutive slides. Thus:
>   - Like $\texttt{IADOT}$, NicheFlow **relies on persistence of structure across time**, but the structure is spatial microenvironment geometry rather than typed LR CCIs.
>   - On embryo data with large temporal gaps and dramatic tissue reorganization, where even local neighborhoods change qualitatively, the OT step in NicheFlow would also face an underconstrained alignment problem.
>
> In summary, **the negative result on the embryoid** illustrates a general limitation of snapshot-based approaches when the process is highly non-stationary and sparsely sampled. $\texttt{IADOT}$ makes this explicit via the $\alpha$-sweep in `Section 5.6` and, crucially, provides a simple mechanism to turn off the structural prior when it ceases to be informative.
>
>
> **References**
>
> [1] Atanackovic, Lazar, et al. "Meta Flow Matching: Integrating Vector Fields on the Wasserstein Manifold."
>
> [2] Sakalyan, Kristiyan, et al. "Modeling microenvironment trajectories on spatial transcriptomics with nicheflow."

---

> ### Author Response · Authors · 2025-11-24
> **Response (4/4)**
>
> # [P5] Stochastic dynamics
> Thank you for the suggestion of considering stochastic dynamics. Following your suggestion, we have added $\texttt{IADOT + SF2M}$, a variant of $\texttt{IADOT}$ which models stochastic dynamics, based on the SF2M framework [1].  We provide details about this variant in `Appendix D.13`.
>
> **Results.** In `Table 2`, we note two things. First, the baseline SF2M underperforms compared to the CFM baseline (which models deterministic dynamics). Second, we find that the velocity field obtained with $\texttt{IADOT+SF2M}$ yields lower interpolation errors than the one based on vanilla SF2M, confirming that $\texttt{IADOT}$ introduces a useful prior regardless of how the dynamics are instantiated.
>
> **Action taken:** Conducted an additional experiment with the baseline SF2M and the variant $\texttt{IADOT + SF2M}$, with results reported in `Table 2`.
>
> **Reference**
>
> [1] Tong, Alexander, et al. "Simulation-free schrodinger bridges via score and flow matching."
>
> ---
> ### Thank you for your feedback
> Thank you for your detailed and constructive feedback, which has greatly helped us to refine our manuscript. We believe the revisions, including the new baselines (MFM, SF2M) and their $\texttt{IADOT}$ variants, the noise sensitivity analysis, and the clarification on applicability *substantially improve* the clarity and rigor of our submission. We are confident that $\texttt{IADOT}$, by tackling the **crucial question of: "What is a good cost function to infer the transport map?"**, provides a practical step toward more biology-aligned methods in scRNA analysis and welcome continued dialogue.

---

### Author Response · Authors · 2025-11-24
**Global response**

We thank the reviewers for their constructive comments and their detailed assessment of our work.

We are encouraged that reviewers recognized $\texttt{IADOT}$ as tackling a "significant and well-motivated problem" (**SwiP**), noting that it "moves beyond the common and limiting assumption" (**QVyK**) of treating cells as independent particles. We are glad they found the proposed extension to FGW "conceptually coherent" (**W1vz**) and the overall formulation "well-presented" (**ja6u**).

On the experimental side, reviewers highlighted the "clear validation of [our] hypothesis" (**SwiP**) regarding structure-aware priors. They noted that our "comprehensive analysis" (**QVyK**) successfully demonstrates that incorporating cell-cell interactions can "refine dynamic modelling" (**SwiP**) and leads to "superior performance" (**W1vz**) in trajectory inference.

We have also taken the reviewers’ feedback into account and made the following key improvements to the paper:

* **Significantly expanded empirical evaluation:** Our revised manuscript now contains comparisons against **9 baselines** (including MFM, UOT-FM, VGFM, SF2M, and MOSCOT)  (`Table 2`, `Sec 5.3`).
* **Clarified methodological novelty:** Explicitly differentiated our **multi-channel, directed biological prior** from generic graph regularization and standard FGW approaches (`Sec 3`).
* **Developed IADOT extensions:** Formulated, implemented and evaluated additional $\texttt{IADOT}$ variants for **unbalanced mass** (`App D.11`), **stochastic bridges** (`App D.13`), and **geometric interpolants** (`App D.12`).
* **Added rigorous sensitivity analyses:** Investigated robustness to LR expression noise (`App E.5`), compared against generic signaling catalogs (`App E.3`), and analyzed Hill function hyperparameters (`App E.6`).
* **Strengthened theoretical and technical grounding:** Added a formal **proof of identifiability** (`App F`), a curvature analysis of inferred trajectories (`App E.7`), and a detailed **computational cost analysis** (`App D.10`).


We have highlighted these changes **in teal** in the updated manuscript. We sincerely thank the reviewers for their valuable feedback on strengthening our work and remain open to further suggestions.

With thanks,

The Authors of #20800

---

### Meta-Review · Area_Chair_YSrw · 2026-01-03

**Summary:**

The paper studies trajectory inference from scRNA-seq snapshot data by proposing IADOT, a method that injects directed, multi-typed ligand–receptor cell–cell interaction structure into an FGW-style optimal transport coupling and then learns continuous-time dynamics via conditional flow matching, with additional perturbation and sensitivity analyses. From a biological perspective, this is an interesting and novel direction: explicitly encoding interaction-aware priors grounded in ligand–receptor biology is well motivated, and the large set of empirical and perturbation studies suggests potential value for interpreting developmental processes. From a machine learning perspective, however, the contribution is more debatable, and several major weaknesses remain despite a substantial rebuttal effort. First, the methodological novelty largely lies in the design of the cost and prior rather than in new learning or optimization principles, as the core components (FGW-style OT and flow matching) are established; while the rebuttal convincingly broadens the baseline set and clarifies the distinctiveness of the multi-channel interaction prior and solver, the advance still reads more as a sophisticated application or extension than as a fundamentally new ML method. Second, the empirical comparisons, although significantly expanded in the rebuttal to include modern interacting-cell and unbalanced OT baselines, remain sensitive to implementation and hyperparameter choices, with some baselines performing unexpectedly poorly and requiring detailed tuning explanations to justify the results. Third, key modeling assumptions remain only partially resolved: interaction information enters primarily through endpoint couplings rather than being enforced along the entire inferred trajectory, and the biological realism of inferred cell–cell interactions without spatial ground truth remains uncertain, even though added noise and perturbation experiments show that the prior is nontrivial. Finally, while the authors added sensitivity analyses and practical guidance for the interaction-weight parameter α and clarified that the method degrades gracefully in regimes where interaction priors are uninformative, this also highlights that turning off the prior is often the fallback in challenging settings rather than a principled solution. In conclusion, the authors clearly invested a major rebuttal effort and substantially strengthened the experimental coverage and presentation, and the work is compelling and creative from a biological modeling standpoint, but from an ML standpoint the novelty, generality, and methodological depth remain insufficiently convincing, and the paper would require an in-depth rewrite and sharper positioning to be acceptable.

**Reviewer Concerns:**

Several substantive concerns were meaningfully addressed in the rebuttal. In particular, the authors broadened and updated the baseline suite (adding modern FM, unbalanced, stochastic, and OT-dynamics baselines and IADOT “plug-in” variants), provided a clearer positioning versus closely related work (including interacting-cell and geometry-aware FM lines), and added additional sensitivity analyses (LR noise, catalog perturbations, and tuning guidance for the structure weight α), along with runtime/memory discussion. These steps respond well to requests about missing baselines, robustness, and computational cost. However, key issues remain outstanding at the level of scientific conviction rather than missing evidence: the work still hinges on a cost/prior design layered onto established FGW/CFM machinery, the comparative results appear sensitive to implementation/hyperparameters (with some baselines behaving unexpectedly poorly, even if partially explained), and the core modeling limitation noted by reviewers (interaction structure affecting endpoint couplings more than being enforced throughout trajectories) is only partially resolved. Finally, despite nontrivial perturbation/noise results, the biological realism of inferred CCI structure from dissociated scRNA-seq without spatial ground truth remains uncertain and is not fully settled by the rebuttal.

**Reviewer Scores:**

- [W1vz] (initial score: 4): Likely a modest increase. Their main asks (recent related work and baselines, LR-quality sensitivity, computational cost, applicability regimes, stochastic setting, and combining with other FM paradigms) were directly addressed with added baselines/variants, LR-noise experiments, runtime discussion, and applicability clarification (including “α near 0” fallback on embryo-like data).
- [SwiP] (initial score: 6): Likely no change, possibly a slight increase if persuaded by the expanded baselines and unbalanced extension, but tempered by their follow-up skepticism about suspicious baseline results and the need for careful hyperparameter justification (which the authors attempted to clarify).
- [QVyK] (initial score: 4): Likely no change (remain near 4). The rebuttal addressed several concrete requests (more recent baselines including MFM, α-tuning guidance, cost discussion, and dataset naming clarifications), but their central objections (limited methodological novelty beyond FGW/CFM usage, concerns about linear interpolations for interacting systems and endpoint-only constraints, and doubts about biological validity of CCI without spatial context) would plausibly remain.

---

### Decision · Program_Chairs · 2026-01-26

Reject